# The genome and population genomics of allopolyploid *Coffea arabica* reveal the diversification history of modern coffee cultivars

**A list of authors and their affiliations appears at the end of the paper**

*Coffea arabica*, an allotetraploid hybrid of *Coffea eugenioides* and *Coffea canephora*, is the source of approximately 60% of coffee products worldwide, and its cultivated accessions have undergone several population bottlenecks. We present chromosome-level assemblies of a di-haploid *C. arabica* accession and modern representatives of its diploid progenitors, *C. eugenioides* and *C. canephora*. The three species exhibit largely conserved genome structures between diploid parents and descendant subgenomes, with no obvious global subgenome dominance. We find evidence for a founding polyploidy event 350,000–610,000 years ago, followed by several pre-domestication bottlenecks, resulting in narrow genetic variation. A split between wild accessions and cultivar progenitors occurred ~30.5 thousand years ago, followed by a period of migration between the two populations. Analysis of modern varieties, including lines historically introgressed with *C. canephora*, highlights their breeding histories and loci that may contribute to pathogen resistance, laying the groundwork for future genomics-based breeding of *C. arabica*.

Polyploidy is a powerful evolutionary force that has shaped genome evolution across many eukaryotic lineages, possibly offering adaptive advantages in times of global change[1,2]. Such whole-genome duplications (WGDs) are particularly characteristic of plants[3], and a great proportion of crop species are polyploid[4–11]. Our understanding of genome evolution following WGD is still incomplete, but outcomes can include genomic shock, in terms of activation of cryptic transposable elements (TEs), subgenome-partitioned gene regulation or fractionation, homoeologous exchange (HE), meiotic instability and even karyotype variation[8,12–16]. Alternatively, few or none of the above phenomena can materialize, and the two subgenomes can coexist harmonically, gradually adapting to new ploidy levels[17]. Regardless, the most common fate of polyploids appears to be fractionation and eventual reversion to the diploid state[18].

With an estimated production of 10 million metric tons per year, coffee is one of the most traded commodities in the world. The most broadly appreciated coffee is produced from the allotetraploid species *Coffea arabica*, especially from cultivars belonging to the Bourbon or Typica lineages and their hybrids[19]. *C. arabica* ($2n = 4x = 44$ chromosomes) resulted from a natural hybridization event between the ancestors of present-day *Coffea canephora* (Robusta coffee, subgenome CC (subCC)) and *Coffea eugenioides* (subgenome EE (subEE)), each with $2n = 2x = 22$. The founding WGD has previously been dated to between 10,000 and 1 million years ago[20–23], with the Robusta-derived subgenome of *C. arabica* most closely related to *C. canephora* accessions from northern Uganda[24]. Arabica cultivation was initiated in fifteenth- to sixteenth-century Yemen (Extended Data Fig. 1). Around 1600, the so-called seven seeds were smuggled out of Yemen[25], establishing Indian *C. arabica* cultivar lineages. A century later, the Dutch began cultivating Arabica in Southeast Asia—thus setting up the founders of the contemporary Typica group. One plant, shipped to Amsterdam

✉e-mail: jarkko@ntu.edu.sg; vaalbert@buffalo.edu; dcrouzillat@gmail.com; alexandre.dekochko@gmail.com; patrick.descombes@rd.nestle.com

**Table 1 | Statistics of the *Coffea* assemblies presented in this paper**

| Assembly | *C. eugenioides* | *C. canephora* | *C. arabica* | *C. arabica* HiFi |
|---|---|---|---|---|
| Projected genome size (Mb)[a] | 682 | 705 | 1,281 | 1,281 |
| Total assembly length (Mb) | 661 | 672 | 1,088 | 1,198 |
| % of projected genome | 96.9% | 95.3% | 84.9% | 93.5% |
| *N* scaffolds | 253 | 3,033 | 8,474 | 132 |
| Scaffold N50 | 61.3 Mb | 50.1 Mb | 32.7 Mb | 53.7 Mb |
| *N* contigs | 5,736 | 3,757 | 11,863 | 238[b] |
| Contig N50[c] (Mb) | 0.40 | 1.35 | 0.23 | 30.0 |
| Pseudochromosomes (Mb) | NA[d] | 583 | 801 | 1,192 |
| % of projected genome | NA | 82.7% | 62.5% | 93.1% |
| *N* genes | 32,192 | 28,880 | 56,670 | 69,314 |
| Genes in pseudochromosomes | NA | 27,881 | 50,410 | 69,067 |
| % genes in pseudochromosomes | NA | 97% | 89% | 99.6% |
| BUSCO genome | | | | |
| Complete | 96.7% | 97.4% | 97.6% | 97.9% |
| Single | 88.5% | 94.8% | 20.1% | 4.3% |
| Duplicated | 8.2% | 2.6% | 77.5% | 93.6% |
| Fragmented | 1.1% | 0.9% | 0.8% | 0.8% |
| Missing | 2.2% | 1.7% | 1.6% | 1.3% |
| Total | 2,326 | 2,326 | 2,326 | 2,326 |
| BUSCO annotation | | | | |
| Complete | 94.9% | 96.2% | 92.1% | 97.3% |
| Single | 82.4% | 92.8% | 33.3% | 4.1% |
| Duplicated | 12.5% | 3.4% | 58.8% | 93.2% |
| Fragmented | 2.1% | 1.5% | 2.8% | 0.8% |
| Missing | 3.0% | 2.3% | 5.1% | 1.9% |
| Total | 2,326 | 2,326 | 2,326 | 2,326 |

[a]From the plant DNA C-values database: https://cvalues.science.kew.org/. [b]After gap filling. [c]The length of the shortest contig for which longer and equal-length contigs cover at least 50% of the assembly. [d]Not applicable.

in 1706, was used to establish Arabica cultivation in the Caribbean in 1723. Independently, the French cultivated Arabica on the island of Bourbon (presently Réunion)[26], and the descendants of a single plant that survived by 1720 form the contemporary Bourbon group. Contemporary Arabica cultivars descend from these Typica or Bourbon lineages, except for a few wild ecotypes with origins in natural forests in Ethiopia. Due to its recent allotetraploid origin and strong bottlenecks during its history, cultivated *C. arabica* harbors a particularly low genetic diversity[20] and is susceptible to many plant pests and diseases, such as coffee leaf rust (*Hemileia vastatrix*). As a result, the classic Bourbon–Typica lineages can be cultivated successfully in only a few regions around the world. Fortunately, a spontaneous *C. canephora* × *C. arabica* hybrid resistant to *H. vastatrix* was identified on the island of Timor[27] in 1927. Many modern Arabicas contain *C. canephora* introgressions derived from this hybrid, ensuring rust resistance, but having also unwanted side effects, such as decreased beverage quality[28].

Modern genomic tools and a detailed understanding of the origin and breeding history of contemporary varieties are vital to developing new Arabica cultivars, better adapted to climate change and agricultural practices[29–31]. Here, we present chromosome-level assemblies of *C. arabica* and representatives of its progenitor species, *C. canephora* (Robusta) and *C. eugenioides* (hereafter Eugenioides). Whole-genome resequencing data of 41 wild and cultivated accessions facilitated in-depth analysis of Arabica history and dissemination routes, as well

as the identification of candidate genomic regions associated with pathogen resistance.

## Results

### The genomes of *C. arabica*, *C. canephora* and *C. eugenioides*

As reference individuals, we chose the di-haploid Arabica line ET-39 (ref. 32), a previously sequenced doubled haploid Robusta[33] and the wild Eugenioides accession Bu-A, respectively. Long- and short-read-based hybrid assemblies were obtained (Methods and Supplementary Sections 2.1 and 2.2), spanning 672 megabases (Mb) (Robusta), 645 Mb (Eugenioides) and 1,088 Mb (Arabica), respectively. Upon Hi-C scaffolding, the Robusta and Arabica assemblies consisted of 11 and 22 pseudochromosomes, and spanned 82.7% and 62.5%, respectively, of the projected genome sizes (Table 1). To improve the Arabica assembly, we generated a second assembly using Pacific Biosciences (PacBio) HiFi technology followed by Hi-C scaffolding (Methods and Supplementary Sections 2.2 and 2.3). This assembly was 1,198 Mb long, of which 1,192 Mb (93.1% of the predicted genome size based on cytological evidence[34]) was anchored to pseudochromosomes (Table 1). Gene space completeness, assessed using Benchmarking Universal Single-Copy Orthologs (BUSCOs)[35], was >96% for all assemblies. Importantly, 93.2% of the BUSCO genes were duplicated in the HiFi assembly (Table 1), indicating that most of the gene duplicates from the allopolyploidy event were retained.

The Robusta and Eugenioides genomes contained, respectively, 67.5% and 59.7% TEs (Supplementary Section 3.2), with Gypsy long

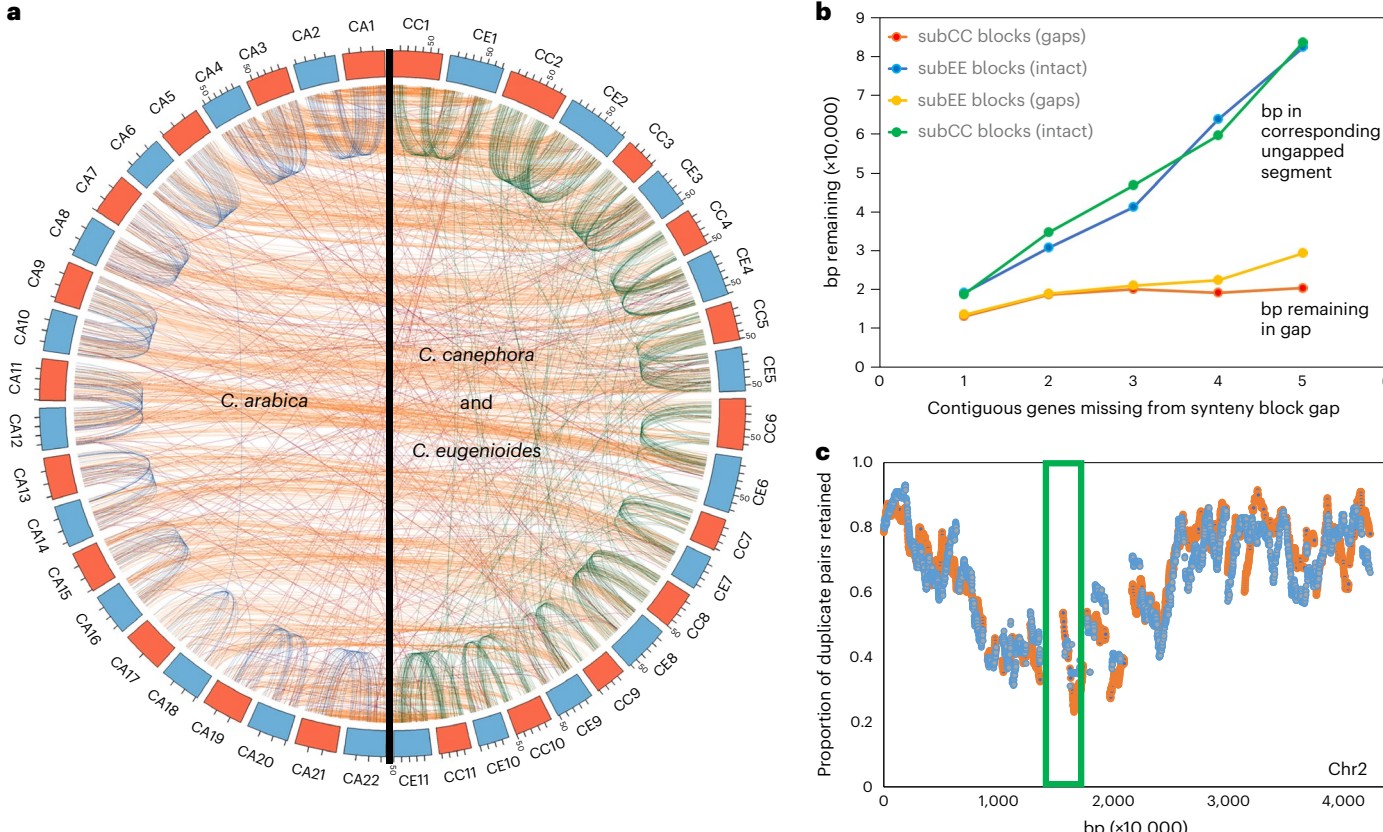

**Fig. 1 | Patterns of synteny, fractionation and gene loss in *C. arabica* and its progenitor species *C. canephora* and *C. eugenioides*. a**, Corresponding syntenic blocks between CA subgenomes subCC (orange) and subEE (blue), and with the CC (orange) and CE (blue) genomes. **b**, The base pairs in intergenic DNA in synteny block gaps caused by fractionation in a subCC–subEE comparison, compared with numbers of base pairs in homoeologous unfractionated regions, as a function of numbers of consecutive genes deleted. **c**, Gene retention rates in synteny blocks plotted along subCC chromosome 2; subCC is plotted in orange and subEE in blue. The green box indicates the pericentromeric region. CA, *C. arabica*; CC, *C. canephora*; CE, *C. eugenioides*.

terminal repeat (LTR) retrotransposons accounting for most of the difference between the two species. This difference was greatly reduced (63.1% and 63.8%) in the two Arabica subgenomes (subCC and subEE, stemming from Robusta and Eugenioides ancestors, respectively), possibly indicating TE transfer via HE. Robusta contained considerably more recent LTR TE insertion elements than Eugenioides. Again, the two Arabica subgenomes showed greater similarity to each other in recent LTR TE insertions than the two progenitor genomes. No major evidence was found for LTR TE mobilization following Arabica allopolyploidization, in contrast to what has been observed in tobacco[36], but similar to *Brassica* synthetic allotetraploids[37]. Observed Arabica genome evolution instead more closely follows the 'harmonious coexistence' pattern[38] seen in *Arabidopsis* hybrids[17,39].

High-quality gene annotations, followed by manual curation of specific gene families (Supplementary Sections 3.1–3.4), resulted in 28,857, 32,192, 56,670 and 69,314 gene models for the Robusta, Eugenioides, PacBio Arabica and Arabica HiFi assemblies, respectively (Table 1). Altogether, ~97% of Robusta and 99.6% of Arabica HiFi gene models were placed on the pseudochromosomes, with 33,618 and 35,449, respectively, to subgenomes subCC and subEE (Table 1). Annotation completeness from BUSCO was ≥95% for Eugenioides and Robusta, and reached 97.3% for Arabica HiFi.

## Genome fractionation and subgenome dominance
Comparison of Arabica subCC and subEE against their Robusta and Eugenioides counterparts revealed high conservation in terms of chromosome number, centromere position and numbers of genes per chromosome (Fig. 1 and Supplementary Section 4). Patterns of gene loss following the *gamma* paleohexaploidy event displayed high structural conservation between Robusta and Eugenioides during the 4–6 million years since their initial species split[22,23] (Supplementary Section 4). Likewise, the structures of the two Arabica subgenomes were highly conserved between each other, with, since the Arabica-founding allotetraploidy event, only ~5% of BUSCO genes having reverted to the diploid state (Fig. 1a and Table 1). Syntenic comparisons revealed that genomic excision events, removing one or several genes at a time in similar proportions across the two subgenomes, have been the main driving force in genome fragmentation both before and after the polyploidy event (Fig. 1b and Supplementary Section 4). Fractionation occurred mostly in pericentromeric regions, whereas chromosome arms showed more moderate paralogous gene deletion (Fig. 1c and Supplementary Section 4). The Arabica allopolyploidy event seemingly did not affect the rate of genome fractionation, which remained roughly constant when comparing deletions in progenitor species versus Arabica subgenomes after the event. In support of the dosage-balance hypothesis[40], subgenomic regions with high duplicate retention rates were significantly enriched for genes that originated from the Arabica WGD (Fisher exact test, $P < 2.2 \times 10^{-16}$). In contrast, low duplicate retention rate regions significantly overlapped with genes originating from small-scale (tandem) duplications (Supplementary Table 1). Genes with high retention rates were enriched in Gene Ontology (GO) categories such as 'cellular component organization or biogenesis', 'primary metabolic process', 'developmental process' and 'regulation of cellular process', while low retention rate genes were enriched in categories such as 'RNA-dependent DNA biosynthetic process' and 'defense response' (in both subgenomes), and 'spermidine hydroxycinnamate conjugate

biosynthetic process' (involved in plant defense[41]) and 'plant-type hypersensitive response' (in subEE) (Supplementary Tables 2–5).

To study possible expression biases between subgenomes, we identified syntelogous gene pairs and removed the pairs showing HEs in the Arabica subgenomes (see under 'Origin and domestication of Arabica coffee' below)[42] (Supplementary Section 5). Overall, no significant global subgenome expression dominance was observed (Supplementary Tables 6 and 7). However, gene families regularly displayed mosaic patterns of expression, including several encoding enzymes that contribute to cup quality, such as *N*-methyltransferase (*NMT*), terpene synthase (*TPS*) and fatty acid desaturase 2 (*FAD2*) families, all having some genes being more expressed in one of the two subgenomes (Extended Data Fig. 2), as per a recent study[43]. Similar gene family-wise patterns occur in other evolutionarily recent polyploids such as rapeseed[10] and cotton[44], which are also at their early stages of transitioning back to a diploid state.

## Origin and domestication of Arabica coffee

To obtain a genomic perspective on the evolutionary history of Arabica, we sequenced 46 accessions, including three Robusta, two Eugenioides and 41 Arabica. The latter included an eighteenth-century type specimen, kindly provided by the Linnaean Society of London, 12 cultivars with different breeding histories, the Timor hybrid and five of its backcrosses to Arabica, and 17 wild and three wild/cultivated accessions collected from the Eastern and Western sides of the Great Rift Valley[45,46] (Supplementary Table 8 and Fig. 2a).

HE between subgenomes has been observed in several recent polyploids[8,10,42]. Arabica generally displays bivalent pairing of homologous chromosomes and disomic inheritance[47], but since the subgenomes share high similarity, occasional homoeologous pairing and exchange may also occur. We therefore explored the extent of HE among Arabica accessions and its possible contribution to genome evolution. Overall, all accessions shared a fixed allele bias toward subEE at one end of chromosome 7, which contained genes enriched for chloroplast-associated functions (Extended Data Fig. 3a, Supplementary Section 5 and Supplementary Table 9). Since the Arabica plastid genome is derived from Eugenioides[48], HE in this region was likely selected for, due to compatibility issues between nuclear and chloroplast genes encoding chloroplast-localized proteins[49]. Surprisingly, all but one accession (BMJM) showed significant (Bonferroni-adjusted *P* values < 0.0005; chi-squared test, each d.f. = 1) 3:1 allelic biases toward subCC. The highly concordant HE patterns, present in both wild and cultivated Arabicas (Extended Data Fig. 4), suggested that (1) the allelic bias is an adaptive trait not associated with breeding and (2) it originated in a common ancestor of all sampled accessions, possibly immediately after the founding allopolyploidy event. Some exchanges, shared by only a few accessions, probably originated more recently (Extended Data Fig. 3b). More recent HE events were also found in some cultivars and also showed a bias toward subCC, except for BMJM, which showed bias toward subEE due to a single large crossover in chromosome 1 (Extended Data Fig. 3a). An interesting hypothesis for future investigation is that in a low-diversity polyploid species such as Arabica, HE could be a major contributor to phenotypic variation observed among closely related accessions[50].

We next studied population genetic statistics for each of the subgenomes (Supplementary Table 10). The 17 wild samples demonstrated low genomic diversities, indicative of small effective population sizes, while negative Tajima's *D* suggested an expanding population, possibly following one or more population bottlenecks. The cultivars and wild population samples had similar genetic diversities, as demonstrated by low fixation index ($F_{ST}$) values. In cultivars, nucleotide diversities were only slightly lower than in wild populations and Tajima's *D* scores were less negative, suggesting that only minor bottlenecks and subsequent population expansions occurred during domestication.

SNP tree estimation and ADMIXTURE analyses (Fig. 2b) identified a three-population solution for subCC: Typica–Bourbon

cultivars (Population 1), wild accessions (Population 2), and Timor hybrid-derived cultivars (Population 3). The old BMJM and the recently established Geisha cultivars showed admixed states on both subgenomes, similar to about half of the wild accessions. Indian varieties encompassed both Typica and Bourbon variation, in agreement with previous studies[20]. The Linnaean sample grouped with the cultivars, supporting its hypothesized origin from the Dutch East Indies[25]. A complementary principal component analysis (PCA) (Extended Data Fig. 5) was in agreement with ADMIXTURE analysis.

In wild accessions, both subgenomes concordantly showed two population bottlenecks (Fig. 2d) in the SMC++ (ref. 51) modeling. Assuming a 21-year generation time[52], the oldest bottleneck initiated abruptly around 350 thousand years ago (ka) and ended around 15 ka, at the start of the African humid period[53], when climatic conditions were more favorable for Arabica growth. The more recent bottleneck initiated more gradually around 5 ka and lasts to this day. Cultivated accessions, however, exhibited the oldest, but not the more recent, bottleneck. In part due to these differences, we also modeled Arabica population history using FastSimcoal2 (ref. 54), modeling the wild population and cultivars as two separate lineages. In the best-fitting model (Fig. 2e), the wild population was predicted to split from the cultivar founding population 1,450 generations ago (~30 ka), that is, before the last glacial maximum. The original founding event was analyzed using the nonadmixed wild individuals, revealing an ancestral population bottleneck at 350 ka (Extended Data Fig. 6a). Divergence estimates based on gene fractionation, the distribution of nonsynonymous mutations (Extended Data Fig. 6b) and calibrated SNP trees (Fig. 2b) suggested the allopolyploid founding event occurred at 610 ka, which is close to previous estimates[22,23]. The 350 ka bottleneck, on the other hand, corresponds to that found in the SMC++ analyses (Fig. 2d). We therefore consider 610–350 ka a likely time range for the polyploidization event (Fig. 2e). The wild and pre-cultivar lineages maintained some gene flow (in terms of migration) until ~8–9 ka, which may have contributed to the modeled increase in effective population size (Fig. 2d,e).

While these data were not able to identify the precise place of origin of the modern cultivated population (see also the following section), the extended period of migration between wild and cultivated accessions suggests that they were separated only by a relatively small geographic distance, such as along the two sides of the African Great Rift Valley (Fig. 2a–c). It is also possible that the cultivated lineage could have extended as far as Yemen and that the end of migration between the two populations could have been caused by the widening of the Bab al-Mandab strait (separating Yemen and Africa) due to rising sea levels[55] at the end of the African humid period. A native Arabica population exists in Yemen[56], which could support this hypothesis. The Linnaean sample, together with the Typica and Bourbon cultivars, originates from this second population, which was also used to establish cultivation in Yemen, as suggested by the SNP, ADMIXTURE and PCA analyses (Fig. 2b and Extended Data Fig. 5).

In conclusion, our analyses suggest that the Arabica allopolyploidy event occurred between 610 and 350 ka, when considering that inbreeding present in *Coffea* populations would accelerate coalescence estimation[57,58]. Earlier work proposing more recent timings, such as 20 ka (ref. 20), could be underestimates stemming from confounding effects of population bottlenecks in cultivated and wild lineages.

## Origin of modern cultivars

The known breeding history of several of our Arabica cultivars provided us with a gold standard set for deducing the Arabica pedigree using Kinship-based INference for Gwas (KING)[59] (Fig. 3). The method correctly identified the relationships between Bourbon and Typica group cultivars and the Bourbon–Typica crosses in subCC. In contrast, the subEE pedigree showed lower (second) order relationships, possibly due to HE in that subgenome (Extended Data Fig. 7).

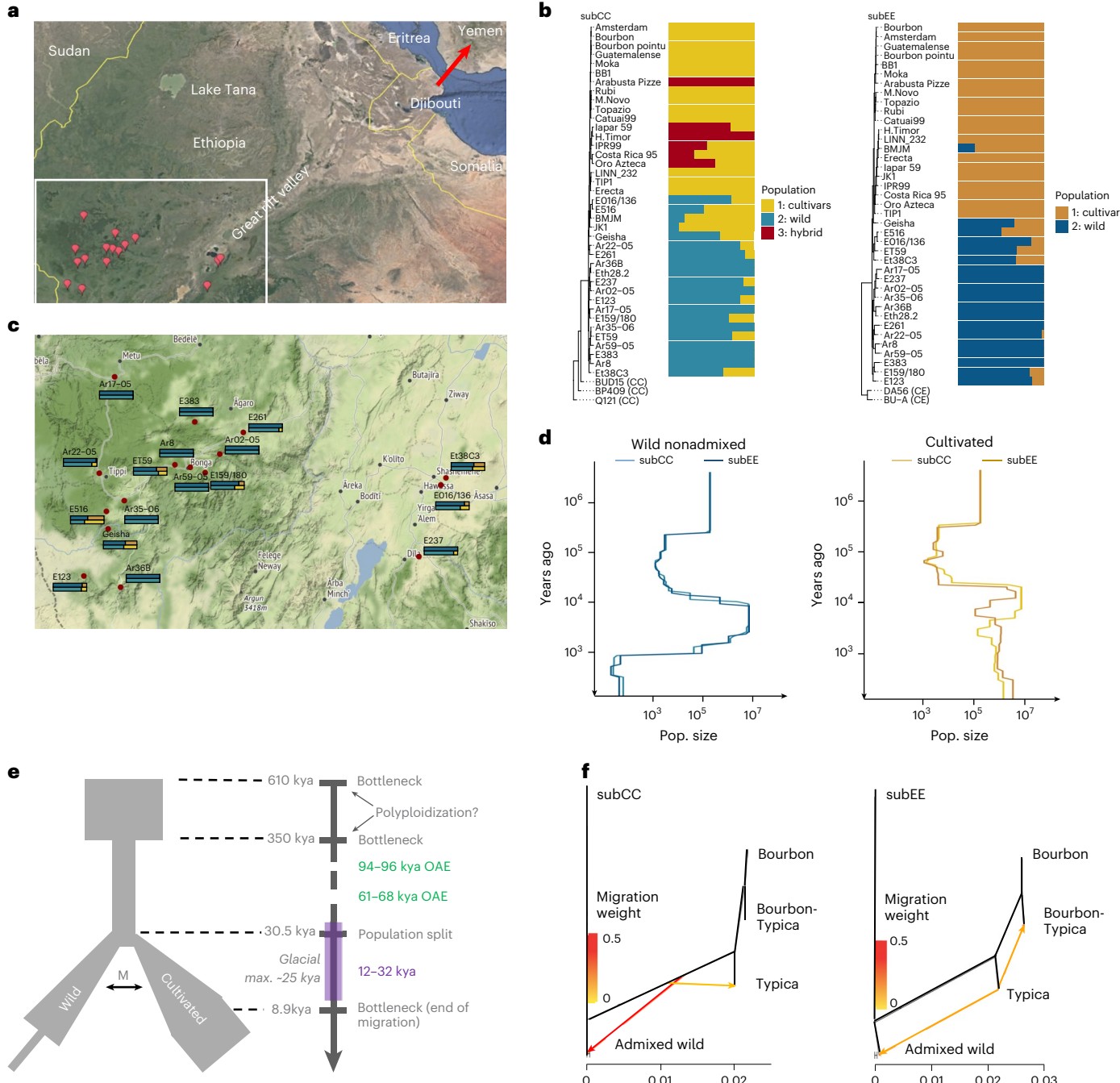

**Fig. 2 | Population history of *C. arabica*. a**, Geographic origin of resequenced wild *C. arabica* accessions (red placeholders). Accession names are given in **c**. The red arrow indicates the probable route of migration to Yemen in historical times. **b**, Ancestral population assignments of *C. arabica* accessions for subCC (left) and subEE (right). Relationships among individuals are illustrated with phylogenetic trees obtained from independent SNPs. For magnified views of the trees, see Supplementary Fig. 37. **c**, Magnification of the bottom left part of **a**, showing the admixture values for each of the accessions in subCC (top) and subEE (bottom); the colors correspond to the analysis in **b**. **d**, Population sizes of wild and cultivated accessions, inferred using SMC++, suggest genetic bottlenecks at ~350 and 1 ka (limited to nonadmixed wild individuals). **e**, FastSimcoal2 output,

suggesting a population split ~30.5 ka, followed by a period of migration between the populations until ~8.9 ka. This timing corresponds with increased population diversity in cultivars at a similar time, calculated using SMC++. Green rectangles along the timeline show 'windows of opportunity', times when Yemen was connected with the African continent wherein human migrations to the Arabian Peninsula may have occurred. The purple rectangle shows the last ice age. M, migration; OAE, out-of-Africa event. **f**, Directional gene flow analysis using Orientagraph suggests two hypotheses: gene flow from the shared ancestral population of all cultivars to the Ethiopian wild individuals (subCC), or gene flow from the Typica lineage to Ethiopia (subEE). Maps in **a** and **c** were generated with Google Earth and Google Maps, respectively.

Timor hybrid-derived accessions did not show significant relationships to mainline cultivars in subCC (likely due to Robusta introgressions in this subgenome that broke the haplotype blocks; see below), while subEE showed second-degree relationships to both the Typica and

Bourbon groups (Fig. 3 and Extended Data Fig. 7), confirming that subEE has not received substantial introgression.

Interestingly, the Typica, Bourbon and JK1 individuals were also first degree related, suggesting direct parent–offspring relationships.

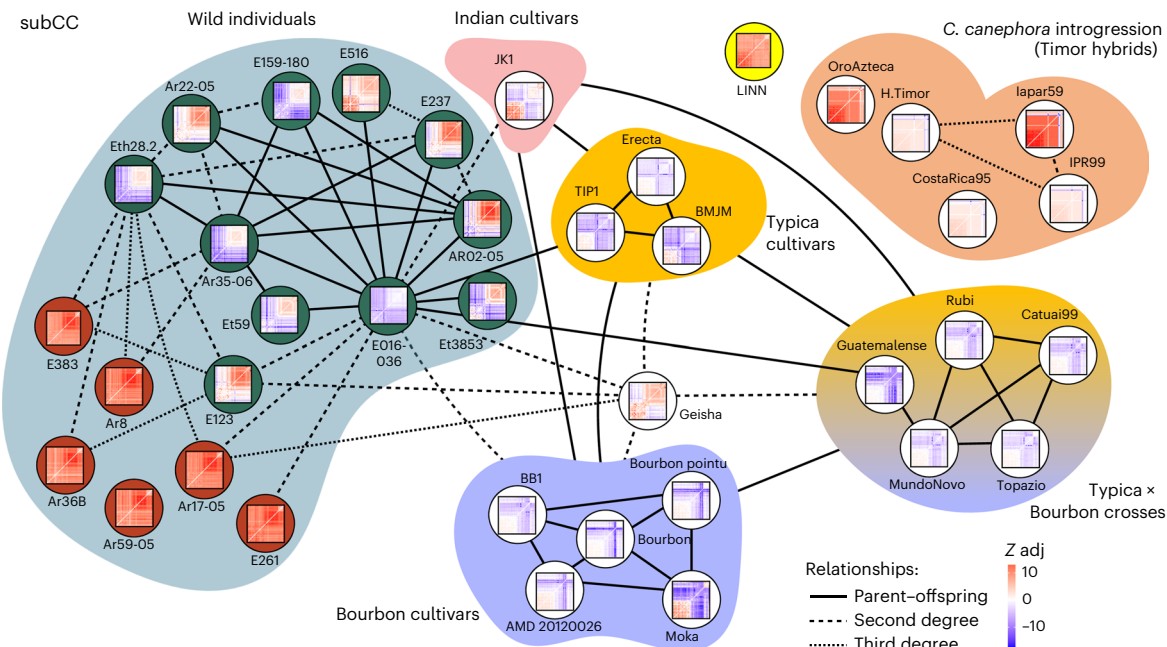

**Fig. 3 | Kinship estimation of *C. arabica* accessions, inferred from SNPs in subCC.** The degree of relatedness was estimated using KING and describes the number of generations between the related accessions. Thumbnail images show FDR-corrected F3 tests of introgression for each of the target individuals. Each cell in the matrix illustrates an F3 test result for the target accession containing introgression from two different sources (*x* and *y* axes); blue color illustrates significant adjusted *Z*-score (*Z* adj; value indicated by color key), indicative of gene flow (or allele sharing via identity by descent[78]) from the two source accessions to the target, while red color illustrates no support for gene flow. See Extended Data Fig. 7 for corresponding analyses in subEE. In the wild accessions, the dark green background highlights the admixed individuals (Fig. 2b), while the nonadmixed individuals are highlighted with red background. Relationships follow standard nomenclature (for example, second degree refers to an individual's grandparents, grandchildren and so on, whereas third degree refers to great-grandparents, great-grandchildren and so on).

Besides confirming their shared Yemeni origins, this finding also underscores the Yemeni germplasm's limited genetic diversity. Further, the old cultivar lines JK1 (Indian), Erecta (Indonesian Typica), BMJM (Caribbean Typica), TIP1 (Brazilian Typica) and BB1 (Brazilian Bourbon) showed second- or higher-degree relationships with a cluster of closely related wild admixed accessions, centered on E016/136 (Fig. 2b). The recently established Geisha cultivar showed similar relationships to the wild admixed individuals and the Bourbon and Typica groups, suggesting common origins. Interestingly, admixed wild accession E016/136 was closely related to both wild and cultivated populations.

In a comparison of geographic origins, wild individuals from the Eastern side of the Great Rift Valley had some levels of admixture and were closely interrelated, while on the Western side, the admixed, related individuals were mostly concentrated around the Gesha region (Figs. 2c and 3). The E016/136 admixed accession, closest to cultivars, demonstrated a first-degree relationship with several wild accessions, of which only Ar35-06 and Eth28.2 were pure representatives of the wild population (Fig. 2b). Therefore, these two accessions are genetically closest, in Fig. 2b, to the hypothetical true wild parent of cultivated Arabica, with E016/136 representing an intermediate form. Ar35-06 was collected near Gesha mountain, close to the origin of the modern Geisha cultivar. Altogether, these data point to the Gesha region as a hotspot of wild accessions amenable to domestication.

Admixed wild samples may have originated from a recent hybridization event that occurred before or after their collection from the wild. A third alternative is that the Yemeni population (and hence the cultivars) originated from an admixed population from the Eastern side of the Great Rift Valley or the Gesha region. Analysis of admixture patterns with Orientagraph[60] (Fig. 2f) suggested hybridization with the common ancestor of the Bourbon and Typica lineages in subCC, and of Typica in subEE. In the case of recent hybridization, introduced

haplotypes would exist as long contiguous blocks (as in the Timor hybridization, which occurred 100 years ago), while for older events, the blocks would be more fragmented due to crossing-over. Analysis using the distance fraction ($d_f$) statistic[61] showed the latter to be the case (Extended Data Fig. 8), indicating that admixture events among wild accessions were not very recent, supporting our third hypothesis.

Domestication and cultivation usually involve strong population bottlenecks based on high wild diversity, resulting in reduced genetic diversity in cultivars[62]. However, Arabica nucleotide diversity was already very low in the wild, probably as a result of earlier bottlenecks (Fig. 2d,e), but only marginally reduced in the pre-cultivated lineage (Extended Data Fig. 9a). Bourbon had lower diversity than Typica, probably resulting from the known single-individual bottleneck in this group. Also, the inbreeding coefficients in the wild and cultivated accessions were similar (Extended Data Fig. 9b), differing from general expectations for a domesticated species[62].

To look for pathways under purifying selection in cultivars, we identified genes with high $F_{ST}$ (95% quantile) between cultivars and wild accessions. This resulted in a set of 1,908 genes that were enriched for the GO categories 'cellular response to nitrogen starvation', 'regulation of innate immune response' and 'regulation of defense response' (Supplementary Table 11), and contained homologs of ammonium transporters *AMT1* and *AMT2*, important for nitrogen uptake in *Coffea*[63]; a homolog of the salicylic acid receptor *NONEXPRESSER OF PR GENES 1* (*NPR1*), required in salicylic acid signaling and systemic acquired resistance[64]; as well as a homolog of the *Arabidopsis LSU2* gene, previously identified as a hub convergently targeted by effectors of pathogens from different kingdoms[65]. A second screen, focused on genes with a large number of high-impact nonsynonymous mutations shared among cultivars (>40% individuals having the mutation), generated a list of 556 genes that were significantly enriched for only one GO category, 'defense response' (Supplementary Table 12). From the 22 genes in this

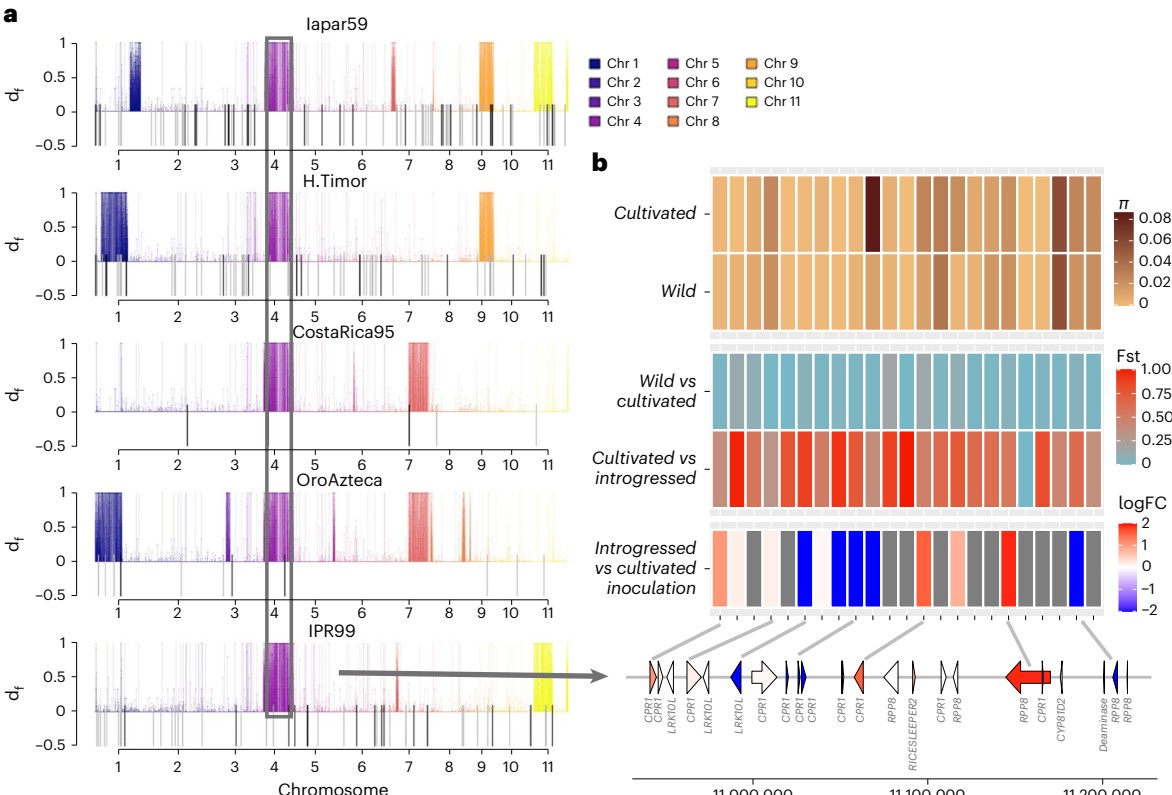

**Fig. 4 | Introgression of *C. canephora* into *H. vastatrix*-resistant *C. arabica* lineages. a**, Introgression $d_f$ statistic estimated for different Timor hybrid derivatives. Colored lines above the axis mark regions of significant introgression in the line under inspection, and are colored by chromosome. The shared introgressed region on chromosome 4 is colored in purple and boxed. TIPs are represented as lines below the *x* axis and exhibit overlap with introgressed regions. **b**, The shared introgressed genomic region on subCC chromosome 4 contains a cluster of R genes (*RPP8*), a cluster of homologs of a negative regulator of R genes (*CPR1*) and a cluster of homologs of Leaf rust resistance 10 kinases

(*LRK10L*) (bottom). The heatmap shows, from the bottom up, (1) log fold change of gene expression after *H. vastatrix* inoculation, when comparing resistant Timor hybrid lineage against a susceptible cultivar; red color means elevated expression in the hybrid, and blue decreased expression. (2) Fixation index ($F_{ST}$) values for the introgressed lines versus cultivars and between cultivars and wild accessions. (3) Nucleotide diversity for the wild and cultivated accessions for each gene coding region, plus the flanking 2 kb upstream and downstream of the region. FC, fold change.

category, 16 were NB-ARC domain-containing resistance (R) genes, and two were members of the leucine-rich repeat (LRR) defense gene family. High diversity in immune-related responses is one possible pathogen resistance mechanism in plant communities[66], and therefore reduced diversity may have compromised modern Arabica cultivar immunity.

The high level of conservation between the Arabica subgenomes and their diploid progenitors may have facilitated spontaneous interspecific hybridization events. This was the case for the Timor hybrid, a spontaneous Robusta × Arabica hybrid resistant to *H. vastatrix*[27]. Our sample set included five descendants of the original Timor hybrid, obtained by backcrossing to Arabica. As expected, the hybridization affected subCC more profoundly, with much higher levels of nucleotide divergence apparent ($F_{ST}$ = 0.185) than in subEE ($F_{ST}$ = 0.0897), when comparing cultivars and hybrids. The divergence from wild populations was even greater, with $F_{ST}$ = 0.254 for subCC and $F_{ST}$ = 0.138 for subEE, illustrating that introgression occurred almost exclusively within subCC.

In the Timor hybrids, the regions found with $d_f$ statistics[61] largely overlapped the introgressed loci identified using $F_{ST}$ scans (Fig. 4a) and were found in large blocks, reflecting recent hybridization, and covering 7–11% of the genome (Fig. 4a and Extended Data Fig. 8). Transposon insertion polymorphisms (TIPs) also overlapped with introgressed regions (Gypsy *P* = 0.0002; Copia *P* = 0.035; Fisher exact test), confirming their recent origin from Robusta (Fig. 4b). The introgressed regions overlapped with regions of higher subgenome fractionation (*P* = 0.001873; Supplementary Table 13), possibly due to heterologous

recombination between subCC and Robusta, resulting in unequal crossing-over.

An introgressed region shared by all Timor hybrid lines was evident on chromosome 4 (Fig. 4a). We identified a set of 233 genes shared by all hybrids (Supplementary Table 14). The set contained members of three colocalized tandemly duplicated blocks of resistance-related genes on chromosome 4, subCC, and showed high $F_{ST}$ values between cultivars and introgressed lines. A tandem array of five genes were homologs of *Arabidopsis RPP8*, a NOD-like receptor resistance locus conferring pleiotropic resistance to several pathogens[67,68]. *RPP8* shows a great amount of variation in *Arabidopsis* alone, where intrachromosomal gene conversion combined with balancing selection contributes to its exceptional diversity[69]. The same subCC region also included a tandem array of ten homologs of *CONSTITUTIVE EXPRESSER OF PR GENES 1* (*CPR1*), a negative regulator of defense response that targets resistance proteins[70,71]. Finally, we identified three duplicates encoding Leaf rust 10 disease-resistance locus receptor-like protein kinases (LRK10L). The LRK10L are a gene family that is widespread across plants. First identified as a protein kinase in a locus contributing leaf rust resistance in wheat[72], they were found to be upregulated during various biotic and abiotic stresses[73] and were confirmed as positive regulators of wheat hypersensitive resistance response to stripe rust fungus[73] and powdery mildew[74].

The high $F_{ST}$ values between cultivated and introgressed, but not wild, individuals (Fig. 4b) indicate that the wild population cannot be the source for allelic asymmetries. Nucleotide diversities further

illustrate this point; some genes demonstrate lower nucleotide diversity in wild individuals, suggesting these genes to have experienced selective sweeps. To further narrow down candidate genes involved in leaf rust resistance, we reanalyzed comparative gene expression data from susceptible and resistant accessions after *H. vastatrix* inoculation[75]. This analysis identified 723 differentially expressed genes, most of which were associated with defense responses (Fig. 4b and Supplementary Tables 14 and 15). The combination of high $F_{ST}$ values, nucleotide diversities and differential expression data highlights several strong candidate genes (one *RPP8*, six *CPR1* and one *LRK10L*) at this locus.

## Discussion

Besides providing genomic resources for molecular breeding of one of the most important agricultural commodities, our Arabica, Robusta and Eugenioides genomes provide a unique window into the genome evolution of a recently formed allopolyploid stemming from two closely related species. Our Arabica data did not suggest a genomic shock induced by allopolyploidy, but, instead, only higher LTR transposon turnover rate. Genome fractionation rates remained basically unaltered before and after the allopolyploidy event. Likewise, no global subgenome dominance in gene expression was observed, but rather a mosaic-type pattern as in other recent polyploids[10,44], affecting the expression of individual gene family members. However, similar to octoploid strawberry[8], we detected genome dominance in terms of biased HEs favoring subCC. Since Robusta has one of the widest geographic ranges in the *Coffea* genus, whereas Eugenioides is more range-limited, this biased HE might be adaptive. This hypothesis was supported by the site frequency spectrum of HE loci, showing signs of directional selection (Extended Data Fig. 3). Intriguingly, transposable insertion polymorphisms significantly overlapped with tandem gene duplications and biosynthetic gene clusters, hinting at their possible roles in cluster evolution.

Domestication of perennial species such as Arabica coffee differs markedly from that of annual crops, consisting instead of three phases: selection of outstanding genotypes from wild forests, clonal propagation and cultivation, and then breeding and diversification[76]. In addition to being a perennial crop, Arabica is also a predominantly autogamous allopolyploid, which puts it in a class of its own. We show here that genetic diversity was already very low among wild accessions, due to multiple pre-domestication bottlenecks, and that the genotypes selected for cultivation by humans (both the ancient cultivated Ethiopian landraces and the recent Geisha cultivar) already were somewhat admixed between divergent lineages. The resequenced accessions displayed a geographic split along the Eastern versus Western sides of the Great Rift Valley, with cultivated coffee variants all placed with the Eastern population. Such admixture has played a large role in breeding many fruit-bearing crops, the nonpolyploid allogamous perennial lychee being one of the most extreme cases[58].

The prevalent autogamy of Arabica, combined with the multiple genetic bottlenecks it underwent in the wild, may have selectively purged deleterious alleles, explaining the capacity of the species to survive single-plant bottlenecks that occurred during its cultivation. An additional element buffering deleterious alleles was probably Arabica's allopolyploidy itself, which provided some level of heterosis[77]. However, the narrow genetic basis of both cultivated and wild modern Arabica constitutes a major drawback, as well as an obstacle for its breeding using wild genepool diversity. On the other hand, the extensive collinearity of its CC and EE subgenomes with those of its Robusta and Eugenioides progenitors is likely to facilitate introgression of interesting traits from these species, as already happened historically in the Timor spontaneous hybrid. The high-quality genome sequences of the three species provided in this work, together with the identification of the genomic region conferring resistance to coffee leaf rust, constitute a cornerstone for the breeding of novel Arabica varieties with superior adaptability and pathogen resistance.

## Online content

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

Jarkko Salojärvi [1,2,3] ✉, Aditi Rambani [4,41], Zhe Yu [5,41], Romain Guyot [6,7,41], Susan Strickler [4,41], Maud Lepelley [8], Cui Wang [2], Sitaram Rajaraman [2], Pasi Rastas [9], Chunfang Zheng [5], Daniella Santos Muñoz [5], João Meidanis [10], Alexandre Rossi Paschoal [11], Yves Bawin [12], Trevor J. Krabbenhoft [13], Zhen Qin Wang [13], Steven J. Fleck [13], Rudy Aussel [8,14], Laurence Bellanger [8], Aline Charpagne [15], Coralie Fournier [15], Mohamed Kassam [15], Gregory Lefebvre [15], Sylviane Métairon [15], Déborah Moine [15], Michel Rigoreau [8], Jens Stolte [15], Perla Hamon [6], Emmanuel Couturon [6], Christine Tranchant-Dubreuil [6], Minakshi Mukherjee [13], Tianying Lan [13], Jan Engelhardt [16], Peter Stadler [16,17], Samara Mireza Correia De Lemos [18], Suzana Ivamoto Suzuki [19], Ucu Sumirat [20], Ching Man Wai [21], Nicolas Dauchot [22], Simon Orozco-Arias [7], Andrea Garavito [23], Catherine Kiwuka [24], Pascal Musoli [24], Anne Nalukenge [24], Erwan Guichoux [25], Havinga Reinout [26], Martin Smit [26], Lorenzo Carretero-Paulet [27], Oliveiro Guerreiro Filho [28], Masako Toma Braghini [28], Lilian Padilha [29], Gustavo Hiroshi Sera [30], Tom Ruttink [12,31], Robert Henry [32], Pierre Marraccini [33], Yves Van de Peer [31,34,35,36], Alan Andrade [37], Douglas Domingues [18], Giovanni Giuliano [38], Lukas Mueller [4], Luiz Filipe Pereira [39], Stephane Plaisance [40], Valerie Poncet [6], Stephane Rombauts [31,36], David Sankoff [5], Victor A. Albert [13] ✉, Dominique Crouzillat [8] ✉, Alexandre de Kochko [6] ✉ & Patrick Descombes [15] ✉

[1]School of Biological Sciences, Nanyang Technological University, Singapore, Singapore. [2]Organismal and Evolutionary Biology Research Programme, University of Helsinki, Helsinki, Finland. [3]Singapore Centre for Environmental Life Sciences Engineering, Nanyang Technological University, Singapore, Singapore. [4]Boyce Thompson Institute, Cornell University, Ithaca, NY, USA. [5]Department of Mathematics and Statistics, University of Ottawa, Ottawa, Ontario, Canada. [6]Institut de Recherche pour le Développement (IRD), Université de Montpellier, Montpellier, France. [7]Department of Electronics and Automation, Universidad Autónoma de Manizales, Manizales, Colombia. [8]Société des Produits Nestlé SA, Nestlé Research, Tours, France. [9]Institute of Biotechnology, University of Helsinki, Helsinki, Finland. [10]Institute of Computing, University of Campinas, Campinas, Brazil. [11]Department of Computer Science, The Federal University of Technology – Paraná (UTFPR), Cornélio Procópio, Brazil. [12]Plant Sciences Unit, Flanders Research Institute for Agriculture, Fisheries and Food (ILVO), Melle, Belgium. [13]Department of Biological Sciences, University at Buffalo, Buffalo, NY, USA. [14]Centre d'Immunologie de Marseille-Luminy, Aix Marseille Université, Marseille, France. [15]Société des Produits Nestlé SA, Nestlé Research, Lausanne, Switzerland.

[16]Department of Computer Science, University of Leipzig, Leipzig, Germany. [17]Interdisciplinary Center for Bioinformatics, University of Leipzig, Leipzig, Germany. [18]Group of Genomics and Transcriptomes in Plants, São Paulo State University, UNESP, Rio Claro, Brazil. [19]Centro de Ciências Agrárias, Universidade Estadual de Londrina, Londrina, Brazil. [20]Indonesian Coffee and Cocoa Research Institute (ICCRI), Jember, Indonesia. [21]University of Illinois at Urbana-Champaign, Urbana, IL, USA. [22]Research Unit in Plant Cellular and Molecular Biology, University of Namur, Namur, Belgium. [23]Departamento de Ciencias Biológicas, Facultad de Ciencias Exactas y Naturales, Universidad de Caldas, Manizales, Colombia. [24]National Agricultural Research Organization (NARO), Entebbe, Uganda. [25]Biodiversité Gènes & Communautés, INRA, Bordeaux, France. [26]Hortus Botanicus Amsterdam, Amsterdam, the Netherlands. [27]Departamento de Biología y Geología, Universidad de Almería, Almería, Spain. [28]Instituto Agronômico (IAC) Centro de Café 'Alcides Carvalho', Fazenda Santa Elisa, Campinas, Brazil. [29]Embrapa Café/Instituto Agronômico (IAC) Centro de Café 'Alcides Carvalho', Fazenda Santa Elisa, Campinas, Brazil. [30]Instituto de Desenvolvimento Rural do Paraná- IAPAR, Londrina, Brazil. [31]Department of Plant Biotechnology and Bioinformatics, Ghent University, Ghent, Belgium. [32]Queensland Alliance for Agriculture and Food Innovation, University of Queensland, Brisbane, Queensland, Australia. [33]CIRAD - UMR DIADE (IRD-CIRAD-Université de Montpellier) BP 64501, Montpellier, France. [34]Department of Biochemistry, Genetics and Microbiology, University of Pretoria, Pretoria, South Africa. [35]College of Horticulture, Academy for Advanced Interdisciplinary Studies, Nanjing Agricultural University, Nanjing, China. [36]Center for Plant Systems Biology, VIB, Ghent, Belgium. [37]Embrapa Café/Inovacafé Laboratory of Molecular Genetics Campus da UFLA-MG, Lavras, Brazil. [38]Italian National Agency for New Technologies, Energy and Sustainable Economic Development, ENEA Casaccia Research Center, Rome, Italy. [39]Embrapa Café/Lab. Biotecnologia, Área de Melhoramento Genético, Londrina, Brazil. [40]VIB Nucleomics Core, Leuven, Belgium. [41]These authors contributed equally: Aditi Rambani, Zhe Yu, Romain Guyot, Susan Strickler. ✉e-mail: jarkko@ntu.edu.sg; vaalbert@buffalo.edu; dcrouzillat@gmail.com; alexandre.dekochko@gmail.com; patrick.descombes@rd.nestle.com

## Methods

### Genome sequencing

For the three *Coffea* species, genomic DNA was extracted from leaf tissue. A Qiagen kit was used for DNA extraction for Illumina sequencing. Illumina short reads and PacBio 20-kilobase (kb) libraries were prepared following the manufacturer's instructions. Sequencing was performed on a HiSeq2000 instrument for the short reads, and the PacBio RSII platform for long reads (specifications given in Supplementary Table 16). For the generation of HiFi reads, DNA was extracted from *C. arabica* leaf tissue following nuclei purification by centrifugation followed by lysis, phenol–chloroform extraction and isopropanol precipitation. DNA was fragmented to 20 kb using a Megaruptor 3. SMRTbell libraries were sequenced on a single SMRTcell on a Sequel IIe platform.

For the resequencing of 39 wild and cultivated *C. arabica* accessions, libraries were prepared using the KAPA HyperPrep Kits (Roche) following the manufacturer's instructions, and paired-end (2 × 125) sequenced on an Illumina HiSeq2500 instrument to ~40× coverage. The Linnaean herbarium sample was sequenced to 46× coverage with Ion Torrent technology.

### Assembly

Contig-level assembly for *C. canephora* was obtained with MHAP[79] and scaffolded using BAC-end sequences and 454 paired-end sequences generated previously[33]. Both *C. eugenioides* and *C. arabica* were assembled with Falcon[80], and *C. arabica* was subsequently phased using Falcon_unzip. All three genomes were error-corrected with Pilon[81] using Illumina short reads (Supplementary Section 2.2). *C. canephora* and *C. arabica* were further scaffolded into pseudochromosomes using Dovetail Hi-C technology. For *C. eugenioides* no more material could be obtained for further improvement of the assembly contiguity, and the assembly was scaffolded into pseudomolecules using *C. canephora* as reference. Gaps in the scaffolds were filled with PBJelly[82], after which six more rounds of polishing were done with Pilon using the Illumina shotgun sequenced genomic DNA as well as RNA sequencing (RNA-seq) reads.

The resulting chromosome assemblies for *C. canephora* were checked and corrected using an ultra-high-density linkage map[83] generated during the project. To further improve the quality of the *C. arabica* assembly, Bionano genome maps were generated.

*C. arabica* HiFi assembly was carried out with hifiasm v.0.16.1 (ref. 84), followed by scaffolding using Hi-C data from Dovetail technology and ALLHiC[85] pipeline. Final quality checks and manual adjustments of the assembly were carried out using 3d-DNA[86] and juicebox[87].

The completeness of the different assemblies was assessed using BUSCO v.5.2.2 (ref. 35) with the eudicots_odb10 database (2,326 genes; Table 1). Telomeric repeats were searched across the chromosomes using CoGeBLAST[88].

To assess the phasing of both subgenomes from *C. arabica*, synonymous nucleotide substition ($K_s$) values were obtained from CoGe[89] and compared between *C. arabica* and each of two diploid outgroups, *C. canephora* and *C. eugenioides*, using scripts in R.

### Linkage map

A reference genetic map was constructed from a cross between a Congolese group genotype (BP409) and a Congolese × Guinean hybrid parent (Q121). The segregating population was composed of 93 F1 individuals[90]. The parents were sequenced to 60× and progeny to 20× coverage using the Illumina HiSeq2000 platform at Nestlé Research. Following quality control with FastQC and trimming with Trimmomatic v.0.36 (ref. 91), the reads were mapped against the *C. canephora* reference assembly using BWA-MEM v.0.7.15 (ref. 92). The linkage mapping was conducted with Lep-MAP3 (ref. 83). The markers were clustered into paternal and maternal linkage groups by using a logarithm of the odds score of 18 in a segregation distortion aware model. The final

curation of the assembly, combining the two parental maps, solving conflicts as well as identification of haplotype alleles, was carried out manually.

### TE annotation and analysis

EDTA[93] was used to de novo identify TEs in the *C. canephora*, *C. eugenioides* as well as *C. arabica* subgenomes. Inpactor2 (ref. 94) was used to recover full-length LTR retrotransposons in the three genomes and to classify them at the lineage level. EDTA and Inpactor2 libraries were merged and clustered using cd-hit[95]. Clusters were manually inspected to remove nested and false predictions. After curation, libraries were used for annotation using RepeatMasker (default parameters). Annotations with length >200 base pairs (bp) were retained. The timing of LTR retrotransposon insertions was studied in the three genomes using individual sequences recovered by Inpactor2 and using an average base substitution rate of $1.3 × 10^{-8}$ (ref. 96), similar to Orozco-Arias et al.[97].

### Gene prediction

RNA-seq and IsoSeq reads were generated to support de novo gene prediction. A MAKER-P pipeline[98] was used to combine several de novo gene callers with the IsoSeq and junction information from short-read RNA-seq. High-evidence gene models with Annotation Edit Distance score < 0.5 were selected for the annotation. For *C. arabica* HiFi assembly, the annotations were first transferred from CC, CE and the previous CA assembly using GeMoMa v.1.9 (ref. 99), and then combined. All genes of interest linked to coffee flavor were subjected to manual inspection and gene model curation. Following the annotation, BUSCO completeness scores were assessed for the CC, CE and CA predicted transcriptomes.

### Gene expression

Three gene families, encoding terpene synthases (*TPS*), *N*-methyltransferases (*NMT*) and fatty acid desaturase 2 (*FAD2*), were further characterized and used to investigate the influence of the presence of the extra gene copies in the allopolyploid using previously published expression data[100]. The expression data presented here are the TPM (transcripts per million) normalized counts with log-scaling: $\log_{10}(x + 1 × 10^{-4})$, where $x$ is the TPM count from STARaligner[101]. For leaf rust differential expression analysis, previously published RNA-seq data[75] were reanalyzed by mapping the reads on *C. arabica* HiFi assembly using STARaligner. Differential expression in Timor hybrid versus susceptible Caturra accession after inoculation with *H. vastatrix* was analyzed with DEseq2 (ref. 102) in R. False discovery rate (FDR) adjustment was carried out using the Benjamini–Hochberg method; adjusted *P* value < 0.05 was considered statistically significant.

### Evolution of synteny and fractionation

Synteny information was obtained using the SynMap tool on the CoGe platform[88,89]. Only genes within synteny blocks were considered, not only gene pairs but also singleton genes in each genome that have lost their counterpart in the other genome due to fractionation or other gene loss.

We used the 'peaks' method[103], as calculated by the R function *geom_density*, for the three events that generate duplicate genomes during genome evolution of *C. arabica*, that is, the gamma triplication at the origin of the core eudicots, the speciation underlying the CC/CE divergence and the allotetraploidization event.

### HE

Syntenic genes between CE, CC, subCC and subEE were identified using the SynMap tool on the CoGe platform. Identification of allele biases was carried out by mapping the *C. arabica* short-read sequencing data against combined CE and CC assemblies using BWA-MEM[92] and calculating sequencing coverages on syntenic genes using bedtools. Differential coverage across the chromosomes was visualized using custom R scripts. To reduce noise, a sliding window of ten genes was

used to calculate the average coverage along chromosomes. The allele balance was calculated as $A = 4 \times ((CC/(CC + EE)) - 0.5)$, where $CC$ and $EE$ are the subCC and subEE syntelog coverages, respectively. Allele balances $<-1.5$ or $>1.5$ were considered homozygous for $EE$, or $CC$, respectively, while balances $<0.5$ and $>-0.5$ were considered equal.

## SNP calling

Following quality control with FastQC[104], Illumina short reads were trimmed using Trimmomatic v.0.36 (ref. [91]) and mapped on the *C. arabica* reference assembly with BWA-MEM v.0.7.16a-r1181 (ref. [105]). For the Linnaean sample, the reads were processed according to the protocols recommended for degraded DNA analysis in MapDamage v.2.0.8 (ref. [106]). GATK (v.3.8.0) pipeline was used for SNP calling. Duplicates were marked and removed using Picard v.2.0.1 and genotype likelihoods were called into GVCF files using HaplotypeCaller (GATK). For the diploid progenitors, to allow interspecies comparisons, the mapping was done to each of the subgenomes separately, including chromosome zero, that is, contigs not assembled into pseudomolecules, in both mappings. Joint calling was carried out using GenotypeGVCFs (GATK)[107] and snpEff v.4.3t was used to assess the impact of the SNPs[108]. To remove regions with cross-species mappings, we removed the SNPs that were called as heterozygous when mapping the di-haploid ET-39 sequencing data to the Arabica reference genome.

Genome-wide nucleotide diversity was calculated with vcftools v.0.1.17 (ref. [109]), by calculating the mean of *pi* values from sliding windows of 100 kb with 10-kb step size. Similarly, genome-wide Tajima's *D* was calculated from the mean of Tajima's *D* values with window size of 100 kb. PCA was run using Plink v.1.90 (ref. [110]). ADMIXTURE v.1.3.0 (ref. [111]) was run for SNP data where the variants in repeat regions were filtered out and the outgroup species (diploid *Coffea* species) were excluded. The SNPs were filtered for linkage disequilibrium (LD) according to the recommendation in the ADMIXTURE manual with (*--indep-pairwise* 50 10 0.1) while allowing maximum 10% missing values (*--geno* 0.1). Admixture analysis was run using tenfold cross-validation. The solution giving lowest cross-validation score was selected as the best solution. Nonsynonymous nucleotide diversity, $\pi_0$, and neutral, intergenic $\pi_s$ were calculated using the PiNSiR R package (https://github.com/jsalojar/PiNSiR) and ANGSD v.0.933 (ref. [112]), similar to ref. [58].

## Analysis of GBS data

Read data from 736 PstI GBS libraries of *C. arabica*[20] were downloaded from the SRA repository (bioproject PRJNA554647). The samples were 100-bp single-end reads sequenced on an Illumina HiSeq2000 instrument. After trimming and quality filtering, the data were mapped onto the reference genome sequence of *C. arabica* using the BWA-MEM algorithm with default settings in BWA v.0.7.17 (ref. [105]). SNPs were called using the Unified Genotyper in GATK v.3.7 (ref. [107]).

## F3 statistics

The Admixtools package[113] was used to calculate the F3 statistics, and the obtained *P* values were subjected to FDR correction using the procedure developed by Salojärvi et al.[114], where the *Z*-scores were converted into *P* values, subjected to FDR correction using Benjamini–Hochberg correction and then converted back to *Z*-scores.

## SNP trees

The SNPs were filtered for repetitive regions, followed by filtering for LD > 0.4 and loci with >40% missing values, as well as minor allele prevalence <10%. The obtained fasta file of the selected sites was input for RAxML with -T 30 -m GTRGAMMA model, using 30 starting trees and 1,000 bootstrap samples[115].

## Pairwise sequentially Markovian coalescent modeling

For each individual, the reads were mapped against the full CA reference assembly. The mappings were then filtered for indels using

bcftools and regions with <8× or >100× coverage. After filtering, the obtained pairwise sequentially Markovian coalescent (PSMC) fastq file was split into subCE and subCC specific parts and PSMC demography was estimated using standard parameter settings (-N25 -t15 -r5)[116]. The inferred history was then visualized using R and *ggplot2* package.

## Ancestral state estimation

The ancestral state was inferred from reads of two representatives of each of the diploid coffee species, *C. canephora* (BUD15, Q121) and *C. eugenioides* (BU-A, DA56), mapped against each of the subgenomes and the unassigned contigs. Subsequently, a majority vote was carried out to infer the ancestral allele using ANGSD v.0.933 (ref. [112]) with options *-doFasta* 2 and *-doCounts* 1. The SNP calls in the VCF file were then flipped to the ancestral states using bcftools +fixref[117].

## SMC++

The input data for SMC++ comprised the VCF file where the ancestral state was used as reference (see above) and the SNPs in repeat regions were filtered out. For the cultivar population, the representatives of Bourbon and Typica lineages were included (TIP1, Bourbon, Mundo Novo, BMJM, Moka, Rubi, Topazio, Bourbon pointu, Catuai99, BB1, Erecta, JK1, Guatemalense, Amsterdam); Geisha was removed from the analysis because of its unknown pedigree. SMC++ parameter selection was carried out using threefold cross-validation (smc++ cv) implemented in SMC++ v.1.15.3 (ref. [51]).

## Kinship analysis

Before kinship analysis, the diploid species were removed from the SNP file and the kinship was estimated using KING software v.2.2.5. with *--kinship* option[59]. The results were visualized using Keynote, for each subgenome separately.

## Introgression analyses

Orientagraph v.1.0 (ref. [60]) was run for each of the subgenomes separately according to the developer recommendations by carrying out filtering for linkage as recommended for TreeMix[118]. PopGenome R package was used to calculate d_f statistics[61]. For the subCE introgression, BUD15 was used as outgroup, DA56 as the source of introgression and E383 as the nonadmixed wild representative. For subCC, DA56 was used as outgroup and BUD15 as the source of introgression. The statistic was calculated in 20-kb nonoverlapping windows using weighted jackknife to assess the significance of introgression. The results were visualized using R.

## Population simulations

FastSimCoal v.2.6 was used for population simulations[54]. Site frequency spectrum was calculated using ANGSD[112] with the VCF file containing wild individuals and repetitive regions filtered out. The ancestral states were estimated as described above. For each of the models, 100 parameter files were simulated. For each parameter file, 1,000,000 simulations were run; monomorphic sites were not used. Maximum composite likelihood estimation of parameters was carried out with 40 expectation-conditional maximization iterations.

## Fixation index

Site-wise $F_{ST}$ values between wild and cultivated individuals were calculated for each gene annotation and 2-kb flanking regions using vcftools[109]. Then, mean $F_{ST}$ values were calculated for each gene model using the R package.

## TE insertion polymorphisms

We studied LTR retrotransposon insertions via analysis of short-read whole-genome resequencing data using TIP_finder[119], using the discordant mapping pair approach.

## Biosynthetic gene clusters

Biosynthetic gene clusters were identified with the Plantismash web server (http://plantismash.secondarymetabolites.org/) following default analysis protocols[120].

## Statistical testing

Statistical significance of overlaps between various gene sets was assessed using Fisher exact test in R. Gene set enrichments were carried out by first assigning each gene to the GO category of the closest Arabidopsis homolog (using $E$-value threshold $1 \times 10^{-5}$). Tests for enrichment were carried out using goatools[121]. Bonferroni-corrected $P$ value of 0.05 was used as threshold for significance. Tests for the allele balance were carried out using chi-squared test; each test had d.f. = 1.

## Reporting summary

Further information on research design is available in the Nature Portfolio Reporting Summary linked to this article.

## Data availability

Coffee genome assemblies are available at CoGe (https://genome-volution.org/): *C. canephora*: 50947; *C. eugenioides*: 67315; and *C. arabica*: 66663 (Pacbio HiFi) and 53628 (Pacbio). The genome data are also available at ORCAE (https://bioinformatics.psb.ugent.be/orcae/overview/Coara and https://bioinformatics.psb.ugent.be/gdb/coffea_arabica/). All sequencing data are available at NCBI under bioproject ID PRJNA698600, and our assemblies are accessioned there as JAZHSI000000000.1, JAZHGF000000000.1, JAZHGH000000000.1 and JAZHGG000000000.1. Genotyping data (VCF files) and syntenic alignments are available in Data Dryad: https://doi.org/10.5061/dryad.qnk98sfpt.

## Code availability

R scripts for calculating the neutral and deleterious nucletide diversities (PiNSiR) are provided in Zenodo[122] (https://zenodo.org/doi/10.5281/zenodo.5136526).

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

## Acknowledgements

We acknowledge the Natural History Museum in London for providing a sample of the *C. arabica* lectotype. Funding was provided from the Academy of Finland (decisions 318288 and 329441) and a Nanyang Technological University start-up grant (J. Salojärvi); Ecos-Nord grant no. C21MA01 and STIC-AmSud grant no. 21-STIC-13 (R.G. and S.O.-A.); the Academy of Finland, grant no. 343656 (P.R.); NAPI Bioinformática from Fundação Araucária and TELearning Project 2021–22 (grant no. 21-STIC-13) from STIC-AmSud (A.R.P.); Research Foundation – Flanders (FWO, grant no. G056517N) (Y.B.); the European Research Council (ERC) under the European Union's Horizon 2020 research and innovation program (grant no. 833522) and Ghent University (Methusalem funding, grant no. BOF.MET.2021.0005.01) (Y.V.d.P.); the Horizon Europe program, PRO-GRACE project (grant no. 101094738) (G.G.); INCT-Café-CNPq/Fapemig (A.A.); São Paulo State Research Foundation (FAPESP), grant nos. 2016/10896-0 and 2017/01455-2 (D.D. and S.M.C.L.); NSERC and the Canada Research Chairs programs (D.S.); the United States National Science Foundation grant nos. 1442190 and 2030871 (V.A.A.); and Nestlé Research (P.D.). J. Salojärvi acknowledges the High Performance Computation Centre at NTU Singapore and University of Helsinki Linux administrators, as well as the CSC – IT Center for Science, Finland, for computational resources. R.G., P.H., E.C., C.T.D., V.P., A.d.K. and Unité Mixte de Recherche - Diversité, adaptation, développement des plantes (UMR DIADE) are grateful to The French National Research Institute for Sustainable Development (IRD).

## Author contributions

A.d.K., D.C. and P.D. conceived the study. A.A., A.N., C.K., E.C., G.H.S., H.R., L.B., L.F.P., L.P., M.S., M.T.B., O.G.F., P. Musoli, P. Marraccini, P.H. and U.S. provided genetic resources. A.C., C.F., D.M., G.L., J. Stolte, L.B., M.K., N.D., P.D. and S.M. carried out DNA sequencing. E.G. performed sequencing of the Linnaean accession. S.S., C.W., J. Salojärvi, S.P. and L.M. carried out genome assembly. P.R., M.R. and J. Salojärvi performed genetic mapping. A.R., S.S., L.M., J. Salojärvi, S. Rombauts, V.P., Z.Q.W., D.D., S.I.S., M.M., R.A., S.M.C.L., M.L., C.T.-D. and G.G. carried out genome annotation. A.R.P., J.E. and P.S. carried out annotation of noncoding RNA. S.O.-A., A.G. and R.G. performed transposable element annotation and analysis. V.A.A. and C.M.W. carried out telomere identification. Z.Y., C.Z., D.S.M., R.G., J.M., D.S., L.C.-P., T.L., T.J.K., V.A.A., S.O.-A., A.G. and J. Salojärvi analyzed genome evolution. Z.Q.W., V.P., D.D., G.G., S.J.F., V.A.A., S. Rajaraman and J. Salojärvi carried out gene family analysis. A.R., S.P., S. Rajaraman and J. Salojärvi performed RNA-seq data analysis. R.H. provided RNA-seq data. J. Salojärvi analyzed population data. Y.B. and R.G. analyzed GBS data. L.M., S. Rombauts and J. Salojärvi arranged online data access. J. Salojärvi wrote the first draft, which was completed with input from G.G., D.S., V.A.A., L.F.P., R.G., S. Rombauts, A.d.K., P.D., V.P., L.M., D.C., D.D., S.P. and A.A., as well as P. Marraccini, Y.B., T.R. and Y.V.d.P., and all co-authors.

## Competing interests

The authors declare no competing interests.

## Additional information

**Extended data** is available for this paper at https://doi.org/10.1038/s41588-024-01695-w.

**Correspondence and requests for materials** should be addressed to Jarkko Salojärvi, Victor A. Albert, Dominique Crouzillat, Alexandre de Kochko or Patrick Descombes.

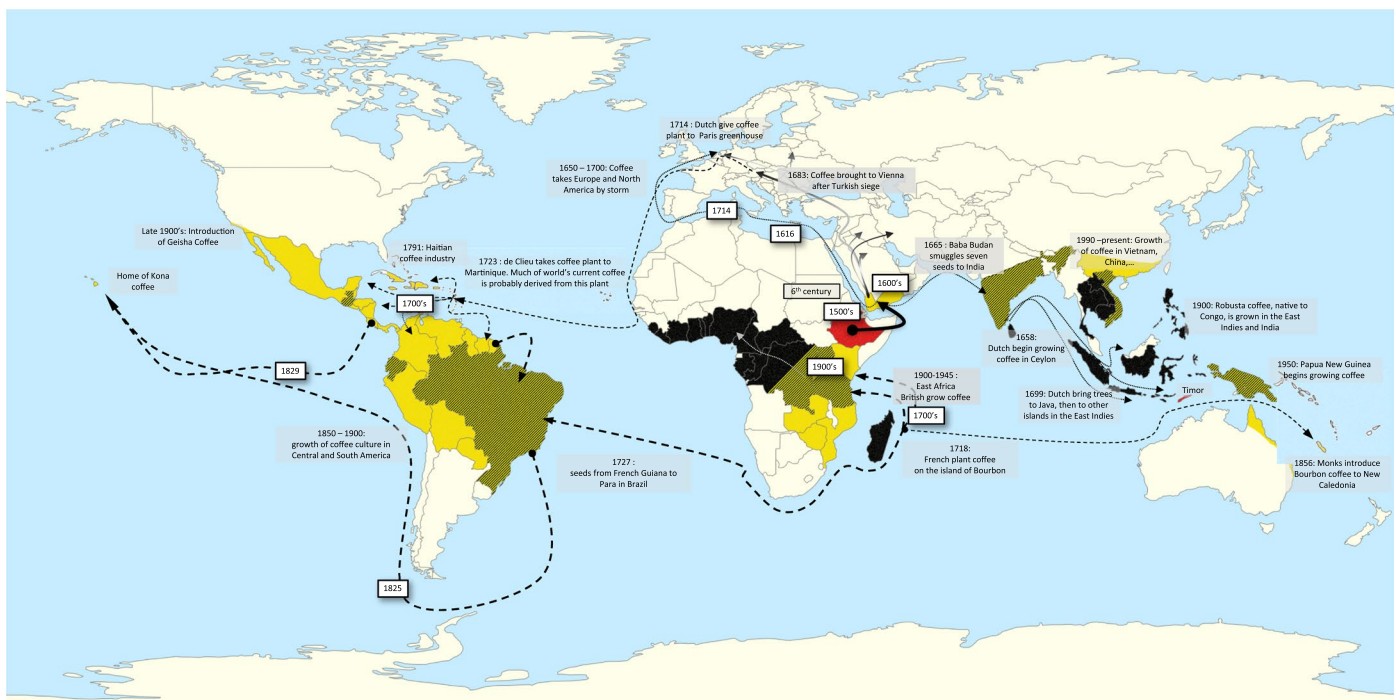

**Extended Data Fig. 1 | Coffee dissemination routes.** Yellow: current *Coffea arabica* cultivation; black: current *C. canephora* cultivation; shaded black/yellow: current cultivation, both species. Solid lines over the Middle East: early spread of coffee consumption; dashed lines: main Bourbon routes; dotted lines: main Typica routes; Ethiopia (the center of origin of Arabica) and Timor Island (the origin of the Timor hybrid) are colored in red. The map was modified from an original available in the public domain from Wikimedia commons.

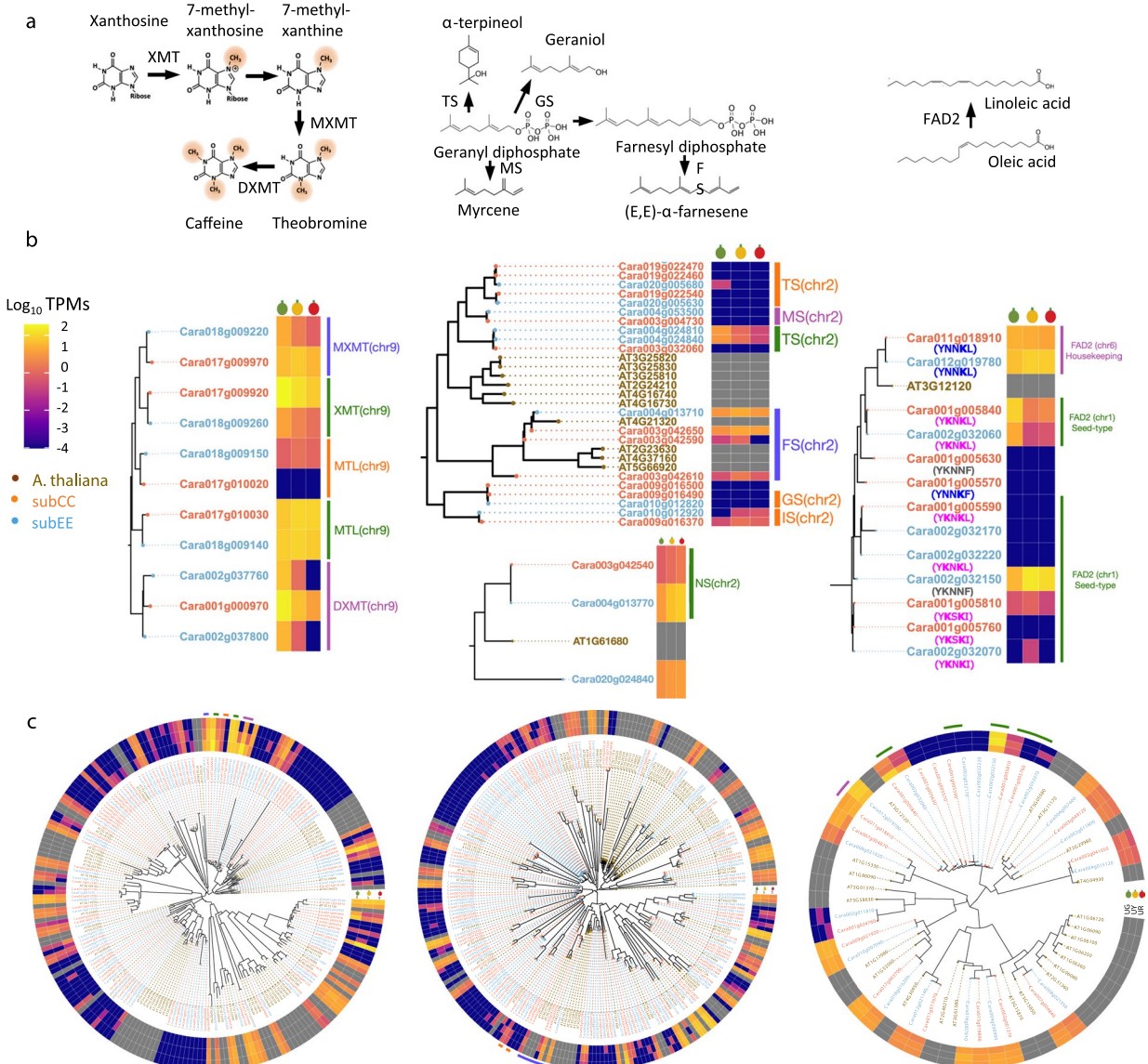

**Extended Data Fig. 2 | Composition and expression of exemplar Arabica gene families contributing to bean quality traits. a**, Schematic biosynthesis of caffeine (left), terpenoids (middle), and unsaturated fatty acids (right). **b**, Phylogenies and expression during fruit development of CA genes for N-methyltransferases (NMTs) mediating caffeine biosynthesis (left), terpene synthases (TPS) (middle), and fatty acid dehydrogenase 2 (FAD2) (right). RNA sequencing was carried out for three biological replicates from three different fruit maturation stages (green, yellow, and red) of the K7 cultivar. **c**, Genome-wide NMT (left), TPS (middle), and FAD2 (right) gene trees and expression patterns during fruit development. Genes located in the two subgenomes are indicated by font color; subCC (red) and subEE (blue). Arabidopsis genes are in brown. Grey wedges in the circular trees highlight the tree portions shown in **b**. XMT: xanthosine methyltransferase; MXMT: 7-methylxanthine methyltransferase; DXMT: 1,7-dimethylxanthine methyltransferase; MTL: N-methyltransferase-like; FS: (E,E)-a-farnesene synthase; GS: Geraniol synthase; IS: Isoprene synthase; MS: myrcene synthase; TS: (-)-a-terpineol synthase; FAD2: Fatty acid desaturase 2. Gene expression is shown by color scale, yellow (positive) through red to blue (negative), in units of log10 transcripts per million (log10 TPM); grey areas indicate the absence of expression data for Arabidopsis genes.

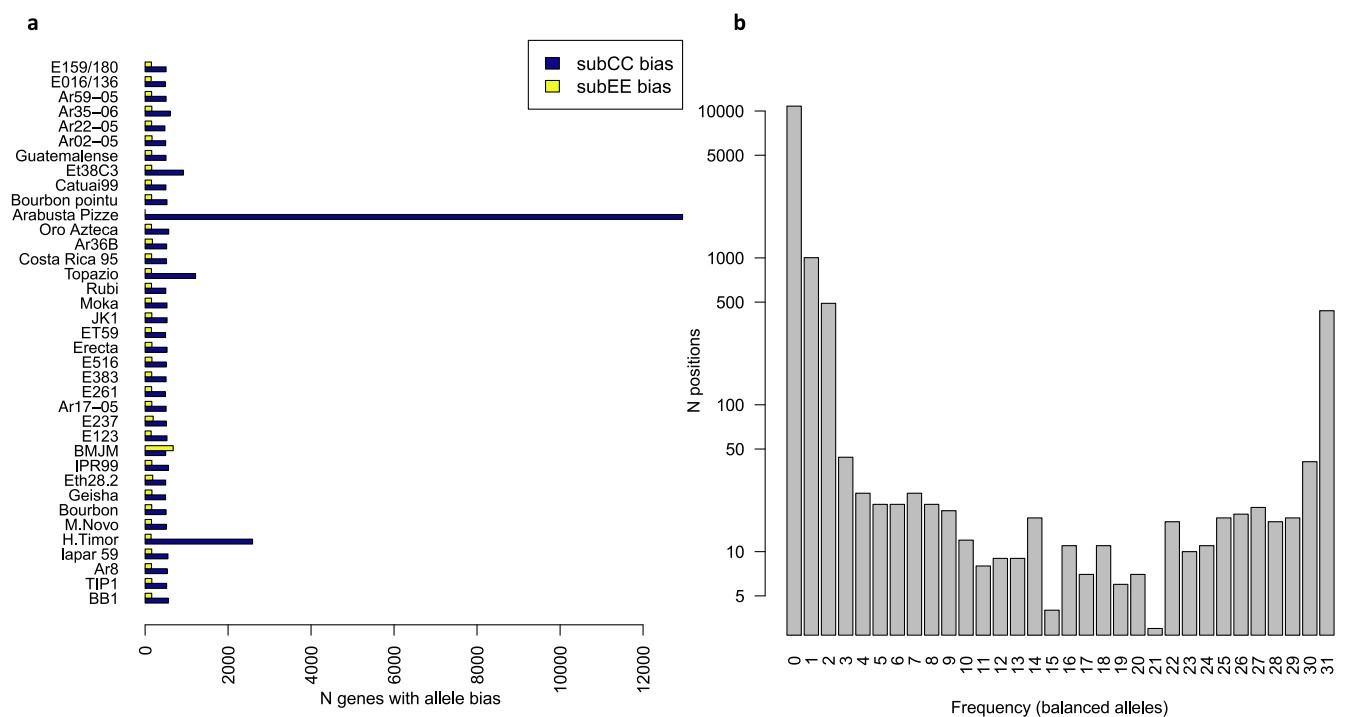

**Extended Data Fig. 3 | Homoeologous exchange. a**, Summary of homoeologous exchange between subgenomes. Blue bars indicate genes with 3:1 allele bias towards subCC, whereas yellow bars indicate genes with allele bias (1:3 or 0:4) towards subEE. **b**, Frequency spectrum of shared homoeologous exchanges at gene level.

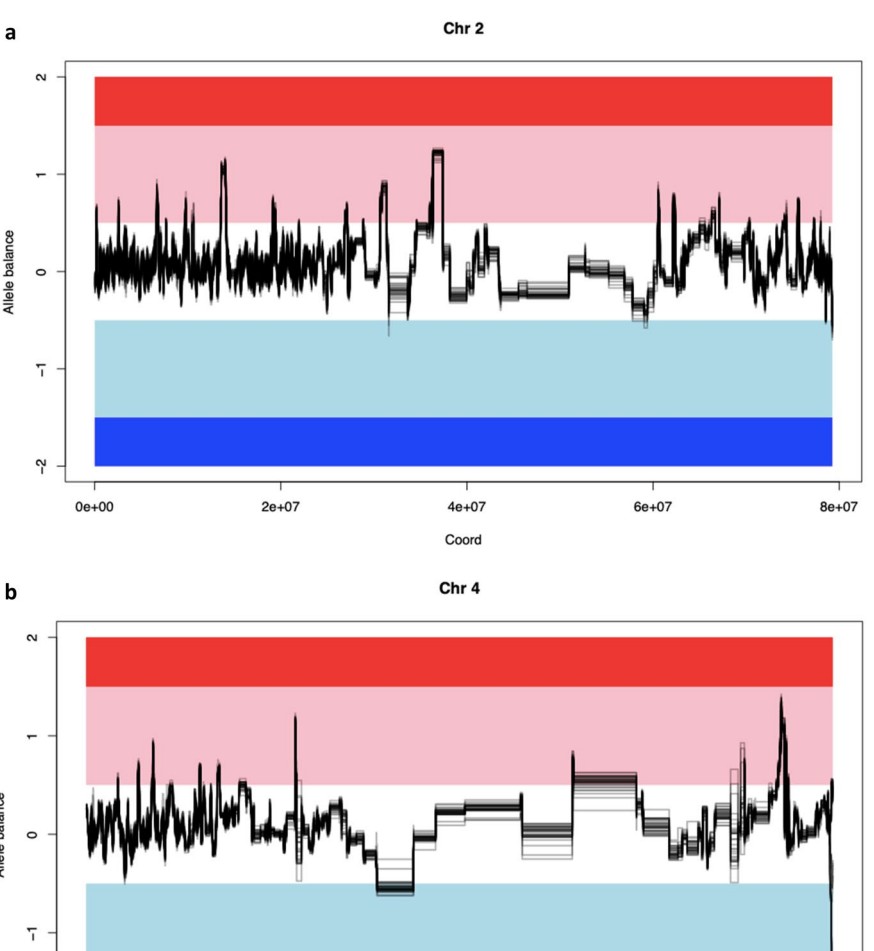

**Extended Data Fig. 4 | Homoeologous exchange plots of chromosomes 2 (a) and 4 (b) overlaying all *Coffea arabica* accessions in this study.** The dark red region indicates 4:0 allele balance in favor of subCC, while the pink region illustrates 3:1, white 2:2, light blue 1:3 and dark blue 0:4 balances, respectively. The grey lines indicate the observed allele balances in syntelog gene pairs for the different Arabica accessions. For a view of all chromosomes and of the genes involved, see Supplementary Figure 35 and Supplementary Table 27.

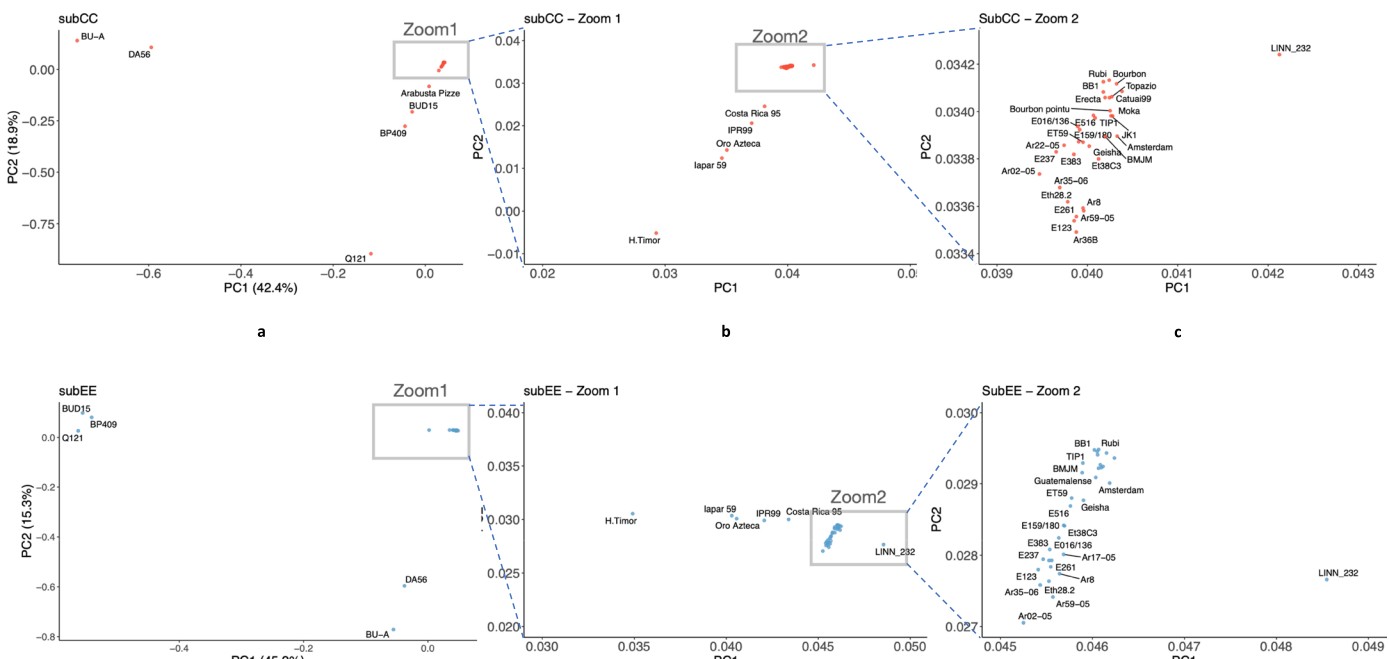

**Extended Data Fig. 5 | PCA plots based on SNPs. a**, From SNP data called on the subCC (top) versus subEE (bottom) subgenomes. The rectangles highlight zoomed-in regions in panels **b** and **c**.

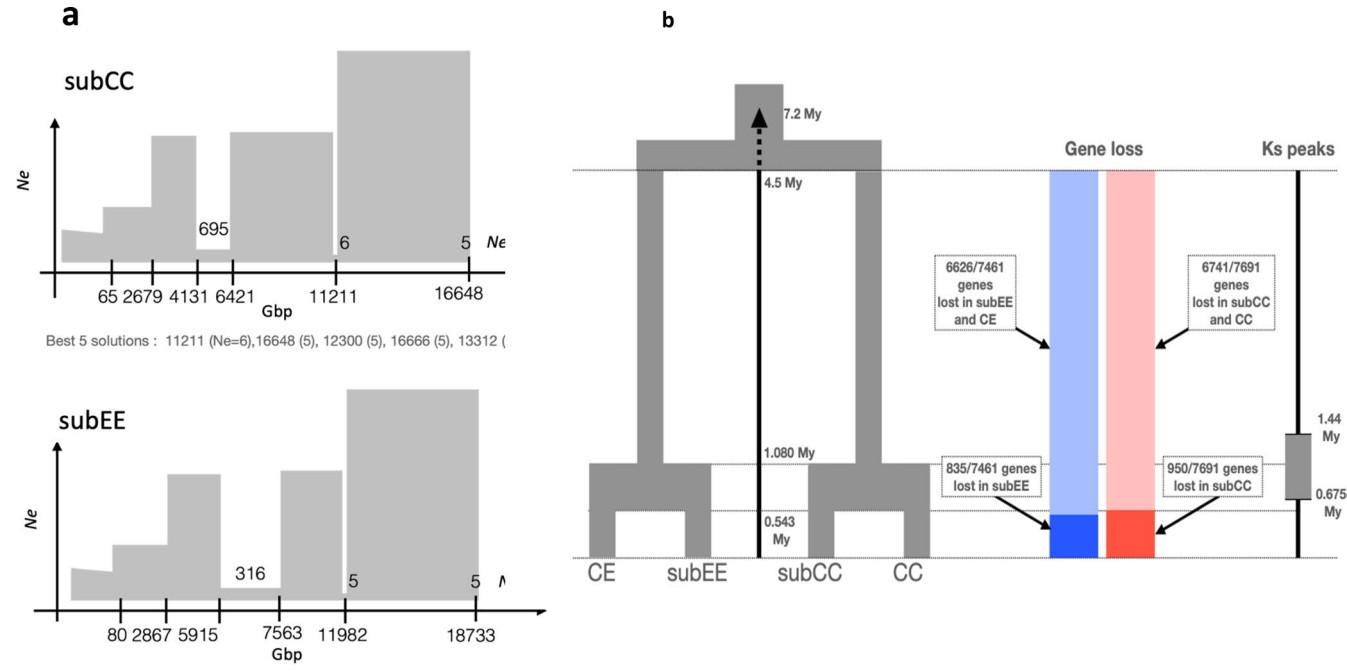

**Extended Data Fig. 6 | Arabica speciation time estimates based on population genetic modeling and rate of genome fractionation. a**, Summary of FastsimCoal2 models for historical effective population sizes (N$_e$). The effective population size (y-axis) is plotted against the number of generations before present (Gbp, x-axis). The bottlenecks were identified using 100 FastSimCoal runs with 10^6 simulations. Maximum composite likelihood estimation of parameters was carried out with 40 expectation-conditional maximization iterations. The plots summarize the best models for subgenomes CC and EE in the wild, non-admixed population. To convert generations to years, an estimate of 21 years/generation was used (Moat el al. 2019). **b**, Summary of the genome fractionation rate and divergence of syntenic gene models. The timing of the splits in the phylogeny (left) reflects the most recent estimates from (Bawin et al., 2020). The rate of gene loss (barplot) is presented as the percent of syntenic genes lost in the Eugenioides/subEE common ancestor (light blue) or only in subEE (blue). A similar analysis was carried out for Robusta-derived genomes, where the percent of genes lost in Robusta/subCC is shown in light red and genes lost only in subCC with dark red. The K$_s$ peaks method (right) scales the divergence time between the subgenomes, estimated from numbers of synonymous mutations between syntenic genes to the timing of the speciation event.

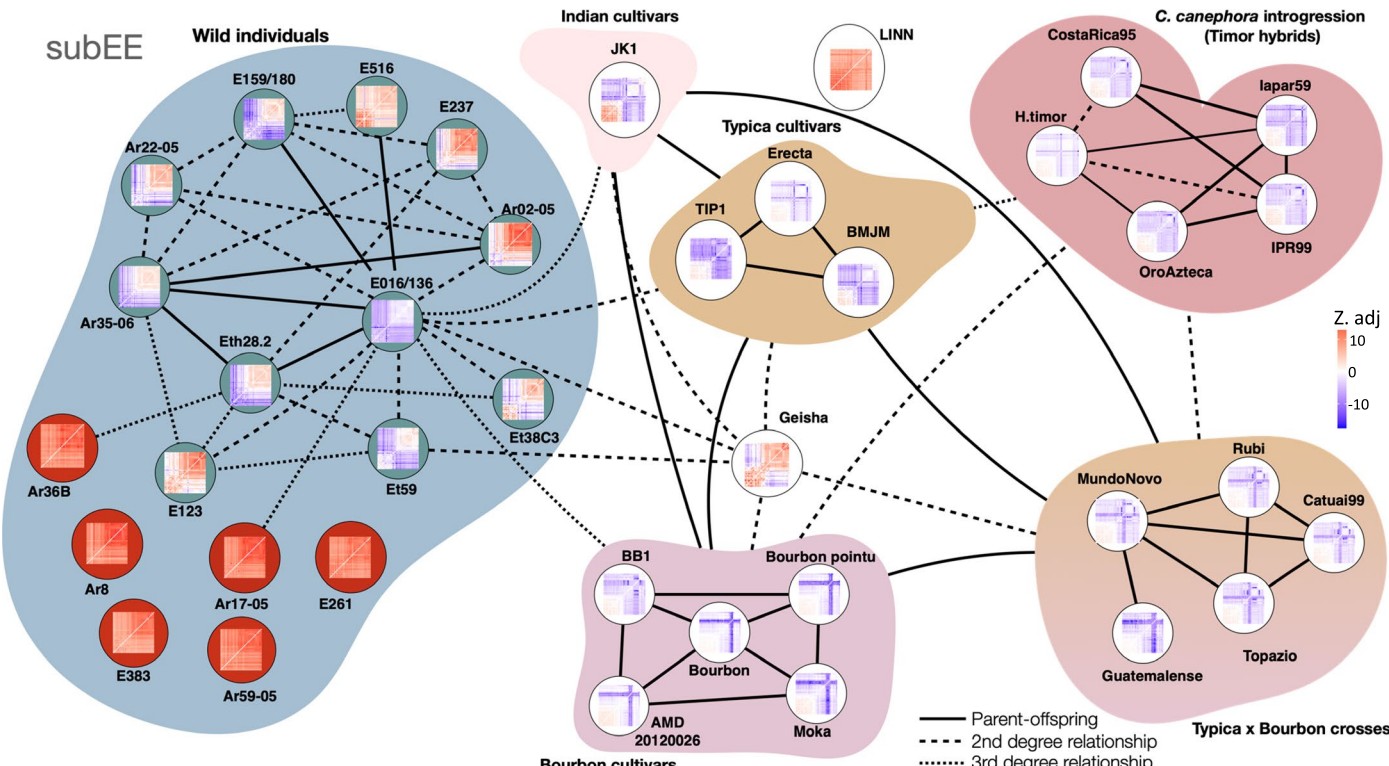

**Extended Data Fig. 7 | Kinship analysis on subEE.** The degree of relatedness was estimated using Kinship-based INference for GWAS (KING); Thumbnail images show false discovery rate corrected F3 tests of introgression Z-statistics for each of the target individuals. Each cell in the matrix illustrates an F3 test result for the target accession containing introgression from two different sources (x- and y-axis); blue color illustrates significant adjusted Z-score (Z. adj; associated value indicated by color key), indicative of gene flow (or allele sharing via identity by descent; IBD) from the two source accessions to the target, while red color illustrates no support for gene flow. The green background in the wild accessions highlights the admixed individuals (Fig. 2b); the non-admixed individuals are highlighted with red. The corresponding analysis on subCC is shown in Fig. 3.

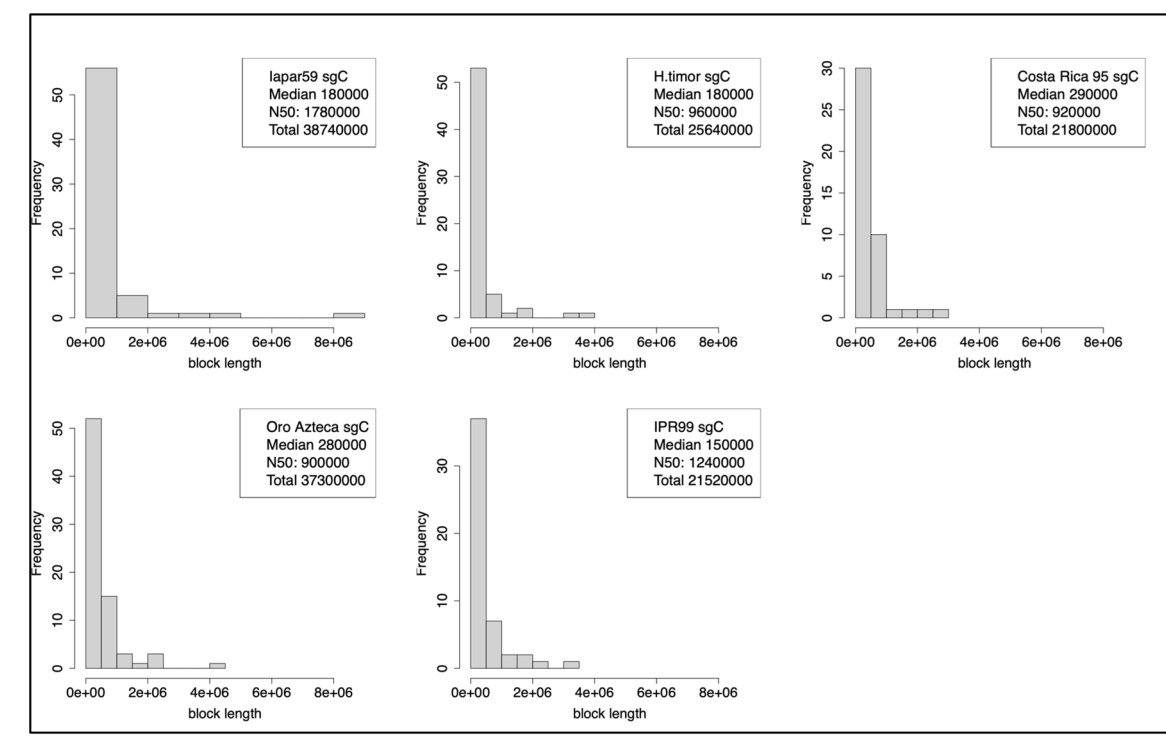

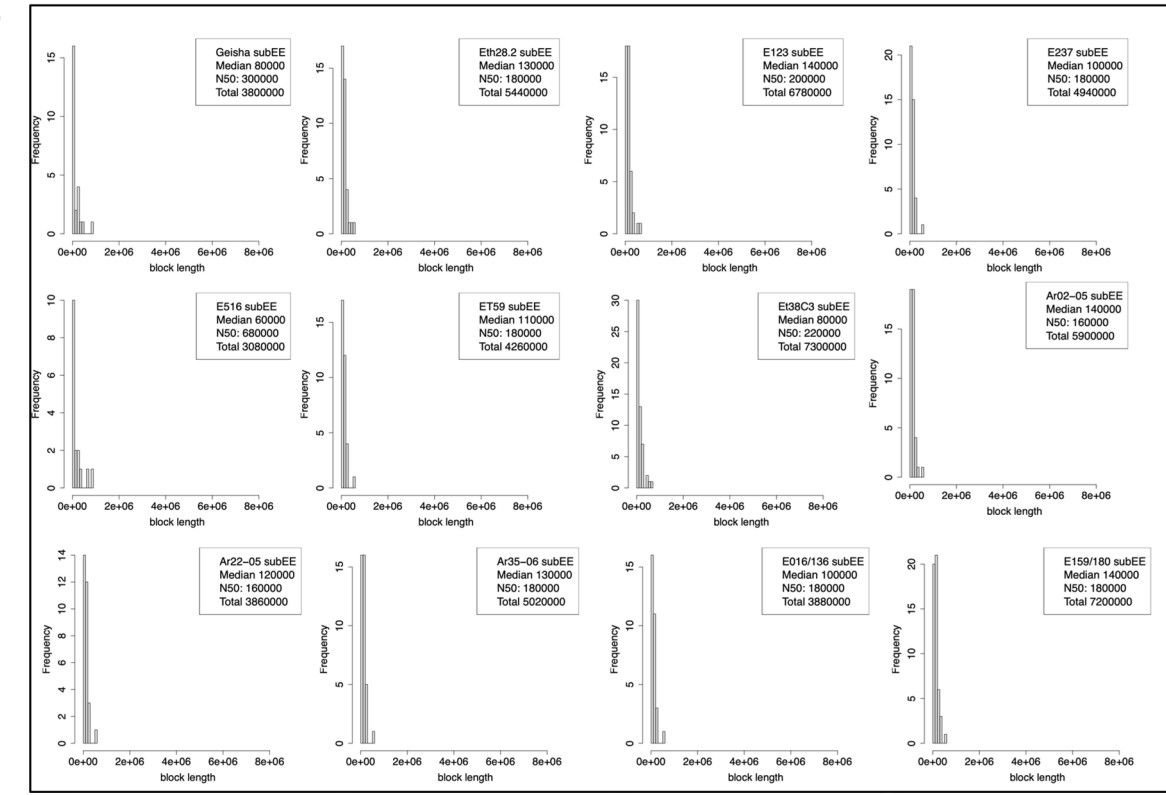

**Extended Data Fig. 8 | Timing of wild Arabica introgression.** Lengths of Robusta introgressed blocks in Timor hybrid accessions (**a**) and, as a control, of Typica introgressed blocks in wild Arabica accessions (**b**).

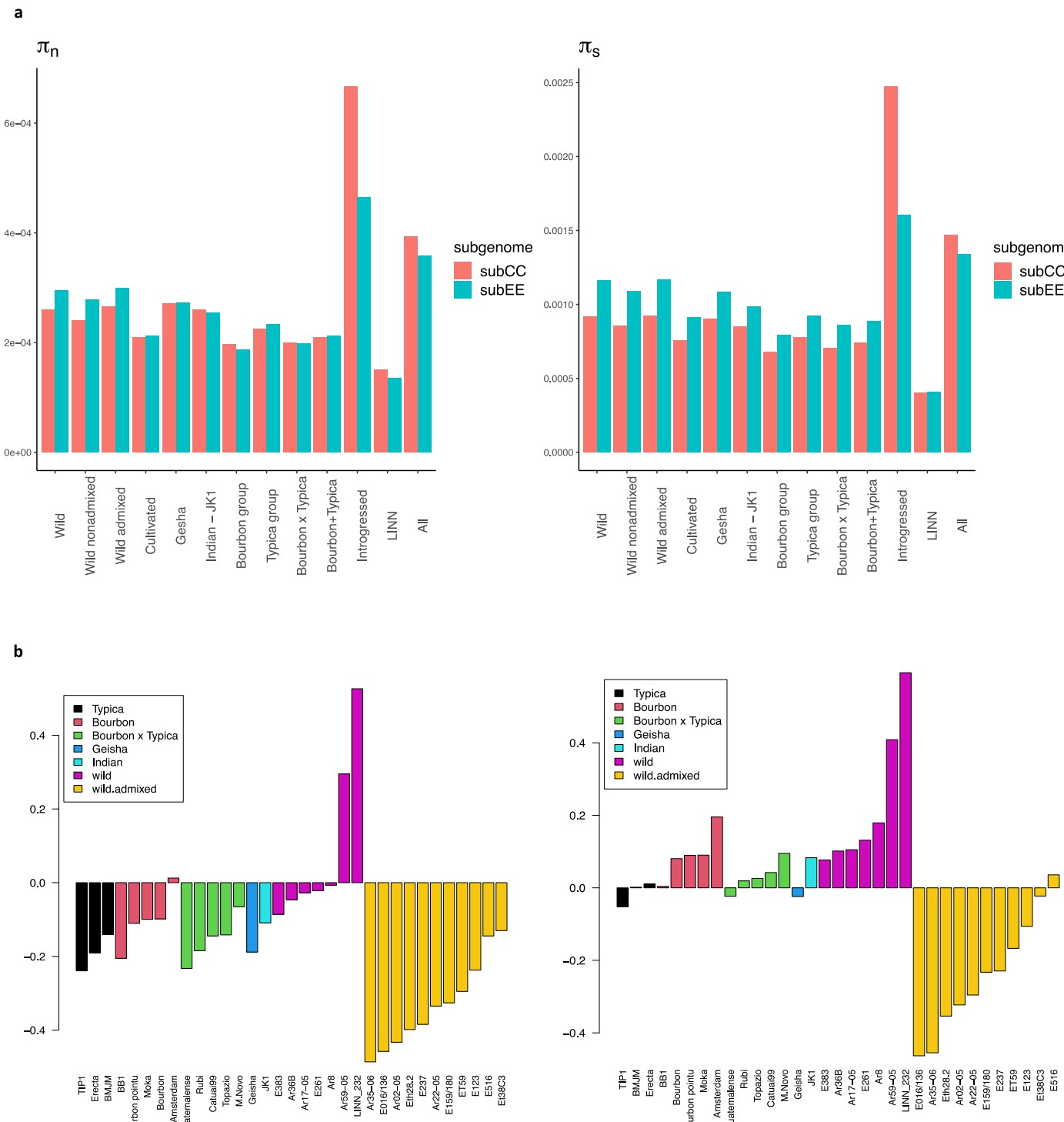

**Extended Data Fig. 9 | Population genetic statistics. a**, Non-synonymous ($\pi_n$; left) and synonymous ($\pi_s$; right) diversity in the different populations of *C. arabica*. **b**, F inbreeding coefficients for wild and cultivated lines, shown separately for subCC (left) and subEE (right).

Last updated by author( 02/24/24

# Reporting Summary

## Statistics

For all statistical analyses, confirm that the following items are present in the figure legend, table legend, main text, or Methods section.

| n/a | Confirmed | |
|---|---|---|
| ☒ | ☐ | The exact sample size ($n$) for each experimental group/condition, given as a discrete number and unit of measurement |
| ☐ | ☒ | A statement on whether measurements were taken from distinct samples or whether the same sample was measured repeatedly |
| ☐ | ☒ | The statistical test(s) used AND whether they are one- or two-sided<br>*Only common tests should be described solely by name; describe more complex techniques in the Methods section.* |
| ☒ | ☐ | A description of all covariates tested |
| ☐ | ☒ | A description of any assumptions or corrections, such as tests of normality and adjustment for multiple comparisons |
| ☐ | ☒ | A full description of the statistical parameters including central tendency (e.g. means) or other basic estimates (e.g. regression coefficient) AND variation (e.g. standard deviation) or associated estimates of uncertainty (e.g. confidence intervals) |
| ☐ | ☒ | For null hypothesis testing, the test statistic (e.g. $F$, $t$, $r$) with confidence intervals, effect sizes, degrees of freedom and $P$ value noted<br>*Give P values as exact values whenever suitable.* |
| ☒ | ☐ | For Bayesian analysis, information on the choice of priors and Markov chain Monte Carlo settings |
| ☒ | ☐ | For hierarchical and complex designs, identification of the appropriate level for tests and full reporting of outcomes |
| ☒ | ☐ | Estimates of effect sizes (e.g. Cohen's $d$, Pearson's $r$), indicating how they were calculated |

*Our web collection on statistics for biologists contains articles on many of the points above.*

## Software and code

Policy information about availability of computer code

| | |
|---|---|
| Data collection | R scripts for calculating the neutral and deleterious nucletide diversities (PiNSiR) are provided in https://zenodo.org/doi/10.5281/zenodo.5136526 |
| Data analysis | Software for genome assembly: fastQC, MHAP, Falcon, Pilon, Dovetail HiRise, SNAP, PBJelly, Trimmomatic v0.36, Lep-MAP3, samtools v1.10, Irys, HiFiasm v0.16.1. Transposable elements: REPET, fast-BLAST, MITE-Hunter, MegaBLAST,Inpactor,Sine_Finder,Repeat Masker,LTR_STRUC, Inpactor2,DensityMap,tRNAscan-SE 2.0,RNAmmer,snoStrip, INFERNAL v1.1.2,cmsearch. RNAseq: AdapterRemoval,HISAT2 v2.2.0, StringTie v2.1.2, gffread v0.12.1, RNAplonc v1.1, BEDTools v2.26.0, Quality control and annotation: quast, BUSCO, webApollo, Portcullis, Mikado, Augustus, Genmark ,SNAP, Maker. Data analysis: CoGe SynMap, R, BWA mem v0.7.16a-r1181, picard v2.18.14, GATK v3.8.0, BWA samse, MapDamage v.2.0.8, snpEff v4.3t, Cutadapt v2.10, VCFtools v.0.1.17, ANGSD v.0.933, Plink v1.90, ADMIXTURE, RAxML, PSMC, bcftools, SMC++, KING v2.2.5, Admixtools, Orientagraph v1.0, Fastsimcoal v. 2.6.0.3, TIP_finder, Adegenet v. 2.1.3. |

For manuscripts utilizing custom algorithms or software that are central to the research but not yet described in published literature, software must be made available to editors and reviewers. We strongly encourage code deposition in a community repository (e.g. GitHub). See the Nature Portfolio guidelines for submitting code & software for further information.

## Data

Policy information about availability of data

All manuscripts must include a data availability statement. This statement should provide the following information, where applicable:

- Accession codes, unique identifiers, or web links for publicly available datasets
- A description of any restrictions on data availability
- For clinical datasets or third party data, please ensure that the statement adheres to our policy

Coffee genome assemblies are available at CoGe (https://genomevolution.org/): C. canephora: 50947, C. eugenioides: 67315, and C. arabica: 66663 (Pacbio HiFi) and 53628 (Pacbio). The genome data is also available at ORCAE (https://bioinformatics.psb.ugent.be/orcae/overview/Coara and https://bioinformatics.psb.ugent.be/gdb/coffea_arabica/). All sequencing data are available at NCBI under bioproject ID PRJNA698600, and our assemblies are accessioned there as JAZHSI000000000.1, JAZHGF000000000.1, JAZHGH000000000.1, and JAZHGG000000000.1. Genotyping data (VCF files) and syntenic alignments are available in Data Dryad: https://doi.org/10.5061/dryad.qnk98sfpt .

# Field-specific reporting

Please select the one below that is the best fit for your research. If you are not sure, read the appropriate sections before making your selection.

☐ Life sciences          ☐ Behavioural & social sciences          ☒ Ecological, evolutionary & environmental sciences

For a reference copy of the document with all sections, see nature.com/documents/nr-reporting-summary-flat.pdf

# Ecological, evolutionary & environmental sciences study design

All studies must disclose on these points even when the disclosure is negative.

| | |
|---|---|
| Study description | We carried out reference genome sequencing, assemblies and genome annotations of Coffea arabica ET-39 di-haploid, C. canephora DH 200-94, and C. eugenioides Bu-A accessions. This resubmission includes a new and high-quality PacBio Hifi-based assembly. Assembly was followed by a study of the genome evolution of the tetraploid C. arabica, analysis of expression dominance in specific biochemically important pathways. Next we studied the population history of C. arabica using 39 whole-genome sequenced accessions including wild and cultivated representatives. Finally, we analysed individuals containing recent introgression from C. canephora. |
| Research sample | A population of wild C. arabica representatives collected from different locations around Ethiopia during the 1960's, representing a large proportion of the geographic range of extant wild populations, was employed. Representatives of the two most commercially important C. arabica cultivar lines, Typica and Bourbon, and their crosses, were also examined. Additionally, the relatively new cultivar Geisha, which has recently become commercially important, and the lectotype individual of C. arabica from the Linnaean Society, dating back to 1700s, were sampled. Five lines were used that descended from a spontaneous C. canephora x C. arabica hybrid identified in Timor. |
| Sampling strategy | Leaves were collected from the selected individuals and sequenced to high coverage using Illumina short-read sequencing. The samples were chosen based on accessibility to material. For the cultivar lines Bourbon and Typica, the sample size was sufficient since both lines date back to a single plant bottleneck in the 1700s. Wild representatives were collected during two missions to Ethiopia in the 1960s, and they represent the diversity of wild C. arabica plants; this was verified by comparing the sequenced individuals to a wider collection where marker-based analysis had already been carried out and published. |
| Data collection | Wild representatives were collected during two missions to Ethiopia in 1960s by FAO and IRD. Cultivars were obtained from plant breeding experts in IRD, IAPAR, EMBRAPA, Nestle, ICCRI, IAC - Campinas, and NARO, as well as from conservation Institutes Amsterdam Botanical Garden and Natural History Museum London. |
| Timing and spatial scale | Wild representatives were collected in the 1960's in Ethiopia. |
| Data exclusions | No exclusion of data. |
| Reproducibility | No biological experiments were carried out in the work. All statistical analyses report a p-value associated with the analyses aimed to assess the reproducibility of the results. |
| Randomization | No clinical experimentation was done in the paper, therefore there was no need for randomisation. |
| Blinding | Blinding was not possible since the data analysis did not contain case vs. control experimental setups. |

Did the study involve field work?    ☐ Yes    ☒ No

# Reporting for specific materials, systems and methods

We require information from authors about some types of materials, experimental systems and methods used in many studies. Here, indicate whether each material, system or method listed is relevant to your study. If you are not sure if a list item applies to your research, read the appropriate section before selecting a response.

## Materials & experimental systems

| n/a | Involved in the study |
|---|---|
| ☒ | ☐ Antibodies |
| ☒ | ☐ Eukaryotic cell lines |
| ☒ | ☐ Palaeontology and archaeology |
| ☐ | ☒ Animals and other organisms |
| ☒ | ☐ Human research participants |
| ☒ | ☐ Clinical data |
| ☒ | ☐ Dual use research of concern |

## Methods

| n/a | Involved in the study |
|---|---|
| ☒ | ☐ ChIP-seq |
| ☒ | ☐ Flow cytometry |
| ☒ | ☐ MRI-based neuroimaging |

## Animals and other organisms

Policy information about studies involving animals; ARRIVE guidelines recommended for reporting animal research

**Laboratory animals**

*For laboratory animals, report species, strain, sex and age OR state that the study did not involve laboratory animals.*

**Wild animals**

*Provide details on animals observed in or captured in the field; report species, sex and age where possible. Describe how animals were caught and transported and what happened to captive animals after the study (if killed, explain why and describe method; if released, say where and when) OR state that the study did not involve wild animals.*

**Field-collected samples**

Wild representative samples were collected from Ethiopia in 1960s during two missions, and since then have been maintained in the field in different locations in Ecuador, Reunion (France), Brasil and Ethiopia.

**Ethics oversight**

*Identify the organization(s) that approved or provided guidance on the study protocol, OR state that no ethical approval or guidance was required and explain why not.*

Note that full information on the approval of the study protocol must also be provided in the manuscript.

