## [Peer Review File · Nature Genetics]

Peer Review Information

Manuscript Title: The genome and population genomics of allopolyploid *Coffea arabica* reveal the diversification history of modern coffee cultivars

Corresponding author name(s): Professor Victor (A) Albert, Patrick Descombes, Dr Alexandre de Kochko, Dominique Cruzillat, Professor Jarkko Salojärvi

Reviewer Comments & Decisions:

Decision Letter, initial version:

23rd Jun 2022

Dear Professor Albert,

Your Article, "The genome and population genomics of allopolyploid *Coffea arabica* reveal the diversification history of modern coffee cultivars" has now been seen by 3 referees. You will see from their comments copied below that while they find your work of considerable potential interest, they have raised quite substantial concerns that must be addressed. In light of these comments, we cannot accept the manuscript for publication, but would be very interested in considering a substantially revised version that fully addresses these serious concerns.

We hope you will find the referees' comments useful as you decide how to proceed. If you wish to submit a substantially revised manuscript, please bear in mind that we will be reluctant to approach the referees again in the absence of major revisions.

To guide the scope of the revisions, the editors discuss the referee reports in detail within the team, with a view to identifying key priorities that should be addressed in revision. In this case, we think all three referees have provided constructive reviews aimed at strengthening the analyses and improving the presentation. Reviewer #1 notes that the sequence of allotetraploid coffee is relatively incomplete; and there are conceptual confusion and misunderstanding with respect to polyploidy and subgenome evolution. Both reviewers #1 and #2 have serious concerns regarding the degree of biological insights provided. In addition, the quality of the figures needs to be substantially improved. We ask that you address all referee comments as thoroughly as possible with appropriate revisions. Please do not hesitate to get in touch if you would like to discuss these issues further.

If you choose to revise your manuscript taking into account all reviewer and editor comments, please highlight all changes in the manuscript text file. At this stage we will need you to upload a copy of the manuscript in MS Word .docx or similar editable format.

*2) If you have not done so already please begin to revise your manuscript so that it conforms to our Article format instructions, available here. Refer also to any guidelines provided in this letter.

Please be aware of our guidelines on digital image standards.

[redacted]

If you wish to submit a suitably revised manuscript we would hope to receive it within 6 months. If you cannot send it within this time, please let us know. We will be happy to consider your revision so long as nothing similar has been accepted for publication at Nature Genetics or published elsewhere. Should your manuscript be substantially delayed without notifying us in advance and your article is eventually published, the received date would be that of the revised, not the original, version.

Thank you for the opportunity to review your work.

Sincerely,
Wei

Wei Li, PhD
Senior Editor
Nature Genetics
New York, NY 10004, USA
www.nature.com/ng

Reviewers' Comments:

Reviewer #1:

Remarks to the Author:

In this report, the authors have generated high-quality genome sequences of cultivated coffee, an allotetraploid that is maintained as di-haploid, and its extant diploid progenitors. The genomic resources will be valuable for the coffee research community, although a similar and prior resource (Scalabrin et al. 2020 Sci. Rep.) has a moderate utility (36/46 ISI/Google citations). However, the report suffers from incomplete sequence assembly, poor quality of most (and probably all) figure panels, confirmatory nature of results, and lack of expertise and skills to organize large datasets and write a succinct genomics paper. The main text consisting of long lists of genomic statistics and features without substantiation of their findings has diminished an appeal to a broader readership in polyploid genetics and genome evolution. Specific comments are as follows.

(1) Although its genome sequence is of high-quality by integrating PacBio and Illumina sequences and SNPs derived from a F1 mapping population (91 plants), the assembly is not as complete as one would have expected. For example, the genome size of 1,088 Mb in the allotetraploid (Coffee arabica, CA) is 30% less than the sum of two diploids, 672 Mb in *C. canephora* (CC) and 645 Mb in *C. eugenioides* (CE). No explanation was offered. Is this possibly related to the di-haploid status (that was first reported by Mendes and Bacchi (1940, J. Agron.)? At the pseudo-chromosome level, they represent only 83% (CC) and 68% (CA, a mistake, could be CE?) of the projected genome size. These and other statistics in the report may indicate relatively incomplete and poor quality of sequence assemblies in both tetraploid and diploid species.

A related comment is on the nomenclature of subgenomes in allotetraploids. By convention, each diploid would be given a genome designation, perhaps C for *C. canephora* (CC, $2n = 2x = 22$) and E for *C. eugenioides* (EE, $2n = 2x = 22$), and the resulting allotetraploid would have the genome designation CCEE ($2n = 4x = 44$). Note that C or E is used as an example and could be replaced by the nomenclature based on cytogenetics and breeding and used in the research community. Using the species names (CC, CE, and CA) to describe subgenome is confusing and cumbersome for polyploid genetics and breeding. As examples, genome designation in wheat is AABB for tetraploid (spaghetti/durum wheat) and AABBDD for hexaploid (bread wheat).

(2) Genome evolution and fractionation in allotetraploid coffee (Fig. 2). The data presented in this

section, including figure panels and description, did not capture much useful information, compared to a previous study in coffee (Scalabrin et al. 2020 Sci. Rep.). The subgenomic stability and conservation has been well documented in wheat, cotton, and Arabidopsis allopolyploids. Discussion was poorly constructed with inaccurate citations. For example, maize was cited to be polyploid species with a hybrid origin (lines 299-301) and again cited as one of recently formed allopolyploids (line 302-305). Whole genome duplication or polyploidy occurred in maize 5-12 million years ago (Schnable et al. 2011, PNAS), and maize is considered to be a paleopolyploid with unknown origin of auto- or allopolyploid (Zhao et al. 2017, Plant Cell). These erroneous statements could stem from conceptual misunderstanding of auto- and allopolyploidy, as well as subgenomic evolution. Most figure panels are not useful and have to be reconstructed to make them informative.

Genome shock has two meanings at genomic and expression levels. Wheat, cotton, and Arabidopsis allopolyploids show rapid gene expression changes (shock), while Brassica neo-polyploids show rapid genomic reshuffling (Xiong et al. 2011, PNAS). In addition, conservation in old (610,000 years-1 mya) coffee allotetraploids does not mean no genome shock at the time of polyploid formation.

(3) Genome-wide subgenome expression dominance (Fig. 3). This section is misleading, as it was narrowly focused on only three gene families (NMT, TPS, and FAD). FAD gene family may be of relevance to coffee physiology (flavour, aroma, etc.). A suggestion is to provide a brief view of genome-wide data, followed by FAD gene family. Again, the quality of figure panels is poor and should be redone, e.g., labels were too small to be legible, in addition to the ambiguous patterns.

(4) Origin and domestication (Fig. 4). This section can be substantially reduced to avoid redundant information from the previous paper (e.g., Scalabrin et al. 2020 Sci. Rep.) and retain part of population statistics to show the relevance of domestication. Again, there is a problem of figure quality as noted above (illegible fonts and patterns). For example, what is the X-axis? What are populations 1, 2, 3?

(5) Population history (Fig. 5). This part of data is interesting but lacks clarity of polyploidization (estimated ~610 kya) and domestication (8.9 kya) after population split (30.5 kya). It is unclear how they determine these events and what are the representative species or accessions, and if these estimates are supported by breeding and systematic information. The problem of admixture would have complicated their analysis and data interpretation (e.g., x generations within and between different stages of divergence).

(6) Kingship estimation and origin of cultivated species (Fig. 6). This part of information is rather confirmatory with the previous finding (Scalabrin et al. 2020 Sci. Rep.). Again, the figure panels and labels are illegible and confusing.

(7) Relationships between wild and cultivated Arabicas (Fig. 7). This part of data can be presented in combination/comparison with previous sections with comments in (5) and (6). Many data figures in these sections can be moved into supplemental information.

(8) Time of admixture and effect on domestication. These could be succinctly discussed along with origin, evolution, and domestication in previous sections. Alternatively, the previous sections could have been substantially reduced to make a few succinct statements.

(9) Mutational load during domestication (Fig. 8). This part of data adds little value and should be

moved to the supplemental information.

(10) Introgression (Fig. 9). The relationships between introgression/heterozygosity, differential recombination (need to define), and subgenomic fractionation are not clearly defined and substantiated by the data. For example, increased level of genetic diversity in subCC could be simply because of the high level of diversity in the progenitor, which is different from the extant *C. canephora* that is sequenced and compared. The correlation with a few NLR resistant loci, including RPP8 homologs, is rather incidental than generalizable.

(11) Retroelements and diversity. This part of data is confirmatory but could be interesting if they can test distribution similarity and difference between subgenomes in allotetraploids.

(12) Discussion. Need to be revised to focus on a few main findings and new perspectives to the field of coffee genomics, polyploid evolution, and crop domestication.

(13) Methods. Simplified and standardized like pipelines. Some of lengthy descriptions in main text could be moved here or as supplemental materials.

In sum, the report will provide another set of useful genomic resources for the coffee research community. The sequence of allotetraploid coffee (and possibly diploid progenitors) is relatively incomplete, and there are conceptual confusion and misunderstanding with respect to polyploidy and subgenome evolution. The quality of most, if not all, figures does not meet standards for any reputable journals, let alone NG. Most conclusions are confirmatory with no insights into what is known in polyploid plants and genomic contribution to domestication. The report is excessively long and poorly written with little effort on justifying biological significance and broader readership. Some additional expertise and skills will be needed to organize large genomic datasets and enunciate a comprehensive genomic paper.

Reviewer #2:

Remarks to the Author:

The manuscript entitled „The genome and population genomics of allopolyploid *Coffea arabica* reveal the diversification history of modern coffee cultivars“ by Salojärvi and colleagues provides yet another genome assembly attempt for the tetraploid, allopolyploid *C. arabica* genome together with improved genome assemblies of *C. canephora* and *C. eugenioides*. The additional analysis of multiple wild and cultivated coffee accession together with the sequencing of an old herbarium sample sheds some light on the history of modern coffee.

A well done and final *C. arabica* would likely help the community as a great resource. At the moment there exist already three *C. arabica* genome assemblies, two of which are of lesser quality i.e. Tran et al and Scalabrin et al. which this genome was benchmarked against as well in table S3.

However the publicly available genome by the JHI? consortium (Zimin et al from the Aldwinckle lab NCBI : RHJU01000000 or GCA_003713225.) unlike the other two previously journal published genomes is chromosome scale, and it seems publicly available without restrictions. More importantly I find it where I expect it (NCBI or EBI). The *C. canephora* genome presented here, is however improved and this manuscript shows a population genetics study shedding some light on coffee history. Interestingly GCA_003713225 is referred to in Table S11 however mostly S11 gives short and cryptic names and shows some numbers without context.

This manuscript clearly has been in the conception stage for some time as it often the case for genome papers. However, the authors should have taken care to delete duplicated and outdated tables and remove information which is no longer needed and given descriptions for supplementary tables within the excel file. I tried to follow some data and have indicated some improvements here and as minor comments, but I would strongly suggest to carefully go through all the data and make data consistent. In addition the whole manuscript could be significantly shortened and flow improved. The authors might have considered to sequence the genome with modern techniques (i.e. HIFI Pacbio) as the major insights stem from the population genomic data which are interesting, but don't quite live up to the promise of the introduction about coffee history and do not necessarily rely on the last genome version.

In detail, the genome was sequenced with old PacBio technology. This is reflected in e.g. BUSCO values of >92% which are not necessarily state of the art. Indeed, the two diploid progenitors show much better BUSCO values of above 96 and 97% respectively and the Aldwinckle genome is reported to have 97% BUSCO completeness by the NCBI data hub albeit using eudicots BUSCO v4 versus embryophyta V4 here (1906) version.

In addition, table S3 (I assume it to be proteome see below but it is not clearly stated) shows a duplication rate of 58.9%. Given the allopolyploidy of *C. arabica* most genes should be duplicated as they should be present in both progenitor subgenomes. Indeed, the Aldwinckle genome has a duplication rate of 67% closer to the expectation at least as reported by NCBI. That said, In Table S11 (which should really have been merged with table S3) the Aldwinckle genome is given with a duplication rate of 58% and a cryptic *C. arabica* v0.6 (likely the presented genome based on the following numbers) is reported with 64.7 % showing at least some improvement.

Consequently, at the moment there are 89% of the genes placed on the *C. arabica* genome (1232) and more than 20% of the genome remain is not placed onto chromosomes (Table S3). In addition, contig N50 length remains at 278kbases (Table S3) leaving room for yet a few other rounds of *C. arabica* genome publications?

I think the authors assembled 22 chromosomes (11 from the *Cc* and 11 from the *Ce* progenitor), I would not necessarily call it phased even though this is semantics as this is a di-haploid, but I would associate 44 chromosomes with it ($2n=4x=44$) see the patchouli genome (with >95% placement) which would be a more prominent technical achievement.

Also the population genomics data brings in new diversity estimates but a population genomics study using GBS which has of course lower sampling density was performed by the Benoite lab already (Scalabrin).

Finally, the genome does not seem to be available in NCBI/EBI/DDBJ or NGDC of China but has been submitted to "Coge".

Minor comments

l197 Does spanning ... projected genome size refer to total assembled space or does this refer to the cytological data please better describe this in the text (cytological data is meant based on supplements)

l202 I failed to find the data on the 91 F1 resequenced samples under the project submission which were used for map construction

l218 "numbers of LTRs (Supp Fig, Table S)." Numbers missing

l226 Tables S9 and S10 was really only one tissue used for Isoseq annotation ? This is not state of the art anymore

figure 2 has generally a low quality and some text looks squashed

l299 maybe describe genome shock a with a few extra words as in the de Meaux group paper

Figure S30 please describe meaning of colors in legend

Figure 4 text is cut off in the sample labels for longer sample names

L448 a typical mutation rate for peach was chosen. As the Arabidopsis mutation rate is very similar this generally seems to be a reasonable estimate. However, Ref 59 and other made the observation that hybrids might show differences in mutation rate see also Bashir et al., 2014 Plant Physiol. Would this potentially affect the calculations given the nature of the *C. arabica* genome? Also Scalabrin tested different mutation rates.

L647 should mention that this is the only significant GO category? This would make the statement stronger

L670 and Figure 8 brings in nucleotide diversity again split into synonymous and non-synonymous, which had been discussed as general total nucleotide diversity already around line 393. It would likely make the manuscript easier to read if this were condensed and it should have been discussed in the light of Scalabrin values which are lower (likely due to the limited GBS technique) but in the same order of magnitude

L 759 using a relaxed p-value but a minimal 2x change seems odd

Table S2 number of gaps "I 0" (does this mean 10 ?)

Table S4 shows different scaffolding steps. I would expect one final result and maybe a validation but not the individual steps anymore.

L907 mentions at least 94% BUSCO (this is a result which shouldn't be in the methods section) but it contrasts the 92.7% in the result (After following all data I think the 92.7 is predicted proteins, the 94 is genome based, but this should be obvious and not left for the reviewer and or reader to determine)

L877 ftp solgenomics is a bit unspecific there were multiple coffee folders but I didn't find vcf data, please specify exact location

Table S31 why to use Arabidopsis annotations? The supplemental material mentions Blast2GO annotations which as all multi-evidence pipelines should be clearly superior ? If it is only to identify orthologs an ortholog pipeline could have been used.

Inconsistent use of L50 and N50. The manuscript chooses to use L50 and length (which makes sense but is uncommon) but Table S4 uses N50 for the same measurement.

It would help to use a nomenclature like Aldwinckle and call the chromosomes 1c and 1e for the one from *canephora* and *eugenioides*.

I did not find any conflict of interest statement.

Reviewer #3:

Remarks to the Author:

The article describes the sequencing and annotation of a di-haploid *Coffea arabica* accession and modern representatives of its diploid progenitors *Coffea eugenioides* and *Coffea canephora*. It shows that there is no subgenome dominance in terms of preferential gene losses for one subgenome (biased fractionation) or transcription levels. A set of 39 wild and cultivated accessions was analyzed to understand the diversification history of *Coffea arabica* and to identify probable origins of cultivars. These results are of great interest and significance to the field. Data and methodology are valid, although I am not an expert of population genomics and am not able to judge that part of the manuscript.

However, a few concerns need to be raised.

1/ Figures are globally of poor quality: most of the text is impossible to read (too tiny). There are too many panels, too small and not informative enough. Most figures are very technical, and hard to understand from the legends (the reader needs to go through the supplementary material in order to

understand what is depicted). The authors should consider reducing the number of panels in the figures, making sure all the text is easy to read when printed, and, most importantly, coming up with more intuitive representations for readers that are not experts of the field. Technical figures should remain in the supplementary material.

2/ The article describes gene fractionation (or gene loss) patterns, and also mentions that genes are preferentially lost in tandemly repeated gene arrays: it would be important to check that these losses do not correspond to assembly artefacts. Such artefacts might occur in tandem repeat clusters. Moreover, if homeologous exchanges have occurred, they could also lead to assembly artefacts. Namely a gene would be found on the first subgenome (the two identical sequences would be collapsed) and the counterpart gene –from the other subgenome- would be deleted from the assembly. It is not clear in the text whether homeologous exchanges can occur in *Coffea arabica* (since a reference from 1940 describing cytological observations suggests that disomic inheritance with bivalent pairing of homologous chromosomes largely prevents recombination between subgenomes (line 381) but other sentences mention possible unequal crossing overs for instance). The authors should really take advantage of the high quality data they collected on a tetraploid and its progenitors (relatives) in order to identify homeologous exchanges (or demonstrate that they did not take place). This could be done by looking at the depth of read mapping on the subgenomes.

3/ The 3 genomes were assembled with different methods, and there are discrepancies in the way they are described in Results and Methods sections:

p5: Pacbio Falcon for CC, Falcon unzip (for CE and CA) + HiC for CC and CA.

p27: Contigs: MHAP for CC, chromosome scale: 454 and BAC-end data for CC, HiC for CA ?

The assembly methods should be homogenized and described more precisely and it would be interesting to discuss the possible biases/differences between the resulting assemblies/annotations and how they could affect genome comparisons between the different species.

Can the authors provide a hypothesis as to why the completion is better for CC than for CA, as seen in supplementary materials section 2.3 “telomeric repeats”: 3 chromosomes with telomeric arrays found at both chromosomal ends vs only 1 in CA.

Below are more specific comments:

Abstract line 93 : “and its diploid progenitors” : correct to “modern representatives of its diploid progenitors” as in discussion line 801.

Line 217: what are the hypotheses for the intriguing observation that both subgenomes showed very similar numbers of LTRs. Numbers of Supp Fig and Table are missing.

Are the differences in LTR retrotransposon proportions significant on Figure S10 ?

Genome fractionation:

After reading of the supplementary materials, it seems that gene fractionation should be defined in the main text: does it include genes lost in the progenitors, or only genes lost after the tetraploidization? Panels in Figure 2 are not all very explicit: the authors should think of other visualizations to provide key messages about gene fractionation (Panel D is very hard to understand without context and could be left in the suppl material) Panel B is not straightforward to understand neither. Maybe a chart

showing the chronology of events (speciation, tetraploidisation...) would help. On panel A, fonts are very difficult to read. I would suggest to name CA chromosomes subCC1 ->subCC11 and subCE1 -> subCE11 (or any nomenclature allowing to relate each CA chr to its progenitors and link homeologous pairs).

Lines 266-280 are hard to understand: it is much easier after reading the supplementary material, but the authors should make an effort to simplify the take-home message (with a simplified figure).

Figure 3 ; the text is impossible to read. The "expression dominance" is not obvious from the figures, maybe because it is impossible to know where the subCC and subCE genes are.

Figure 4: the text cannot be read

Figure 5: panel A is too large, panels B and C are very tiny (and text impossible to read).

Line 646 : could the authors define the term "altering" (the sentence is hard to understand).
 Lines 653-654 : is the diversity in resistance genes reduced in CA? The paragraph is hard to follow: is it mentioning that there are numerous disease resistance genes with altered alleles in CA? Then how/why have they been selected?

Figure 8: again the text is impossible to read.

Line 687 : Coffeaa -> Coffea (I noticed a few other typos in the suppl materials but will not go into detail: I suggest the authors to check carefully)

Figure 9 : panel b is too tiny, it is impossible to read the text.

Line 743: gene conversion occurring between RPP8 paralogs would be expected to homogenize the sequences. However, the authors write that RPP8 shows a great amount of variation. Could the authors discuss that point?

Line 768 : what would be the mechanism for purifying selection ? Is it compatible with the fact that recombination between subgenomes does not take place in Coffea Arabica (line 381) ?

Line 772: tandem duplicates are known targets of homeologous exchange and gene conversion events: do such events occur in Coffea arabica?
 At the end of the paragraph, references showing the similarity between functions of TIP-overlapping and tandem duplicates are proposed for other plants, but are there similarities between genes in TIPS and tandem repeats in coffee?

Line 785 : "direct involvement": what is involved? TIP insertion? Maybe reformulate, the sentence is hard to understand.

Line 786 : please define "biosynthetic gene clusters"

Discussion: lines 819-821, the fact that genomic excisions are occurring within tandemly duplicated genes raises some questions: is the assembly contiguous around tandemly duplicated genes? Could assembly biases in tandem repeats lead to gaps in the assembly and to erroneous gene fractionation

detection?

Lines 825-826 : unequal crossing overs could lead to excisions in one subgenome, but it would be interesting to look at the homeologous region: is it single copy or was it duplicated (◊ homeologous exchange) ? It could be checked with the depth of mapping of reads : it would be double that of other regions.

(redundant with point 2/ above)

Line 939 : Do the previously published expression data correspond to the same accession as the sequenced CA?

Figure S9 : "differences were obvious" : are they statistically significant?

Figure S11 : "slightly more" -> what is the statistical significance?

Paragraph 2.5 of suppl material: could the authors define the "AED score" ? Is it supposed to be higher when genes have more evidence? Then if set to 1 for genes without any support, how can there be "high evidence gene models with AED score <0.5" ?

Suppl figures would gain in clarity if they contained panels A,B, etc: they would be easier to call than "left panel" / "right panel", etc...

Paragraph 3.1 : fractionation in Coffea.

Here the numbers are very hard to follow in a text and it would be very helpful to have a chart displaying the different categories, with links between the 4 genomes (CC, CE, subCC, subCE, as for instance in Figures S13 and S15 in the article describing the Brassica napus genome (DOI: 10.1126/science.1253435).

(The parts describing "gene fractionation" patterns for Figures S24 and S25 are also hard to follow and would gain in clarity with such a chart).

After reading this paragraph, it becomes hard to understand whether gene fractionation means "gene loss" (that could have occurred in the progenitors before the allotetraploidization event), or whether it corresponds to genes lost after the duplication (genuine gene fractionation). The distinction should be made clear in the main text.

Figure S26 : the distributions look slightly different between subCC and subCE: any hypothesis why?

Figure S29 : on the bottom panel there are no gene IDs

Paragraph 4.1.1: the genes (from subCC) are expressed "differently" than their homologs in C. canephora: are they over or under expressed? It is written that it drastically reducing caffeine accumulation : how? The following sentence is saying that subCE genes are significantly less expressed than subCC genes, "explaining the lower caffeine accumulation" in Arabica compared to CC. That is easier to understand (and maybe sufficient?).

Author Rebuttal to Initial comments

Reviewers' Comments:

Reviewer #1:

Remarks to the Author:

In this report, the authors have generated high-quality genome sequences of cultivated coffee, an allotetraploid that is maintained as di-haploid, and its extant diploid progenitors. The genomic resources will be valuable for the coffee research community, although a similar and prior resource (Scalabrin et al. 2020 Sci. Rep.) has a moderate utility (36/46 ISI/Google citations). However, the report suffers from incomplete sequence assembly, poor quality of most (and probably all) figure panels, confirmatory nature of results, and lack of expertise and skills to organize large datasets and write a succinct genomics paper. The main text consisting of long lists of genomic statistics and features without substantiation of their findings has diminished an appeal to a broader readership in polyploid genetics and genome evolution. Specific comments are as follows.

Thank you for this comment. To clarify, we actually presented three genomes; in addition to an assembly and annotation for *C. arabica* (CA) we also provided an improved assembly of *C. canephora* (CC) and a new *C. eugenioides* (CE) assembly. Here, based on the referee suggestions we now present an additional, high-quality and chromosome-level assembly using the latest PacBio HiFi technology. We apologize for the poor quality of some of our original figures, which had in part been due to copy-pasting problems. We did prepare high resolution figures to be submitted in an accompanying Powerpoint, but this was inadvertently missing from the previous submission, a situation now corrected here. In this revision we have considerably shortened the text, and focus instead on the main messages of the paper.

(1) Although its genome sequence is of high-quality by integrating PacBio and Illumina sequences and SNPs derived from a F1 mapping population (91 plants), the assembly is not as complete as one would have expected. For example, the genome size of 1,088 Mb in the allotetraploid (Coffee arabica, CA) is 30% less than the sum of two diploids, 672 Mb in *C. canephora* (CC) and 645 Mb in *C. eugenioides* (CE). No explanation was offered. Is this possibly related to the di-haploid status (that was first reported by Mendes and Bacchi (1940, J. Agron.)? At the pseudo-chromosome level, they represent only 83% (CC) and 68% (CA, a mistake, could be CE?) of the projected genome size. These and other statistics in the report may indicate relatively incomplete and poor quality of sequence assemblies in both tetraploid and diploid species.

Based on flow cytometry results the predicted genome sizes for CC, CE and CA are 682, 705 and 1,281 Mb (see a new Table 1 in the submitted revision). Thus, CA size is not equal to CC + CE, presumably due to genome fractionation after its allopolyploidization event. The Falcon unzip CA assembly covered 84.9% of the projected genome size; the smaller size is most likely due to the incomplete separation of the highly homologous regions of the C and E subgenomes in the Falcon unzipping step. This was also reflected in CA's BUSCO statistics, in which only 58.8% of the BUSCO genes were duplicated. Another major drawback of the previous assembly (as pointed out by the referees) was the low fraction anchored to pseudochromosomes (62.5%, compared to 82.7% of CC). These drawbacks for some of our analyses have now been corrected by state-of-the-art HiFi sequencing and assembly using Hifiasm, which resolved the highly homologous regions. The new assembly is 1,198 Mb (99.8% of the projected genome size) of which 1,192 Mb (99.3%) is anchored to pseudochromosomes and consists of just 238 scaffolds. The percentage of duplicated BUSCO genes is now 93.6%, reflecting a much improved assembly into the two subgenomes. The difference between the old and the new CA assembly versions is illustrated in Table 1. In terms of gene space, the older assembly is still accurate enough for many analyses, as is illustrated by the syntenic alignment between the new PacBio HiFi (y-axis) and the older assembly (x-axis), see below (and also added as a new Supplementary Figure S18).

As such, our paper now presents two high quality chromosome-level assemblies, for CA and CC, and a long-read-based CE assembly which we were able to scaffold into pseudochromosomes using synteny with CC. Together these genomes constitute a formidable dataset for studying post-polyploidization genome events, and provide novel genomic tools for advancing coffee breeding.

A related comment is on the nomenclature of subgenomes in allotetraploids. By convention, each diploid would be given a genome designation, perhaps C for *C. canephora* (CC, $2n = 2x =$

22) and E for *C. eugenoides* (EE, $2n = 2x = 22$), and the resulting allotetraploid would have the genome designation CCEE ($2n = 4x = 44$). Note that C or E is used as an example and could be replaced by the nomenclature based on cytogenetics and breeding and used in the research community. Using the species names (CC, CE, and CA) to describe subgenome is confusing and cumbersome for polyploid genetics and breeding. As examples, genome designation in wheat is AABB for tetraploid (spaghetti/durum wheat) and AABBDD for hexaploid (bread wheat).

Thank you for the suggestion. The revised manuscript follows the suggested nomenclature.

(2) Genome evolution and fractionation in allotetraploid coffee (Fig. 2). The data presented in this section, including figure panels and description, did not capture much useful information, compared to a previous study in coffee (Scalabrin et al. 2020 Sci. Rep.). The subgenomic stability and conservation has been well documented in wheat, cotton, and Arabidopsis allopolyploids. Discussion was poorly constructed with inaccurate citations. For example, maize was cited to be polyploid species with a hybrid origin (lines 299-301) and again cited as one of recently formed allopolyploids (line 302-305). Whole genome duplication or polyploidy occurred in maize 5-12 million years ago (Schnable et al. 2011, PNAS), and maize is considered to be a paleopolyploid with unknown origin of auto- or allopolyploid (Zhao et al. 2017, Plant Cell). These erroneous statements could stem from conceptual misunderstanding of auto- and allopolyploidy, as well as subgenomic evolution. Most figure panels are not useful and have to be reconstructed to make them informative.

The assembly in Scalabrin et al. is not reference quality, and its size is 1.5 Gb, consisting of 164,254 scaffolds with approximately 19kb N50 value. The assembly is considerably larger than independent flow cytometry-based estimates which hover around 1.28 Gbp. This may mean that the Scalabrin et al. assembly contains several (diploid) genome regions where the haplotypes have been assembled separately, which could make many of the analyses questionable. The analyses of genome fragmentation presented in our work strongly rely on the contiguity of the assembly and therefore are not possible with the Scalabrin et al. version. For example, a scaffold size of 19kb does not allow more than a few genes to lie on the same scaffold, thus making all analyses of tandem duplications, synteny and subsequent fractionation impossible. In our case, our improved assembly of 22 chromosomes includes 14 that were sequenced from telomere to telomere.

General consensus suggests that maize is indeed an allopolyploid (see eg Yin et al 2022) with the subgenomes demonstrating biased fractionation patterns (Schnable et al 2011, Hufford et al 2021), different levels of purifying selection (Pophaly and Tellier 2015), expression dominance (Zhao et al. 2017, Plant Cell), as well as differences in epigenetic profiles and fractionation (Renny-Byfield et al 2017). This makes it a good model for comparing against species with no observed subgenome dominance. Regarding the terminology, we have now clarified the text on polyploids in other crop species; thank you for pointing out the inaccuracies.

We have paid special attention to clarifying the figure panels and reducing their overall number.

Genome shock has two meanings at genomic and expression levels. Wheat, cotton, and Arabidopsis allopolyploids show rapid gene expression changes (shock), while Brassica neopolyploids show rapid genomic reshuffling (Xiong et al. 2011, PNAS). In addition, conservation in old (610,000 years-1 mya) coffee allotetraploids does not mean no genome shock at the time of polyploid formation.

Thank you, we make this distinction now in the text and discuss the shock phenomena on both genome and expression levels. In our view, genome shock is the same overall phenomenon that results from the merging of two genomes which are to some extent incompatible. This manifests itself in different ways, which can be altered gene expression (perhaps arising from differential methylation), or genome rearrangements. In our case, we observed that the subgenome originating from *C. canephora* (subCC) plays a dominant role through biased homoeologous exchange; this result stems from a new analysis requested by Referee 3 (see below).

(3) Genome-wide subgenome expression dominance (Fig. 3). This section is misleading, as it was narrowly focused on only three gene families (NMT, TPS, and FAD). FAD gene family may be of relevance to coffee physiology (flavour, aroma, etc.). A suggestion is to provide a brief view of genome-wide data, followed by FAD gene family. Again, the quality of figure panels is poor and should be redone, e.g., labels were too small to be legible, in addition to the ambiguous patterns.

Thank you for this comment. We had actually done the analysis on the entire genome level, and although the results were reported in the supplementary tables, the Methods description was missing. In the main paper, instead of reporting a negative result of global expression dominance we instead decided to address some gene families (known from studies in the coffee literature) as examples of mosaic subgenome expression, but our point clearly didn't come through. With the new genome assembly we have redone the subgenome dominance analysis while also taking into account the effects of homeologous exchange (which was requested by referee #3 in later comments). The text has been rewritten such that it more emphasizes our results on the transcriptome-wide analysis of expression dominance, and we thereafter provide the three families as example cases.

(4) Origin and domestication (Fig. 4). This section can be substantially reduced to avoid redundant information from the previous paper (e.g., Scalabrin et al. 2020 Sci. Rep.) and retain part of population statistics to show the relevance of domestication. Again, there is a problem of figure quality as noted above (illegible fonts and patterns). For example, what is the X-axis? What are populations 1, 2, 3?

We respectfully disagree regarding redundancy, since our analyses are able to provide a much deeper analysis of the place of origin and domestication for Arabica coffee. Scalabrin et al.

presented a population structure and admixture analyses using data from 698 genomic loci, whereas our analyses are from high coverage full-genome sequencing. Our result on the timing of the allopolyploidization event differs greatly from that of Scalabrin et al, and therefore we considered it important to demonstrate why our result diverges. Besides using less than 700 loci, the assembly in Scalabrin et al. is, again, 25% too large, possibly meaning that many of the marker positions are in fact measuring haplotypes instead of diploid loci. This may in fact explain Scalabrin et al.'s lower nucleotide diversity values, and therefore possibly one of the reasons why they estimated the allopolyploidy event as occurring only 10,000 -20,000 years ago, the second being the assumption of no subsequent bottlenecks in *C. arabica* history.

Populations 1,2, and 3 are ancestral populations inferred by ADMIXTURE software. The method is unsupervised and splits the data into hypothetical ancestral populations that best explain the data in terms of best cross-validation score. We now provide interpretations to the groupings in the text: Population 1 is the cultivar population, Population 2 the wild population, and Population 3 is the set of hybrids with CC introgression. In Figure 3B the x-axis is not calibrated and merely represents the relationships between the accessions, as inferred by alignment of 23,000 (subCC) and 25,000 (subEE) independent SNPs. We now also provide time-calibrated trees in the supplement (see Fig S.42).

(5) Population history (Fig. 5). This part of data is interesting but lacks clarity of polyploidization (estimated ~610 kya) and domestication (8.9 kya) after population split (30.5 kya). It is unclear how they determine these events and what are the representative species or accessions, and if these estimates are supported by breeding and systematic information. The problem of admixture would have complicated their analysis and data interpretation (e.g., x generations within and between different stages of divergence).

Thank you; we have clarified this part in the paper. We obtained the polyploidization estimate from several different analyses, based on fractionation rate, distribution of synonymous mutations between syntenic genes, time calibrated SNP trees, as well as demographic modeling with SMC++ and FastsimCoal. In the case of the timing of the allopolyploidy event, admixture does not confound the analyses, since the ancestral populations are derived from the same event. For the population split and domestication event, the timing is obtained from FastsimCoal2 analysis, together with the observation that the cultivars with few breeding cycles and a set of wild individuals appear to be admixed. Since *C. arabica* stems from one allopolyploidy event, there needs to be a population split followed by separation before admixture can arise. Note that the admixture event at 8.9 kya is not necessarily human-induced, it could also be spontaneous, with the beneficial qualities of the ancestral cultivar discovered by humans only later (as has happened in case of Geisha). Domestication could have happened also more recently.

To avoid complications from admixture we carried out analyses with only non-admixed individuals (this gave us the timing of the population split), as well as by combining all sampled individuals, including the admixed ones (this gives the timing of the admixture event and the migration).

(6) Kingship estimation and origin of cultivated species (Fig. 6). This part of information is rather confirmatory with the previous finding (Scalabrin et al. 2020 Sci. Rep.). Again, the figure panels and labels are illegible and confusing.

We respectfully disagree, as Scalabrin et al. did not analyse kinship, but instead only showed a general split into two populations. Our results show that all cultivated Arabica coffees are highly interrelated, even to the extent that the Bourbon lineage could be the parent of the Typica group. Furthermore, the novel cultivar Geisha, which has been considered by some a new lineage of cultivars, in our results appears to be a wild coffee that may date back to the same admixture event that resulted in cultivated Arabica lineages.

(7) Relationships between wild and cultivated Arabicas (Fig. 7). This part of data can be presented in combination/comparison with previous sections with comments in (5) and (6). Many data figures in these sections can be moved into supplemental information.

This section illustrates how genome data can be used to track down and identify the relationships between different individuals. In addition to the known relationships, we identify a parent-child relationship between Bourbon and Typica, and that the cultivars stem from the same ancestral admixture event. We have condensed the text, combined figures into multi-panel plots and moved Figure 8 into supplementary.

(8) Time of admixture and effect on domestication. These could be succinctly discussed along with origin, evolution, and domestication in previous sections. Alternatively, the previous sections could have been substantially reduced to make a few succinct statements.

Thank you, we have followed the second proposal, since we felt that the additional analyses provided in these parts required deeper elaboration that could not be obtained if combined with the overall analysis presented earlier.

(9) Mutational load during domestication (Fig. 8). This part of data adds little value and should be moved to the supplemental information.

We have moved Figure 8 to supplementary and revised the text to summarize the main findings.

(10) Introgression (Fig. 9). The relationships between introgression/heterozygosity, differential recombination (need to define), and subgenomic fractionation are not clearly defined and substantiated by the data. For example, increased level of genetic diversity in subCC could be simply because of the high level of diversity in the progenitor, which is different from the extant *C. canephora* that is sequenced and compared. The correlation with a few NLR resistant loci, including RPP8 homologs, is rather incidental than generalizable.

For the Timor hybrids, we know that the introgression happened with the Typica lineage, since no Bourbon has been cultivated in Timor. Therefore the increased level of nucleotide diversity in subCC (when compared to Typica diversity) is most likely due to the introgression event. We did not compare the diversities against the *C. canephora* (or *C. eugenioides* modern representatives). We have clarified the definitions now.

We have added a sentence in the manuscript which explains how introgression and admixture affect heterozygosity and effective population size. The differential recombination is now explained more accurately:

“Most of the introgressed regions shared by all Timor hybrid lines were on chromosomes 1, 4, 7, and 11, where the descendants of two sister lineages (HT832/1 and HT832/2) displayed highly contrasting patterns due to differences in recombination, most likely tracing back to differences in the original sister lineages, followed by varying number of back-cross generations”

In our view, the fact that the analysis finds NLR genes is not a coincidence, because this is the main reason why Timor hybrid lineages are used in breeding. *RPP8* is a gene family with high heterozygosity due to balancing selection and gene conversion, and therefore its reduced diversity in cultivar lineages and thus increased susceptibility makes sense. This provides an exciting hypothesis which should be tested in subsequent work.

(11) Retroelements and diversity. This part of data is confirmatory but could be interesting if they can test distribution similarity and difference between subgenomes in allotetraploids.

Thank you, we have now analyzed transposable element differences between subgenomes and between subgenomes and their respective diploid parents. First, we didn't observe significant differences between the composition of subCC and CC and between subEE and CE (Student's t-test, Fig S9 and S10). Second, we compared the genome wide phylogeny of the reverse transcriptase domain between subCC and CC and between subEE and EE (Fig S12 and S13). We didn't observe any clear amplification of any particular lineage in s allotetraploid subgenomes versus their respective diploid parents (except for the CRM lineage in subEE). Altogether our data suggests very little impact of allotetraploidy on transposable element activity.

(12) Discussion. Need to be revised to focus on a few main findings and new perspectives to the field of coffee genomics, polyploid evolution, and crop domestication.

We have fused Results and Discussion sections and left a much shorter Conclusions section, focused on the aspects suggested by the reviewer.

(13) Methods. Simplified and standardized like pipelines. Some of lengthy descriptions in main text could be moved here or as supplemental materials.

We have moved the detailed descriptions of methods into supplementary material.

In sum, the report will provide another set of useful genomic resources for the coffee research community. The sequence of allotetraploid coffee (and possibly diploid progenitors) is relatively incomplete, and there are conceptual confusion and misunderstanding with respect to polyploidy and subgenome evolution. The quality of most, if not all, figures does not meet standards for any reputable journals, let alone NG. Most conclusions are confirmatory with no insights into what is known in polyploid plants and genomic contribution to domestication. The report is excessively long and poorly written with little effort on justifying biological significance and broader readership. Some additional expertise and skills will be needed to organize large genomic datasets and enunciate a comprehensive genomic paper.

Thank you, we have now addressed these concerns in the revision. The text is heavily rewritten and the supplementary tables and methods are revised. We also present a new HiFi assembly which is of very high quality.

Reviewer #2:

Remarks to the Author:

The manuscript entitled „The genome and population genomics of allopolyploid *Coffea arabica* reveal the diversification history of modern coffee cultivars“ by Salojärvi and colleagues provides yet another genome assembly attempt for the tetraploid, allopolyploid *C. arabica* genome together with improved genome assemblies of *C. canephora* and *C. eugenioides*. The additional analysis of multiple wild and cultivated coffee accession together with the sequencing of an old herbarium sample sheds some light on the history of modern coffee.

A well done and final *C. arabica* would likely help the community as a great resource. At the moment there exist already three *C. arabica* genome assemblies, two of which are of lesser quality i.e. Tran et al and Scalabrin et al. which this genome was benchmarked against as well in table S3.

However the publicly available genome by the JHI? consortium (Zimin et al from the Aldwinckle lab NCBI : RHJU01000000 or GCA_003713225.) unlike the other two previously journal published genomes is chromosome scale, and it seems publicly available without restrictions. More importantly I find it where I expect it (NCBI or EBI).

Yes, this is correct. The quality of our *C. arabica* assembly was also pointed out by referees 1 and 3, and therefore we now present a highly improved *C. arabica* genome assembly using data from PacBio HiFi platform. The assembly is considerably better than any other published or unpublished but publicly available assembly (see Table 1 for comparisons). Our three genome assemblies will be made publicly available upon manuscript acceptance, providing a much needed resource to the coffee community that stimulated our execution of this project.

The *C. canephora* genome presented here, is however improved and this manuscript shows a population genetics study shedding some light on coffee history. Interestingly GCA_003713225 is referred to in Table S11 however mostly S11 gives short and cryptic names and shows some numbers without context.

This manuscript clearly has been in the conception stage for some time as it often the case for genome papers. However, the authors should have taken care to delete duplicated and outdated tables and remove information which is no longer needed and given descriptions for supplementary tables within the excel file.

Yes, a long timespan for our genome project is a correct assumption, and as a result there were indeed many outdated tables. We have now revised all the text and cleaned the tables; we also present a new high quality assembly for *C. arabica* using PacBio HiFi reads. We respectfully refrain from comparing against the Aldwinckle lab assembly, as we've learned that they are working on an updated version and the one in NCBI might be left unpublished. However, we performed syntenic alignments against the GCA_003713225 assembly and found it to be of lower quality than ours; there are large deletions and collapsed parts in the assembly and the BUSCO scores are not as good, see the attached figure, GCA_003713225 is on x-axis and our PacBio HiFi assembly on the y-axis. Note, for example, extensive DNA gaps in the third and fourth x-axis scaffolds of GCA_003713225 compared against our scaffolds 18 and 19 on the y-axis. In contrast, first and second x-axis GCA_003713225 scaffolds are very similar in DNA content to our scaffolds 21 and 22 on the y-axis.

A comparison of the assembly qualities (the same as Table 1 in the paper, but including the GCA_003713225) is given below.

Assembly	C. eug.	C. can.	C. arab.	C. arab. HiFi	C. can. ***	C. arab. ****	GCA_003713225
Projected genome size (Mb)*	682	705	1281	1281	705	1,281	1,281
Total assembly length (Mb)	661	672	1,088	1,198	532	1,536	1,094
% of projected genome	96.90%	95.30%	84.90%	93.50%	75%	120%	85%
N scaffolds	253	3,033	8,474	132	13,345	164,254	2,833
Scaffold N50	61.3 Mb	50.1 Mb	32.7 Mb	53.7 Mb	1.3 Mb	22.3 Kb	42.5 Mb
N contigs	5,736	3,755	11,863	238**	25,216	336,238	3,487
Contig N50 (Mb)	0.4	0.76	0.23	30	0.015	0.005	3.9
Pseudochromosomes (Mb)	n.a.	583	801	1192	364	n.a.	990
% of projected genome	n.a.	82.70%	62.50%	93.10%	52%	n.a.	77%
N. genes	33,505	28,880	56,670	69,314	25,574	78,311	104,680
Genes in pseudochromosomes	n.a.	27,881	50,410	69,067	21,971	n.a.	98,564
% genes in pseudochromosomes	n.a.	97%	89%	99.60%	86%	n.a.	94.16%
BUSCO genome							
complete	96.70%	97.40%	97.60%	97.90%	95.00%	91.00%	96.20%
single	88.50%	94.80%	20.10%	4.30%	92.30%	21.10%	28.20%
duplicated	8.20%	2.60%	77.50%	93.60%	2.70%	69.90%	68.00%
fragmented	1.10%	0.90%	0.80%	0.80%	1.50%	3.50%	1.20%
missing	2.20%	1.70%	1.60%	1.30%	3.50%	5.50%	2.60%
total	2,326	2,326	2,326	2,326	2,326	2,326	2,326
BUSCO annotation							
complete	94.90%	96.20%	92.10%	97.30%	91.50%	87.00%	97.30%
single	82.40%	92.80%	33.30%	4.10%	89.30%	18.60%	18.70%
duplicated	12.50%	3.40%	58.80%	93.20%	2.20%	68.40%	78.60%
fragmented	2.10%	1.50%	2.80%	0.80%	3.40%	4.80%	0.60%
missing	3.00%	2.30%	5.10%	1.90%	5.10%	8.20%	2.10%
total	2,326	2,326	2,326	2,326	2,326	2,326	2,326

I tried to follow some data and have indicated some improvements here and as minor comments, but I would strongly suggest to carefully go through all the data and make data consistent. In addition the whole manuscript could be significantly shortened and flow improved.

Thank you, we have now made these improvements, which were also suggested by referee 1.

The authors might have considered to sequence the genome with modern techniques (i.e. HiFi PacBio) as the major insights stem from the population genomic data which are interesting, but don't quite live up to the promise of the introduction about coffee history and do not necessarily rely on the last genome version.

Thank you, this is exactly what we have now done and the genome assembly has improved considerably. The size is now 1.25Gb and 97% of the genome is assembled into pseudochromosomes (see new Table 1 for summary statistics). We also analyse homeologous exchange between the two subgenomes, as was requested by referee 3.

In detail, the genome was sequenced with old PacBio technology. This is reflected in e.g. BUSCO values of >92% which are not necessarily state of the art. Indeed, the two diploid progenitors show much better BUSCO values of above 96 and 97% respectively and the Aldwinckle genome is reported to have 97% BUSCO completeness by the NCBI data hub albeit using eudicots BUSCO v4 versus emphyophyta V4 here (I906) version.

Thank you, we would like to point out that our new assembly is now comparable or better than the diploid progenitors with genome-level BUSCO (v5) score of 97.9%.

In addition, table S3 (I assume it to be proteome see below but it is not clearly stated) shows a duplication rate of 58.9%. Given the allopolyploidy of *C. arabica* most genes should be duplicated as they should be present in both progenitor subgenomes. Indeed, the Aldwinckle genome has a duplication rate of 67% closer to the expectation at least as reported by NCBI. That said, In Table S11 (which should really have been merged with table S3) the Aldwinckle genome is given with a duplication rate of 58% and a cryptic *C. arabica* v0.6 (likely the presented genome based on the following numbers) is reported with 64.7 % showing at least some improvement.

Indeed, we agree. The BUSCO score for the new *C. arabica* PacBio HiFi assembly now shows 93.7% duplication rate, which is a considerable improvement on the old ones. We have now merged all the genome statistics into one joint table (Table 1 in main text).

Consequently, at the moment there are 89% of the genes placed on the *C. arabica* genome (1232) and more than 20% of the genome remain is not placed onto chromosomes (Table S3). In addition, contig N50 length remains at 278kbases (Table S3) leaving room for yet a few other rounds of *C. arabica* genome publications?

The new assembly made it now more difficult for subsequent improvements as we now have 68,681 gene models (97%) on chromosomes, leaving only 1,927 unmapped, and the contig N50 is now 30Mb.

I think the authors assembled 22 chromosomes (11 from the Cc and 11 from the Ce progenitor), I would not necessarily call it phased even though this is semantics as this is a di-haploid, but I would associate 44 chromosomes with it ($2n=4x=44$) see the patchouli genome (with >95% placement) which would be a more prominent technical achievement.

Thank you, we have clarified the nomenclature and do not use the term “phasing”, as it could be misunderstood.

Also the population genomics data brings in new diversity estimates but a population genomics study using GBS which has of course lower sampling density was performed by the Benoit lab already (Scalabrin).

Thank you. However, we would like to point out that the estimates from the earlier RAD-seq data also used a low quality assembly; their total assembly size was 1.5Gb when the genome is in reality ~1.25Gb, which means a heavily under-assembled genome. This most likely causes artefacts in SNP calling, since highly heterozygous regions are assembled as separate haplotypes and will show reduced diversity. The lower nucleotide diversity values (and, in turn, very recent coalescence) in Scalabrin et al most likely result from those artefacts.

Finally, the genome does not seem to be available in NCBI/EBI/DDBJ or NGDC of China but has been submitted to “Coge”.

Genome sharing through CoGe platform (<https://genomeevolution.org/coge/>) is becoming quite standard, but we also share it through the Solanaceae FTP site (<https://solgenomics.net/ftp>) maintained by Cornell University Prof. Lukas Mueller. We will also make the assembly public on NCBI upon acceptance.

Minor comments

I197 Does spanning ... projected genome size refer to total assembled space or does this refer to the cytological data please better describe this in the text (cytological data is meant based on supplements)

This refers to cytological data, we have now clarified this.

I202 I failed to find the data on the 91 F1 resequenced samples under the project submission which were used for map construction.

Thank you, this was a very good point. It turns out that the raw data for these accessions was not yet deposited. We have now included it under the *C. arabica* Bioproject ID.

l218 “numbers of LTRs (Supp Fig, Table S).” Numbers missing

The sentence has been removed when revising and shortening the manuscript.

l226 Tables S9 and S10 was really only one tissue used for Isoseq annotation ? This is not state of the art anymore

Yes, there was only one tissue. However, due to work scope, budgetary, and time constraints we have had to restrict the sampling of more tissues for a follow-up work. There are still many genome sequencing papers which don't do any Isoseq, so it is still an improvement over the standard approach.

figure 2 has generally a low quality and some text looks squashed

Thank you; we believe we have made the plots better. The submission was supposed to include a Powerpoint with high resolution figures but this appears to be missing from the final version. It is now included.

l299 maybe describe genome shock a with a few extra words as in the de Meaux group paper

OK, done; thank you.

Figure S30 please describe meaning of colors in legend

We removed this figure from the Supplementary since it was based on manual annotation of the lower quality PacBio + HiC assembly.

Figure 4 text is cut off in the sample labels for longer sample names

OK, thank you; this is corrected. This is now new Figure 3B.

L448 a typical mutation rate for peach was chosen. As the Arabidopsis mutation rate is very similar this generally seems to be a reasonable estimate. However, Ref 59 and other made the observation that hybrids might show differences in mutation rate see also Bashir et al., 2014 Plant Physiol. Would this potentially affect the calculations given the nature of the C. arabica genome? Also Scalabrin tested different mutation rates.

Yes we agree and express this extent of uncertainty in the text now.

L647 should mention that this is the only significant GO category? This would make the statement stronger

Thank you for the good suggestion, done.

L670 and Figure 8 brings in nucleotide diversity again split into synonymous and non-synonymous, which had been discussed as general total nucleotide diversity already around line 393. It would likely make the manuscript easier to read if this were condensed and it should have been discussed in the light of Scalabrin values which are lower (likely due to the limited GBS technique) but in the same order of magnitude

The purpose of splitting the nucleotide diversity into neutral vs. non-synonymous makes it possible to look into the effects of domestication, which is why we included it here. We have provided explanation and references to relevant literature. The results largely followed what is expected for a recently domesticated crop species. We have shortened the text and moved Figure 8 to supplementary.

L 759 using a relaxed p-value but a minimal 2x change seems odd

The gene expression was done from the same cultivar at different time points, so they are not biological replicates. Pathogen responses are generally assumed to occur in a course of time where different layers of immunity are triggered at different time points as the disease progresses (and at varying levels, depending on the genotype). Therefore, combining the different time points is not an optimal way of analysing inoculation data. However, this was the only way for assessing the statistical significance. Naturally expression differences in the different time points will result in lower average fold changes. We would like to point out that this was the reasoning for coming up with these thresholds; we did not try any other values to avoid overfitting.

Table S2 number of gaps "1 0" (does this mean 10 ?)

This is now new Table 1 that is part of the main paper. The information has been updated and corrected.

Table S4 shows different scaffolding steps. I would expect one final result and maybe a validation but not the individual steps anymore.

We agree, this was an intermediate result; the table has been removed from the submission.

L907 mentions at least 94% BUSCO (this is a result which shouldn't be in the methods section) but it contrasts the 92.7% in the result (After following all data I think the 92.7 is predicted proteins, the 94 is genome based, but this should be obvious and not left for the reviewer and or reader to determine)

Thank you. We have now updated the BUSCO scores to reflect the new assembly. All statistics are now collected in Table 1, and BUSCO scores are given at both genome and proteome level.

L877 ftp solgenomics is a bit unspecific there were multiple coffee folders but I didn't find vcf data, please specify exact location

The ftp site has been under revision after our initial submission. The vcf data is now in the *Coffea arabica* folder in the solgenomics site (the new address <https://solgenomics.net/ftp>); the data is split according to the subgenome (CC or EE) as explained in the Methods section. The same folder also contains the new assembly and annotation.

The vcf files are also accessible from Google Drive

subCC:

<https://drive.google.com/file/d/1jccdlZAMe0rc2sw2ejc4jrlhRzKKlrGa/view?usp=sharing>

subEE:

https://drive.google.com/file/d/1jdr7RA3KkIsVu-wDvSX_3QaqSIHjwoqB/view?usp=sharing

Table S31 why to use Arabidopsis annotations? The supplemental material mentions Blast2GO annotations which as all multi-evidence pipelines should be clearly superior? If it is only to identify orthologs an ortholog pipeline could have been used.

Arabidopsis is the model plant where most of the mechanistic evidence for gene function is validated. BLAST2GO finds the GO annotation of best BLAST hits, but this is usually information that has been computationally transferred from Arabidopsis as the ultimate source. To avoid propagation of this type of indirect evidence, we opted to go directly to the “source”.

Inconsistent use of L50 and N50. The manuscript chooses to use L50 and length (which makes sense but is uncommon) but Table S4 uses N50 for the same measurement.

OK, corrected; thank you. We now report N50 which is more commonly used in genome papers.

It would help to use a nomenclature like Aldwinckle and call the chromosomes 1c and 1e for the one from *canephora* and *eugenioides*.

OK, thank you. Referee 1 also suggested renaming of the chromosomes,; we have followed that suggestion and use the general notation for polyploids, EE for the (diploid) subgenome from *C. eugenioides* and CC for the *C. canephora* subgenome. When necessary we also use subCC and subEE to highlight that we are talking about subgenomes of an allopolyploid species.

I did not find any conflict of interest statement.

OK, added.

Reviewer #3:

Remarks to the Author:

The article describes the sequencing and annotation of a di-haploid *Coffea arabica* accession and modern representatives of its diploid progenitors *Coffea eugenioides* and *Coffea canephora*. It shows that there is no subgenome dominance in terms of preferential gene losses

for one subgenome (biased fractionation) or transcription levels. A set of 39 wild and cultivated accessions was analyzed to understand the diversification history of *Coffea arabica* and to identify probable origins of cultivars. These results are of great interest and significance to the field. Data and methodology are valid, although I am not an expert of population genomics and am not able to judge that part of the manuscript.

However, a few concerns need to be raised.

1/ Figures are globally of poor quality: most of the text is impossible to read (too tiny). There are too many panels, too small and not informative enough. Most figures are very technical, and hard to understand from the legends (the reader needs to go through the supplementary material in order to understand what is depicted). The authors should consider reducing the number of panels in the figures, making sure all the text is easy to read when printed, and, most importantly, coming up with more intuitive representations for readers that are not experts of the field. Technical figures should remain in the supplementary material.

The submission was intended to include a Powerpoint with high resolution figures, but this was inadvertently missing from our upload. We have now clarified and reduced the main text figures, and also included the high resolution files.

2/ The article describes gene fractionation (or gene loss) patterns, and also mentions that genes are preferentially lost in tandemly repeated gene arrays: it would be important to check that these losses do not correspond to assembly artefacts. Such artefacts might occur in tandem repeat clusters.

Moreover, if homeologous exchanges have occurred, they could also lead to assembly artefacts. Namely a gene would be found on the first subgenome (the two identical sequences would be collapsed) and the counterpart gene –from the other subgenome- would be deleted from the assembly. It is not clear in the text whether homeologous exchanges can occur in *Coffea arabica* (since a reference from 1940 describing cytological observations suggests that disomic inheritance with bivalent pairing of homologous chromosomes largely prevents recombination between subgenomes (line 381) but other sentences mention possible unequal crossing overs for instance). The authors should really take advantage of the high quality data they collected on a tetraploid and its progenitors (relatives) in order to identify homeologous exchanges (or demonstrate that they did not take place). This could be done by looking at the depth of read mapping on the subgenomes.

Thank you for this important suggestion. We have now developed a PacBio HiFi assembly that, owing to long, accurate reads greatly reduces the possibility of assembly artifacts due to homeologous exchange. Using this assembly as a reference, we carried out homeologous exchange (HE) analysis for all the accessions in our data, which provided a considerable amount of new information with respect to the genome evolution of *C. arabica*. In general, we identified only one fixed HE event where subEE (the subgenome from *C. eugenoides*) sequence has replaced subCC alleles, and gene ontology enrichment suggested this to be connected with

genes encoding chloroplast-targeted proteins. This clearly makes sense since *C. eugenioides* is the maternal progenitor.

Additionally, we detect several loci where the allele ratio was 3:1 in favor of subCC, and the site frequency spectrum of the Arabica accessions suggested strong selection associated with these loci, suggesting that the biased HE provides an adaptive advantage. We interpret that biased HE represents a new form of subgenome dominance in recent polyploids.

3/ The 3 genomes were assembled with different methods, and there are discrepancies in the way they are described in Results and Methods sections:

p5: PacBio Falcon for CC, Falcon unzip (for CE and CA) + HiC for CC and CA.

p27: Contigs: MHAP for CC, chromosome scale: 454 and BAC-end data for CC, HiC for CA ?

Thank you for pointing this out. The description in the supplementary (and p27) is correct. We also carried out PacBio Falcon for CC, but selected the assembly produced by MHAP since the quality was higher.

The assembly methods should be homogenized and described more precisely and it would be interesting to discuss the possible biases/differences between the resulting assemblies/annotations and how they could affect genome comparisons between the different species.

Since we had different genomic resources available for each species and limited funding, we decided to pragmatically use different approaches that were deemed to give the best end results. Furthermore, since *C. canephora* and *C. arabica* are the commercially important species we decided to focus on their assemblies. This meant leaving *C. eugenioides* in a relatively poorer state of assembly, but we believe that there will be future work on its improvement outside the scope of the current manuscript. In this revision, due to the comments from referees 1 and 2, we carried out a new round of sequencing for *C. arabica* with PacBio HiFi. The best assembly methods for HiFi data are different from the older, more error-prone PacBio data, which further makes comparison and homogenization difficult. We have included in the supplementary (**Fig. S18**) a syntenic alignment between the old PacBio and the new PacBio HiFi assembly, which highlights at least some regions where the assembly was challenging with the old error-prone PacBio data.

Can the authors provide a hypothesis as to why the completion is better for CC than for CA, as seen in supplementary materials section 2.3 “telomeric repeats”: 3 chromosomes with telomeric arrays found at both chromosomal ends vs only 1 in CA.

The new PacBio HiFi data extends the CA chromosomes into telomeric regions, so this feature was only due to assembly quality and read lengths in the raw data.

Below are more specific comments:

Abstract line 93 : “and its diploid progenitors” : correct to “modern representatives of its diploid progenitors” as in discussion line 801.

OK, thank you; done

Line 217: what are the hypotheses for the intriguing observation that both subgenomes showed very similar numbers of LTRs. Numbers of Supp Fig and Table are missing.

Are the differences in LTR retrotransposon proportions significant on Figure S10 ?

Thank you, we were also excited about the finding. Our hypothesis for the homogenization of the TE content is homeologous exchange (HE) between the subgenomes. This is now discussed in the TE analyses section, and we carried out an extensive analysis of HE events in our *C. arabica* accessions, showing that HE is a major contributor to intraspecific diversity even in the cultivated Arabicas.

Statistical significances for individual LTR families cannot be measured (because it's a single value), but we tested for the overall significance in Copia and Gypsy families. The families did not show significant differences between the subgenomes / progenitors (see revised Fig S10 and the associated text). However, individual lineages may still demonstrate large differences which we point out in the associated text.

Genome fractionation:

After reading of the supplementary materials, it seems that gene fractionation should be defined in the main text: does it include genes lost in the progenitors, or only genes lost after the tetraploidization?

The analysis looked at both of these, since the high quality genomes gave a perfect opportunity of analysing it. Most of the differences between the subgenomes were found to exist already in the progenitors. We have clarified this part in the main text and provide a detailed analysis in the supplementary; we have also added a new summarizing Figure S29 which illustrates that the proportions of genes lost is directly proportional to the divergence times of the subgenomes and diploid progenitors.

Panels in Figure 2 are not all very explicit: the authors should think of other visualizations to provide key messages about gene fractionation (Panel D is very hard to understand without context and could be left in the suppl. material) Panel B is not straightforward to understand neither. Maybe a chart showing the chronology of events (speciation, tetraploidisation...) would help. On panel A, fonts are very difficult to read. I would suggest to name CA chromosomes subCC1 ->subCC11 and subCE1 -> subCE11 (or any nomenclature allowing to relate each CA chr to its progenitors and link homeologous pairs).

We have considerably simplified the figure now, such that only the main message is visualized in Figure 1 in the revised submission; more detailed discussion with additional visualizations is put in the supplementary. The linking that has been used in coffee breeding is that odd-

numbered chromosomes originate from CC and even-numbered from CE, such that CA1 = subCC1 = CC1 and CA2 = subEE1 = CE1.

Lines 266-280 are hard to understand: it is much easier after reading the supplementary material, but the authors should make an effort to simplify the take-home message (with a simplified figure).

Thank you, we have now clarified the entire paragraph on fractionation.

Figure 3 ; the text is impossible to read. The “expression dominance” is not obvious from the figures, maybe because it is impossible to know where the subCC and subCE genes are.

This is now Figure 2 in the revised manuscript. We now also provide a high resolution image as powerpoint. The genes located in subCC and subEE are indicated with different colored fonts, we explain this in the figure legend now.

Figure 4: the text cannot be read

Figure 5: panel A is too large, panels B and C are very tiny (and text impossible to read).

Our apologies, we have revised all the figures and provide an additional power point file with high resolution figures. This was also the plan in the original submission but the Powerpoint was lost in the submission.

Line 646 : could the authors define the term “altering” (the sentence is hard to understand).

OK, thank you; we rephrased the sentence to the more commonly used “amino acid changing mutations among cultivars” .

Lines 653-654 : is the diversity in resistance genes reduced in CA? The paragraph is hard to follow: is it mentioning that there are numerous disease resistance genes with altered alleles in CA? Then how/why have they been selected?

We have clarified this paragraph considerably now.

An F_{ST} scan identifies regions that differ between two populations (wild vs cultivars), where a high value can result if individuals in one population are very similar while the other population has more variance; thus F_{ST} is often plotted against nucleotide diversity (π) ratios between two populations, to identify the population with the reduced diversity. A combination of low π and high F_{ST} is then taken as a sign of a selective sweep. In our case we looked for hallmarks of cultivation, i.e., genes with a high abundance of derived amino acid-changing alleles; this could result from selection for a specific derived allele, or hitchhiking effects. The scan identified resistance (R) genes as an enriched group.

In natural populations R genes are among the most diverse ones, both within and between individuals. This is linked with pathogen resistance, since a population with a more diverse R gene palette has different ways of detecting and reacting to the pathogen. In our result, the diversity in an individual cultivated CA is not likely reduced (they have the same number of R genes as any other accession), but at the population level, since most cultivars are derived from one parent-child duo, and usually one cultivar is grown in the field, the population-level diversity is very low, corresponding to a single individual; this is the common problem in all monoculture crops. Therefore the scan identifies a side effect of selection, the reduction in diversity of resistance genes due to breeding.

Figure 8: again the text is impossible to read.

Our apologies, we have revised all the figures and provide an additional PowerPoint file with high resolution figures.

Line 687 : Coffeaa -> Coffea (I noticed a few other typos in the suppl. materials but will not go into detail: I suggest the authors to check carefully)

OK, thank you for pointing this out.

Figure 9 : panel b is too tiny, it is impossible to read the text.

Our apologies, we have revised all the figures and provide an additional PowerPoint file with high resolution figures.

Line 743: gene conversion occurring between RPP8 paralogs would be expected to homogenize the sequences. However, the authors write that RPP8 shows a great amount of variation. Could the authors discuss that point?

The *RPP8* family is among the most diverse NLR gene families in Arabidopsis, and because of that the generative mechanisms underlying its polymorphism has been studied as an exemplar system. Gene conversion between distal sites may homogenize the variation overall, but between two alleles this can lead to accelerated divergence. There are theoretical results showing that intrachromosomal gene conversion increases the diversity, since segments of genes are being copied at the same time, instead of mutational accumulation of one SNP at a time (Innan 2003, Teshima & Innan 2012). When coupled with balancing selection the level of heterozygosity observed in the *RPP8* gene family can be explained (MacQueen et al. 2019).

Line 768 : what would be the mechanism for purifying selection ? Is it compatible with the fact that recombination between subgenomes does not take place in Coffea Arabica (line 381) ?

This part has been removed from the revised version, but in general, gene coding regions and their immediate neighbourhoods are depleted of transposable element insertions (=under

purifying selection). This is due to fitness decreasing effects; even if the gene coding region would be intact, methylation required for silencing the transposable element may affect gene expression and its control.

Line 772: tandem duplicates are known targets of homeologous exchange and gene conversion events: do such events occur in *Coffea arabica*?

Thank you, we have now analysed homeologous exchange between the *C. arabica* subgenomes. We confirm several such events, and it turns out that they contribute to the subgenome dominance in *C. arabica*. Most likely there are also gene conversion events but considered these outside the scope of the current paper.

At the end of the paragraph, references showing the similarity between functions of TIP-overlapping and tandem duplicates are proposed for other plants, but are there similarities between genes in TIPS and tandem repeats in coffee?

The overlap between TIPs and tandemly duplicated genes was significant; we have clarified and shortened this in the revised manuscript.

Line 785 : “direct involvement”: what is involved? TIP insertion? Maybe reformulate, the sentence is hard to understand.

We have now shortened this section considerably; now we only mention the significant overlap between TIPs and biosynthetic gene clusters. A detailed analysis will be exciting but better to be left outside of this paper, as a conclusive study would need the analysis of a set of several high quality genomes.

Line 786 : please define “biosynthetic gene clusters”

OK, thank you; done. Added also reference to a recent review on the topic.

Discussion: lines 819-821, the fact that genomic excisions are occurring within tandemly duplicated genes raises some questions: is the assembly contiguous around tandemly duplicated genes? Could assembly biases in tandem repeats lead to gaps in the assembly and to erroneous gene fractionation detection?

We have now carried out an improved assembly for the CA subgenomes using PacBio HiFi. The nearly error-free long read sequencing allowed us to assign most of the assembly into pseudomolecules, and syntenic alignments against the old assembly showed considerably improved assemblies in, e.g., repeat-rich pericentromeric regions. Therefore repetitive regions are assembled much more accurately. In the improved assembly we still see strong evidence of excisions being the main mode of fractionation, so the result was not due to assembly problems.

Lines 825-826 : unequal crossing overs could lead to excisions in one subgenome, but it would be interesting to look at the homeologous region: is it single copy or was it duplicated (with homeologous exchange) ? It could be checked with the depth of mapping of reads : it would be double that of other regions.
(redundant with point 2/ above)

Thank you for the suggestion, we have now included HE analysis to the paper. In sum, we find quite a large amount of HE between the subgenomes, showing a significant bias from subCC to subEE (see new supplementary Section 4.1)

Line 939 : Do the previously published expression data correspond to the same accession as the sequenced CA?

No, this is not the same accession; our apologies for any confusion. While for *C. arabica* we believe this is not an issue (due to low diversity in the species), this is not necessarily so in case of Timor hybrid based individual with introgression from *C. canephora*. For an in-depth analysis a separate long read assembly of Timor hybrid would be necessary. However, this is outside of the scope of the current paper.

Figure S9 : “differences were obvious” : are they statistically significant?

With a single reference genome (=one replicate) per species it is not possible to assess significance of single families. However, also the underlying data has now changed with the new genome assembly and transposable element annotation. We tested overall statistical significance for the families with a paired t-test; we did not obtain significant differences between the subgenomes or diploid progenitors. The p-values are reported in the supplementary. Even though there are no differences in repeats overall (because when one family is increased, some other family must go down in proportion), there are still noteworthy differences in individual lineages, which is why we want to point them out.

Figure S11 : “slightly more” -> what is the statistical significance?

The analysis and the figure has been redone for the new genome assembly. We tested statistical significance of the TE families; we did not detect significant differences.

Paragraph 2.5 of suppl material: could the authors define the “AED score” ? Is it supposed to be higher when genes have more evidence? Then if set to 1 for genes without any support, how can there be “high evidence gene models with AED score <0.5” ?

Thank you for pointing this out. We have now explained the annotation edit distance score briefly and provided a reference. The score 1 corresponds to a case with no support whereas score 0 denotes perfect agreement with supporting evidence. Additionally, there was a mistake in the explanation of the AED filtering; it now reads “High evidence gene models with AED score <0.5 were selected for the annotation.”

Suppl figures would gain in clarity if they contained panels A,B, etc: they would be easier to call than “left panel” / “right panel”, etc...

Thank you, we have followed this suggestion.

Paragraph 3.1 : fractionation in Coffea.

Here the numbers are very hard to follow in a text and it would be very helpful to have a chart displaying the different categories, with links between the 4 genomes (CC, CE, subCC, subCE, as for instance in Figures S13 and S15 in the article describing the Brassica napus genome (DOI: 10.1126/science.1253435).

We have improved the figure and now illustrate the proportions of genes lost during common ancestors of subEE and CE, as well as subCC and CC, see **Figure S29**. The figure illustrates that the proportion of lost genes is directly proportional to the divergence times of the different genomes. Moreover, we also show that the synonymous mutation peaks coincide with the *C. arabica* speciation time. They also largely illustrate that we do not see a genomic shock in these processes.

(The parts describing “gene fractionation” patterns for Figures S24 and S25 are also hard to follow and would gain in clarity with such a chart).

After reading this paragraph, it becomes hard to understand whether gene fractionation means “gene loss” (that could have occurred in the progenitors before the allotetraploidization event), or whether it corresponds to genes lost after the duplication (genuine gene fractionation). The distinction should be made clear in the main text.

Gene fractionation is the same as gene loss, but concerns events where one or more genes are excised from the genome, which could originate from multiple different points in time, either after the divergence of the progenitors or following the allopolyploidy event. We also consider loss events before the allotetraploid event; the loss rates are similar before and after the event, as illustrated in the new Figure S29.

Figure S26 : the distributions look slightly different between subCC and subCE: any hypothesis why?

This comment refers to the new Figure S27. Our interpretation is that the differences are only due to random variation; subCC has more fragments of ~200 bp while subEE has more fragments of ~400bp. The most important message from the barplot is that there are surprisingly few fragments overall, and when there are fragments they are much shorter than the average length of a gene. The situation would be very different if the main mode of gene loss would be pseudogenization.

Figure S29 : on the bottom panel there are no gene IDs

This figure has been removed from the revised supplementary when we shortened and developed a more coherent manuscript.

Paragraph 4.1.1: the genes (from subCC) are expressed “differently” than their homologs in C. canephora: are they over or under expressed? It is written that it drastically reducing caffeine accumulation : how? The following sentence is saying that subCE genes are significantly less expressed than subCC genes, “explaining the lower caffeine accumulation” in Arabica compared to CC. That is easier to understand (and maybe sufficient?).

Yes, thank you; it is sufficient, and we follow this suggestion now.

Decision Letter, first revision:

27th Jul 2023

Dear Professor Albert,

Your Article, "The genome and population genomics of allopolyploid *Coffea arabica* reveal the diversification history of modern coffee cultivars" has now been seen by 3 referees. You will see from their comments below that while they find your work of interest, some important points are raised. We are interested in the possibility of publishing your study in Nature Genetics, but would like to consider your response to these concerns in the form of a revised manuscript before we make a final decision on publication.

To guide the scope of the revisions, the editors discuss the referee reports in detail within the team with a view to identifying key priorities that should be addressed in revision. In this case, we think all referees have provided constructive reviews aimed at strengthening the analyses and improving the presentation and data sharing, and we particularly ask that you address their technical comments as thoroughly as possible with appropriate revisions. We hope that you will find the prioritized set of referee points to be useful when revising your study.

We therefore invite you to revise your manuscript taking into account all reviewer and editor comments. Please highlight all changes in the manuscript text file. At this stage we will need you to upload a copy of the manuscript in MS Word .docx or similar editable format.

*2) If you have not done so already please begin to revise your manuscript so that it conforms to our Article format instructions, available here.

*3) Include a revised version of any required Reporting Summary:

Please be aware of our guidelines on digital image standards.

[redacted]

We hope to receive your revised manuscript within 3 to 6 months. If you cannot send it within this time, please let us know.

Nature Genetics is committed to improving transparency in authorship. As part of our efforts in this direction, we are now requesting that all authors identified as 'corresponding author' on published papers create and link their Open Researcher and Contributor Identifier (ORCID) with their account on the Manuscript Tracking System (MTS), prior to acceptance. ORCID helps the scientific community achieve unambiguous attribution of all scholarly contributions. You can create and link your ORCID from the home page of the MTS by clicking on 'Modify my Springer Nature account'. For more information please visit please visit www.springernature.com/orcid.

Sincerely,
Wei

Wei Li, PhD
Senior Editor
Nature Genetics
New York, NY 10004, USA
www.nature.com/ng

Reviewers' Comments:

Reviewer #1:

Remarks to the Author:

In this revision, the authors attempted to improve the quality of sequence assemblies as well as data interpretation and presentation. However, the revision still suffered from conceptual confusion, inaccurate description of data, poor quality of most figure panels, and lack of appeal to broader readership in polyploid genetics and genome evolution.

(1) The introduction is solely focused on coffee and has no discussion about its broader impact on polyploid genetics and genome evolution.

(2) Completeness of the genomes: Despite the genome assembly was improved, they failed to explain why the tetraploid genome is substantially smaller than the extant diploid progenitor-related species, as they stated that the two subgenomes are highly conserved and maintained. Even with improved PacBio HiFi and Hi-C approaches, the CA (HiFi) genome (1,198 Mb) is ~90% of predicted from the sum of the two (Table1). By the way, Table 1 seems to indicate only the CA genome (CA-HiFi) was improved, while CC and CE genomes remained unchanged. Table 1 should be simplified with fewer sequence statistics, excluding unimproved CA. The statistics of previously published genomes should be eliminated.

The usage of terms is still confusing in the manuscript. The definition should be clearly stated in the text using *C. canephora* (CC, $2n = 2x = 22$), *C. eugeniodies* (EE, $2n = 2x = 22$), and (CCEE, $2n = 4x = 44$), respectively.

(3) Their response to the comment on allopolyploidization is simply incorrect. Maize is a diploidized tetraploid (paleopolyploid), while coffee, cotton, and oilseed rape are clearly allotetraploid. Maize does not have subgenomes in terms of cytology and homoeologous chromosome pairing. In addition, they cited some work on *Arabidopsis* interspecific hybrids, which are not allotetraploids; there is a plethora of work in *Arabidopsis* allopolyploids for them to cite and discuss. Their discussions and reference citations on allopolyploid subjects are inaccurate. As noted, the group lacks expertise in polyploid genomics and genome evolution.

(4) Lines 182-193: TE results did not support rapid genomic reshuffling or homoeologous recombination. If they speculated a similarity to the rapid genomic reshuffling, as observed in Brassica or *Tragopogon*, they should provide cytological evidence. Moreover, this statement is contradictory to the results of sequence assemblies, while the subgenomes were highly conserved relative to their extant donor-related genomes.

(5) Conservation vs. subgenomic fractionalization – confusion and confusion. “Comparison of the Arabica CC and EE subgenomes against...revealed high conservation in terms of chromosome number, centromere position, and number of genes per chromosome.” (lines 206-212) Moreover, there is little evidence to support fractionation of CC or EE subgenome after tetraploidization (Fig. 1B) (lines 227-233). The blocked area (Fig. 1C) is more related to poor assembly in the centromeric than in other regions. In summary (lines 249-257), as noted in the previous comments, “genome shock” was defined at the EARLY EVENTS of interspecific hybridization (McClintock 1984, Science) or polyploidization (many papers not cited). It has nothing to do with what was observed in *C. arabica*. In addition, there is no evidence of mis-regulation of transcriptome and epigenome ... in rampant activation of TE elements (lines 252-253).

(6) Gene-family subgenome expression dominance. This is like the fractionation is more nonsense than evidence. “The global expression patterns did not show significant subgenome expression dominance patterns in different bean developmental stages...” (lines 267-269). The genes involved in three pathways were hand-picked and did not show any expression dominance. For example, homoeolog expression difference was observed in only 1 out of 5 NMT gene family members, with similar homoeolog expression patterns for all other family members (Fig. 2B). Moreover, the

description of subCC or subEE expression dominance in the text is incorrect when an individual gene or family member was concerned. It should be expression difference between CC and EE homoeologs, instead of subCC and subEE genomes. Fig. 2C is useless and should be removed. Again, labels in Fig. 2A and Fig. 2B are barely visible. They have not fixed the problem of poor quality in data presentation.

(7) Homoeologous exchanges (HE) were rare and did not contribute to the origin and domestication of *C. arabica* (lines 347-371). Why did they want to twist their arms and emphasize this?

(8) Population history: There is more speculation than evidence (lines 406-423), as they also noted at the end of discussion. The figure legends in Fig. 3 are poorly constructed. For example, the legends in Fig. 3F should reflect the content of the figure panel but not the conclusion of the data. This problem exists across all figure legends.

(9) Kinship estimation (Fig. 4). The diagrams and legends are so confusing by referencing several different supplemental figures with multiple-coloured backgrounds that fail to help understand the results and discussions in the text. For thumbnail images...what are the x-axis and y-axis? What is the second- and third-degree relationship? The overall presentation is too long with too many speculations through the analysis of selected genes.

(10) Introgression (Fig. 5). The figure cannot help substantiating the results and discussion, and the legends lack clarity of elements present in the figure. For example, which dot/colour represents which cultivar or accession/pair analysed (Fig. 5A)? The analysis of TEs (Fig. 5B) is incomprehensive. Colour scale is missing, and the label of "16" (top of the bottom panel) should be an error. Again, the labels and fonts are barely legible.

(11) The manuscript can be improved with language skills and technical issues. For example, Fig. S29 appeared after Fig. S1; no other Fig. S was found in between. Some citations are confusing such as Figs. 3B, S42, etc. Another bigger issue is about absent or mis-citation of most, if not all, original literatures related to stable allopolyploidy, homoeolog expression dominance, and epigenomic diversification. There is no data to speculate epigenetic regulation. This along with genomic studies in other allopolyploid species can become a paragraph in the Introduction.

In sum, the authors have attempted to improve the data presentation and interpretation. The revision has retained some old problems such as lack of significance and advance in polyploid genomics and evolution (incorrect interpretation and references of genome shock, subfractionation, and genome expression dominance) and technical validity (poor qualities of data and figures and incomprehensible legends). As a result, the revised manuscript does not meet the standards for publication in this journal.

Reviewer #2:

Remarks to the Author:

The authors have addressed my and referee #1's main concern, which is the subpar assembly, and have given sensible names to the chromosomes. While the assembly of coffee has been improved, the authors had to make some efforts to obtain this genome at the current level, resorting to gene-base sorting, likely given that hifi sequencing reads were only provided at 10x, which is low for hifi assembly. Similarly, as an answer to reviewer #3, the authors mention limited resources, which,

given the advent of the Pacbio Revio, should not have been an issue but of course would have needed even more reanalyses. Now, in any case, the genome has improved substantially over the last version and is much more usable and better than the one from Aldenwinkle, both in assembly metrics and in gene completeness, including a sensible (high) number duplicated BUSCO values, making it the best assembly at the time - until the mentioned improved one by Aldenwinkle comes out.

While I completely disagree with not sharing data, the authors pledge to share data via NCBI upon publication and have uploaded it to solgenomics, which, for historical reasons, has supported the coffee community.

However, I still can't find either the genome or the vcf data in any of these resources, and I find it a bit odd that I should try out all folders in the solgenomics ftp site. It was for sure not in coffee, as data was from 2005, nor in genomes *Coffea canephora* or *coffea humbotiana*, but I trust Dr. Muller will make it available. In addition, when I searched on COGE with the keywords *Coffea*, coffee, or arabica, I was only able to retrieve GCA_003713225. Coffeabase was password protected, and the only mention of PRJNA698600 I found was in another coffee manuscript. Of course, it would have made it easier to actually check any claims made by the authors if only a reviewer link (which NCBI provides for this reason) would have been provided.

Regarding my comment with Blast2GO, the authors themselves show a picture of what Blast2GO is - it is much more than just a "BLAST" to Arabidopsis but can include profile scans and also does some computation of results - like any of the modern (plant) functional annotation pipelines, which consider different evidence chains. What is worrying for me is that none of these require much effort and indeed can be used as web services as well. Additionally, I want to point out that, of course, we are not in 2000 anymore, and most plant research is done on crops by now, so we have a lot of experimental evidence for crop species. Since I couldn't obtain the annotation files, I couldn't provide the authors with autolinks to some annotated resources.

Finally, the figures are still not of decent quality as they contain wiggly red lines under names - this is really easy to fix, so why this wasn't done I don't quite get.

I would expect some care to quality given the outlet.

Reviewer #3:

Remarks to the Author:

The article presents a new version of the previously submitted one, with an improvement of the *Coffea arabica* assembly using HiFi reads, as well as additional analyses, in particular the identification of homeologous exchanges. The authors addressed all the points previously raised by the referee and made the manuscript easier to read. The article provides interesting results regarding homeologous exchanges in tetraploid coffee, as well as evidence for gene fractionation through excision, and description of a mosaic pattern of subgenome expression dominance. In addition, the resequencing of 39 wild and cultivated Arabica accessions allowed to an in-depth analysis of the breeding history and dissemination routes of *C. arabica*.

Below are a few questions to address:

1/ The number of genes annotated is much higher on the new HiFi assembly than on the previous assembly, and each subgenome contains several thousands more genes than the progenitor genomes. How can the author explain this finding: were there genes missed in other annotations (maybe

tandemly repeated genes, less correctly inferred in more fractionated assemblies), or is there an over prediction in *C. arabica* (HiFi) ? Can it be caused by the annotation transfer performed with GeMoMa ? Also, in table S5, some new models do not correspond to old models, but if the new models were obtained by transfer of old annotation, how can there be more?

2/ line 285 ; « at decreased level » (add « compared to subCC » ?)

3/ Line 301 : « seed-type FAD2 gene expression was dominated by subCC » . This statement is not clear from figure: gene « Cara002g032150 » seems the most expressed (subEE), can the authors be more precise ?

4/Line 355 : surprisingly all but one accession showed significant 3:1 towards subCC.

Supp data & Figure S36 : BMJM has a significant bias towards subEE (specific subEE region in the beginning of chr 1) . Do the authors have an interpretation as to why the dominance is skewed towards CC in all accessions except BMJM ?

Line 750 : *C. canephora* has one of the widest ranges : here it is not clear to what metrics « range » refers to ? Is it a range of geographical distribution?

Supplementary materials :

The text and figures are very well organized and easy to read. The supplementary material is very useful to help understanding the main text. Unfortunately, there is no paragraph on the detection of biosynthetic gene clusters, just a link to the plantimash web server. It would be useful to have a section describing the gene clusters identified and displayed in table S31.

Figure S27 : a) is missing

Figure S31 : There are some peaks in the « 2:2 » zone for *C. eugenioides* DA56 (beginning of chr1 , and chr9) : could the authors comment on those ? What could they correspond to ? (maybe differences between the accession in the reference and DA56 ?)

Figure S55 is truncated

Author Rebuttal, first revision:

Reviewers' Comments:

Reviewer #1:

Remarks to the Author:

In this revision, the authors attempted to improve the quality of sequence assemblies as well as data interpretation and presentation. However, the revision still suffered from conceptual confusion, inaccurate description of data, poor quality of most figure panels, and lack of appeal to broader readership in polyploid genetics and genome evolution.

(1) The introduction is solely focused on coffee and has no discussion about its broader impact on polyploid genetics and genome evolution.

Response: Thank you very much for the suggestion; we have now added a brief paragraph on polyploid genetics and post-polyploid genome evolution to the Introduction, including references to recent work. We trust that this will make the paper more broadly accessible, and better alert the community to aspects of the paper's content.

(2) Completeness of the genomes: Despite the genome assembly was improved, they failed to explain why the tetraploid genome is substantially smaller than the extant diploid progenitor-related species, as they stated that the two subgenomes are highly conserved and maintained. Even with improved PacBio HiFi and Hi-C approaches, the CA (HiFi) genome (1,198 Mb) is ~90% of predicted from the sum of the two (Table1).

Response: We do understand that genome size evolution in polyploid organisms can be complicated, and therefore some clarification is required:

First, concerning technical aspects, our assembly sizes are in fact very much in agreement with estimates from flow cytometry measurements. Indeed, our *C. canephora*, *C. eugenioides*, and *C. arabica* HiFi genome assemblies cover >93% of the projected genome sizes for each of the three species (Table 1). Due to the strong concordance between the cytological predictions and our assemblies, genome size in *C. arabica* is very unlikely to stem from an assembly artefact.

We can find no clear biological rationale for why the Arabica genome should equal the *C. canephora* plus *C. eugenioides* extant genomes in size. First, the genome sizes of the original diploid progenitors are unknown; only extant representatives of these species have been

sequenced, and moreover, based on flow cytometry results in Noirot et al. (<https://doi.org/10.1093/aob/mcg183>) there appears to be considerable within-species variation in genome size that we have no clear way to attribute to the actual Arabica ancestral individuals. For example, in Noirot et al. it is reported that five individual plants from four different *C. canephora* populations yielded genome sizes spanning a large range between 662 and 760Mb. Such variation in genome sizes is also found in smaller genomes, such as *Arabidopsis* (see, e.g., Long et al. 2013, <https://doi.org/10.1038/ng.2678>).

Second, and perhaps most important, the genome size of an allopolyploid species is only rarely the sum of the genome sizes of its progenitors, since the genome becomes fractionated -often very rapidly - following a polyploidy event, by undergoing deletion of redundant or non-essential genome fragments. Since *Coffea arabica* formed as an allotetraploid hybrid 400-600 ky ago, its genome has been evolving independently of the diploid progenitors since then, and moreover, extant representatives of these progenitor genomes have also continued to evolve. Since the two progenitors are closely related and share high levels of genomic synteny (they diverged from each other only around 6 million years ago) there was initially a high amount of redundant content in the new allopolyploid that could be easily lost without reducing the fitness of the polyploid hybrid. Our comparisons found gene orders in the Arabica subgenomes to be highly conserved, but it is important to note that many genome fragments (including many consisting of several genes and intergenic regions) have been excised since the polyploidy event (Figure 1b), which naturally resulted in reduction of genome size in the hybrid species. This fractionation is also reflected in the BUSCO genes, of which around 5% have reverted to the diploid state (**Table 1**). The fractionation process has been observed in other (allo)polyploid species as well, and it can be surprisingly rapid. For example, *Arabidopsis suecica* is a very recent hybrid (it formed only around 16,000 years ago), but its genome size is around 307 Mb (Burns et al. 2021), i.e. 24 Mb shorter than the sum of its progenitor genomes (~135Mb for modern *A. thaliana* and 196 Mb for modern *A. arenosa* (Lysak et al. 2009; <https://doi.org/10.1093/molbev/msn223>).

By the way, Table 1 seems to indicate only the CA genome (CA-HiFi) was improved, while CC and CE genomes remained unchanged. Table 1 should be simplified with fewer sequence statistics, excluding unimproved CA. The statistics of previously published genomes should be eliminated.

Response: That is correct, we only sequenced *C. arabica* a second time using PacBio HiFi. We have removed the statistics of the previously published assemblies and provide a summary consisting only of the assemblies presented in this paper.

The usage of terms is still confusing in the manuscript. The definition should be clearly stated in the text using *C. canephora* (CC, $2n = 2x = 22$), *C. eugenioides* (EE, $2n = 2x = 22$), and (CCEE, $2n = 4x = 44$), respectively.

Response: Thank you for this comment; we have now explained the notation in the introduction, as suggested.

(3) Their response to the comment on allopolyploidization is simply incorrect. Maize is a diploidized tetraploid (paleopolyploid), while coffee, cotton, and oilseed rape are clearly

allotetraploid. Maize does not have subgenomes in terms of cytology and homoeologous chromosome pairing. In addition, they cited some work on *Arabidopsis* interspecific hybrids, which are not allotetraploids; there is a plethora of work in *Arabidopsis* allopolyploids for them to cite and discuss. Their discussions and reference citations on allopolyploid subjects are inaccurate. As noted, the group lacks expertise in polyploid genomics and genome evolution.

Response: We have removed the references to maize, as the origin of its polyploidy, which even if inconclusive, is not directly relevant for the current paper. We have instead added a reference to the genome assembly and analysis of *Arabidopsis suecica* (Burns et al. 2021; <https://doi.org/10.1038/s41559-021-01525-w>), this allopolyploid genome does not demonstrate genomic shock but rather a gradual adaptation to increased ploidy levels. In separate work, (Göbel et al. 2018; <https://doi.org/10.1093/gbe/evy095>) indeed looked at interspecific hybrids that were not allopolyploids. That work is relevant for the context of genomic shock, however, since the phenomenon has been suggested to result from genomic conflicts resulting from interspecific hybridisation, with genome duplication providing a rescue by restoring the fertility in the hybrid (eg Abbott & Lowe 2004; <https://doi.org/10.1111/j.1095-8312.2004.00333.x>). As such, Göbel et al. have demonstrated previously that interspecific hybrids are an important model, independent of the polyploidy mode.

(4) Lines 182-193: TE results did not support rapid genomic reshuffling or homoeologous recombination. If they speculated a similarity to the rapid genomic reshuffling, as observed in Brassica or Tragopogon, they should provide cytological evidence. Moreover, this statement is contradictory to the results of sequence assemblies, while the subgenomes were highly conserved relative to their extant donor-related genomes.

Response: In our analyses, the TE contents in the *C. arabica* subgenomes were found to be intermediate between the ones in the extant representatives of the diploid progenitors (CC and CE). This suggests that there has indeed been some exchange between the subgenomes, otherwise their TE content would be expected to resemble that of the extant representatives more than they do. We also found support for this genomic transfer when analyzing for homoeologous exchanges. On the other hand, as already noted, we did observe conservation in gene order between the two subgenomes (and/or their extant representatives). These results are not in conflict with each other, since gene order may be conserved in the presence of intergenic (TE-rich) homoeologous exchanges. For similar phenomena in other species see eg *A. suecica*, Burns et al 2021; <https://www.nature.com/articles/s41559-021-01525-w> and peanut, Bertoli et al. 2019; <https://www.nature.com/articles/s41588-019-0405-z>).

Since we do not observe genomic rearrangements but rather strong conservation of gene order, the exchanges that have occurred likely involved syntenic homoeologous blocks between the two subgenomes, which would tend to homogenize intergenic DNA content. This is precisely suggestive of homoeologous exchange rather than homoeologous recombination. We agree that “reshuffling” might imply genomic rearrangements in terms of translocations, so we now avoid the term.

(5) Conservation vs. subgenomic fractionalization – confusion and confusion. “Comparison of the Arabica CC and EE subgenomes against...revealed high conservation in terms of chromosome number, centromere position, and number of genes per chromosome.” (lines 206-212) Moreover, there is little evidence to support fractionation of CC or EE subgenome after tetraploidization (Fig. 1B) (lines 227-233). The blocked area (Fig. 1C) is more related to poor assembly in the centromeric than in other regions.

Response: While synteny between the two subgenomes is largely conserved, this gene order similarity does not exclude genome fractionation processes where genes and intergenic regions are lost from one of the subgenomes. Overall genomic synteny is still retained despite losses of genes alternately on the subgenomic halves.

Technically speaking, synteny analyses (such as MCScanX and CoGe SynMap) generally first collapse tandemly duplicated genes into one representative model and then carry out the identification of syntenic blocks with the remaining genes. Therefore, the proportion of conserved syntelogs gets reduced in regions with high tandem duplicate counts when calculated over all gene models. Figure 1B shows the fraction of syntenic duplicates along a chromosome. The genome fractionation observed in the centromeric region (the blocked area) is low precisely due to the presence of tandemly duplicated genes in this region; this decreases the proportion of syntenic genes discoverable. The data on the presence of tandem duplicates in the regions with low proportion vs high proportion (that is, high vs low fractionation rate) of syntenic duplicates is analysed in Table S9, showing that tandem duplicates are significantly enriched in regions with high fractionation rate.

In summary (lines 249-257), as noted in the previous comments, “genome shock” was defined at the EARLY EVENTS of interspecific hybridization (McClintock 1984, Science) or polyploidization (many papers not cited). It has nothing to do with what was observed in *C. arabica*. In addition, there is no evidence of mis-regulation of transcriptome and epigenome ... in rampant activation of TE elements (lines 252-253).

Response: We apologize for any misunderstanding; it was never our intention to suggest a “genome shock” to have occurred in *C. arabica*; indeed, quite to the opposite, we stated that we do not see any evidence of a shock event. The text has been revised to further clarify this point.

(6) Gene-family subgenome expression dominance. This is like the fractionation is more nonsense than evidence. “The global expression patterns did not show significant subgenome expression dominance patterns in different bean developmental stages...” (lines 267-269). The genes involved in three pathways were hand-picked and did not show any expression dominance. For example, homoeolog expression difference was observed in only 1 out of 5 NMT gene family members, with similar homoeolog expression patterns for all other family members (Fig. 2B). Moreover, the description of subCC or subEE expression dominance in the text is incorrect when an individual gene or family member was concerned. It should be expression difference between CC and EE homoeologs, instead of subCC and subEE genomes. Fig. 2C is useless and should be removed. Again, labels in Fig. 2A and Fig. 2B are barely visible. They have not fixed the problem of poor quality in data presentation.

Response: In the revised version we first report the results from a global analysis of subgenome expression dominance using the new reference genome; lines 290-292 provide results from that analysis (together with tables S15-S16). The global analysis was carried out in a similar manner as in Bird et al. 2021, in a case where both diploid progenitors and the allopolyploid reference genome were available. Briefly, we first identified homoeologs (=syntelogs) between *C. canephora* and *C. eugenioides*. After that, we identified the syntelogs in the subgenomes: subCC in case of *C. canephora* and subEE in case of *C. eugenioides*. Finally, we removed the syntelogs where the expression bias could possibly be explained by homoeologous exchange (accomplished in a separate analysis). Altogether, we identified 10,281 syntelogs that were shared across the three species that presented no evidence for having undergone homoeologous exchange. For the global transcriptome analyses we then mapped the *C. arabica* transcriptome data to the *C. arabica* HiFi assembly and compared the expression of the above 10,281 syntelogs.

After this global analysis, which did not show any statistically significant subgenome expression dominance, we “hand-picked” examples of the three major gene families that are well-known to contribute to coffee cup quality. Besides illustrating the overall patterns of gene expression, these three families also illustrate the differences between the genomes in terms of tandem duplicate content. Expression patterns among these duplicates are in many cases suggestive of sub-/neo-functionalization of genes retained on alternate subgenomes. As such, although clearly selected *ad hoc*, these gene families well represent the sort of mosaic subgenome-wise transcriptional change seen at the gene family level within a number of polyploid species, for example in wheat (Pfeifer et al. 2014; <https://www.science.org/doi/10.1126/science.1250091>) and rapeseed (Chalhoub et al. 2014; <https://www.science.org/doi/10.1126/science.1253435>). Moreover, the three gene families display this mosaic transcriptional pattern to greater and lesser extents, making the examples valuable from a comparative aspect. We therefore believe our focused analyses to be of both general polyploid molecular evolutionary interest as well as providing data that may be informative of the genome differences between coffee species resulting in different cup qualities.

We increased the font size of the labels in Figs 2A and 2B and the gene names in the phylogenetic trees; we also adjusted the expression color palette for better visibility. Regarding Fig. 2C, we believe it is useful for visualizing the differential expression of other members of the same gene families that do not contribute to cup quality, better illustrating the extent of transcriptional mosaicism. If needed, this panel could be moved to the supplementary. We leave the decision to the editor.

(7) Homoeologous exchanges (HE) were rare and did not contribute to the origin and domestication of *C. arabica* (lines 347-371). Why did they want to twist their arms and emphasize this?

Response: On the contrary, we in fact found thousands of genes that were affected by homoeologous exchange (HE) (as shown in Supplementary figures S33-S35, see S36 for a summary). There was only one region where the HE is *fixed* towards EE subgenome, but our analysis identified numerous regions with 3:1 allele ratio, suggesting CC to have had a dominating role in this process, see Supplementary section 5.

The high similarity of the HE patterns in different cultivars and wild accessions suggests that most of the events occurred in a common ancestor, likely close to the time when the stable tetraploid was established. Interestingly, we also identified one cultivar (BMJM) that shows a large HE event at the end of chromosome 1, with a bias towards EE. This suggests that HE events still take place, and at least some of them do not decrease fitness for coffee production. This finding is important, since it suggests that in allopolyploid cultivar species with low genome diversity (=low effective population size), HE might be one way of promoting phenotypic diversity. While based on the present data this remains a hypothesis, the topic has been studied in allotetraploid rice (Wu et al. 2020; <https://academic.oup.com/nsr/article/8/5/nwaa277/5960157>). We have clarified this part of the text now; please also see our reply to Referee #3 on the question about HE.

(8) Population history: There is more speculation than evidence (lines 406-423), as they also noted at the end of discussion. The figure legends in Fig. 3 are poorly constructed. For example, the legends in Fig. 3F should reflect the content of the figure panel but not the conclusion of the data. This problem exists across all figure legends.

Response: Population genetics models are powerful tools that provide objective views into population history. The various methods that have been developed have limitations with respect to predicting timing of events, as they require good estimates of the mutation rate as well as generation time. When those are known, as in the case of the human model system, population genomic methods are well able to reconstruct past history to highly accurate detail.

However, mutation rate and generation time parameters only affect the timing, not the overall trajectory of historical changes of population sizes. For *C. arabica*, the different methods concordantly show several historical bottlenecks and admixture of two populations that have been isolated for a considerable time. This admixture has apparently contributed towards the development of cultivated Arabica lineages. The fact that the older cultivar accessions have a higher level of admixture than their offspring well shows the effect of breeding cycles and confirms the hypothesis that high-yielding cultivars in general have experienced some form of demographic change event that increased overall genomic heterozygosity during their history.

We have clarified the text to point out the limitations of the models: “While demographic modeling provides accurate estimates of the changes in historical population sizes, the timing of the events should be treated with caution since there is considerable uncertainty in mutation rate, factors contributing to it, and generation time estimates in plants. In the case of *C. arabica*, the generation time was estimated from empirical data, but the precise mutation rate is not known.”

(9) Kinship estimation (Fig. 4). The diagrams and legends are so confusing by referencing several different supplemental figures with multiple-coloured backgrounds that fail to help understand the results and discussions in the text. For thumbnail images...what are the x-axis and y-axis? What is the second- and third-degree relationship? The overall presentation is too long with too many speculations through the analysis of selected genes.

Response: The figure summarizes several analyses on kinship and admixture, where each of the results are reported in greater detail in their own supplementary figures or tables. We

have added an explanation for the x and y axes in the thumbnails, together with a reference to the supplementary figures that show these results in detail, as well a description of the different color backgrounds for the wild accessions. Our F3 results therefore agree well with ADMIXTURE analysis, PCA, as well as the relatedness analysis with KING.

The degree of relationship follows the standard terminology from genealogy and describes how related the individuals are. An explanation was added to the figure legend.

(10) Introgression (Fig. 5). The figure cannot help substantiating the results and discussion, and the legends lack clarity of elements present in the figure. For example, which dot/colour represents which cultivar or accession/pair analysed (Fig. 5A)? The analysis of TEs (Fig. 5B) is incomprehensive. Colour scale is missing, and the label of "16" (top of the bottom panel) should be an error. Again, the labels and fonts are barely legible.

Response: Thank you very much for pointing out these inconsistencies; we have replaced the "16" with the proper cultivar ID and increased font sizes. We also made a new panel B that better illustrates multiple evidence pointing towards introgression and its possible contribution to coffee leaf rust resistance. We also clarified the explanations in the plots and descriptions of the axes and colors. Regarding the analysis of TEs, we agree, panel 5B cannot comprehensively illustrate the analysis carried out; however, deeper explanation in the main text would significantly prolong the manuscript. Therefore, we now refer interested readers to Supplementary section 7.

We have pasted all the figures as vector graphics (as PDF) in the manuscript, which should improve readability. However, the included Powerpoint should be the main source for reviewing the figures in high resolution.

(11) The manuscript can be improved with language skills and technical issues. For example, Fig. S29 appeared after Fig. S1; no other Fig. S was found in between. Some citations are confusing such as Figs. 3B, S42, etc. Another bigger issue is about absent or mis-citation of most, if not all, original literatures related to stable allopolyploidy, homoeolog expression dominance, and epigenomic diversification. There is no data to speculate epigenetic regulation. This along with genomic studies in other allopolyploid species can become a paragraph in the Introduction.

Response: The jump in the running number from S1 to S25 happened because our manuscript style involved referencing entire sections in the supplementary material, including all figures inside those sections. This was done in order to reduce the amount of referencing needed in the main text, where the thinking was that the reader would need to read through the supplementary section with included figures to understand the analysis that was carried out. In some cases, however, the discussion in the main text required reference only to a specific figure within a supplementary section. We can revert back to a version where all the figures are referenced in the main text, but we leave this final decision to the editor.

We added to the introduction a paragraph on allopolyploidy, as suggested. Regarding epigenomic diversification, due to the lack of subgenome expression dominance it is unlikely

to play a major role in *C. arabica* subgenome expression, and there is currently no methylation data on *C. arabica*.

In sum, the authors have attempted to improve the data presentation and interpretation. The revision has retained some old problems such as lack of significance and advance in polyploid genomics and evolution (incorrect interpretation and references of genome shock, subfractionation, and genome expression dominance) and technical validity (poor qualities of data and figures and incomprehensible legends). As a result, the revised manuscript does not meet the standards for publication in this journal.

Response: Coffee is among the most traded products in the world and its origin and history has naturally attracted a lot of research. Our results confirm some of these historical hypotheses and reject others (which we do not discuss in the paper). We presented high quality genomes for *Coffea canephora*, *Coffea arabica* and *Coffea eugenioides*, and carried out whole genome sequencing of a set of cultivars that represent roughly 95% of global Arabica coffee production. In addition to these valuable resources, our research showed how the genome of *C. arabica* has evolved both before and after the founding allopolyploidy event, and further population genetics modelling showed historical fluctuations in *C. arabica* population since the establishment of the species, and how this population history has contributed to the genetic diversity of current cultivated Arabica lineages.

We also contend that our results have broader general impact on plant breeding, as they show that the reduction of the complexity of resistance genes is one of the first processes affected in breeding, while the results on new hybrid Timor hybrid lineages provide genomic insight into how this reduction has been compensated for through introgression from other species, followed by back-crossing to improve cultivar properties. Furthermore, our analysis on homoeologous exchange provides evidence for how phenotypic variety may be generated in cultivars with limited genetic variation, and sheds light on the early stages of polyploid formation in Arabica.

We added to the introduction a paragraph on allopolyploidy, as suggested, and went to great lengths citing many original papers and reviews (refs 1, 2, 4, 5, 6, 8, 9, 10, 11, 13, 14, 16, 17, 18, 20, 21, 38, 39, 40, 43, 44, 47, 48, 49, 50, 51, among others). If the reviewer feels that there is a key reference that was omitted, we'll be happy to add it.

Reviewer #2:

Remarks to the Author:

The authors have addressed my and referee #1's main concern, which is the subpar assembly, and have given sensible names to the chromosomes. While the assembly of coffee has been improved, the authors had to make some efforts to obtain this genome at the current level, resorting to gene-base sorting, likely given that hifi sequencing reads were only provided at 10x, which is low for hifi assemblies. Similarly, as an answer to reviewer #3, the authors mention limited resources, which, given the advent of the Pacbio Revio, should not have been an issue but of course would have needed even more reanalyses. Now, in any case, the genome has improved substantially over the last version and is much more usable and better than the one from Aldenwinkle, both in assembly metrics and in gene completeness, including

a sensible (high) number duplicated BUSCO values, making it the best assembly at the time - until the mentioned improved one by Aldenwinkle comes out.

Response: Thank you very much for your suggestions in the previous review; HiFi technology indeed helped us to obtain a much higher quality assembly, partly because of the high accuracy of the reads, which results in a much lower coverage needed in the assembly process. We would like to correct that for the assembly we actually had 44.95Gb of PacBio HiFi data, which means 33.3x coverage. This is within the 22x-40x range giving optimal results with HiFiiasm (<https://arxiv.org/pdf/2306.03399.pdf>; https://link.springer.com/protocol/10.1007/978-1-0716-2561-3_23)

PacBio Revio was not yet available when we carried out the sequencing, but except for the reduction in sequencing costs it would not have generated data of higher quality or increased read length compared to the Sequel IIe that we used. Since we limited the complexity of the assembly task by sequencing a di-haploid lineage, we believe we may have a better assembly than Aldenwinkle et al., especially if we continue upgrading it.

While I completely disagree with not sharing data, the authors pledge to share data via NCBI upon publication and have uploaded it to solgenomics, which, for historical reasons, has supported the coffee community.

However, I still can't find either the genome or the vcf data in any of these resources, and I find it a bit odd that I should try out all folders in the solgenomics ftp site. It was for sure not in coffee, as data was from 2005, nor in genomes *Coffea canephora* or *coffea humbotiana*, but I trust Dr. Muller will make it available. In addition, when I searched on COGE with the keywords *Coffea*, coffee, or arabica, I was only able to retrieve GCA_003713225. Coffeabase was password protected, and the only mention of PRJNA698600 I found was in another coffee manuscript. Of course, it would have made it easier to actually check any claims made by the authors if only a reviewer link (which NCBI provides for this reason) would have been provided.

Response: We apologize for the current lack of publicly available data; we are of course committed to sharing all data and results. The old solgenomics ftp site was phased out due to security issues and the new site was still under construction when we submitted the revision, and apparently the construction is still ongoing.

Data is now publicly available at:

https://bioinformatics.psb.ugent.be/gdb/coffea_arabica/

The genome can also be viewed publicly at ORCAE:

<https://bioinformatics.psb.ugent.be/orcae/overview/Coara>

We have also granted public access to genome assemblies in CoGe, the IDs are

C. arabica: 66663 (Pacbio HiFi), 53628 (Pacbio)

C. canephora: 50947

C. eugenioides: 60235

All reads and assembly have been deposited in NCBI (bioproject ID: PRJNA698600) and will be made publicly available upon acceptance of the paper. The data at solgenomics ftp will be accessible as well when the revised site is fully operational.

Regarding my comment with Blast2GO, the authors themselves show a picture of what Blast2GO is - it is much more than just a "BLAST" to Arabidopsis but can include profile scans and also does some computation of results - like any of the modern (plant) functional annotation pipelines, which consider different evidence chains. What is worrying for me is that none of these require much effort and indeed can be used as web services as well. Additionally, I want to point out that, of course, we are not in 2000 anymore, and most plant research is done on crops by now, so we have a lot of experimental evidence for crop species. Since I couldn't obtain the annotation files, I couldn't provide the authors with autolinks to some annotated resources.

Response: We have now carried out functional annotation using Interproscan, which is the tool the commercial software BLAST2GO uses in the background. The annotation is provided as a tab-separated text file in https://bioinformatics.psb.ugent.be/gdb/coffea_arabica/C.arabica_Interpro.tar.gz

Since Arabidopsis is an intensively studied model species, its annotations and gene functional descriptions are constantly updated, which is why we preferred to carry out a BLAST-based homolog identification. For the same reasons we believe that these two approaches will end up yielding largely similar results, as the best matching Arabidopsis gene is going to have the annotation information from Interproscan. The benefit in our approach is that the best match Arabidopsis ID also provides a starting point for finding molecular work carried out on the specific homolog. However, no comparisons of the different approaches was carried out, and while this may be important for transferring functional annotations across species, we have decided that such comparison is outside of the scope of the current work. As our publicly available genomic annotations are improved over time, that can hopefully be addressed.

Finally, the figures are still not of decent quality as they contain wiggly red lines under names - this is really easy to fix, so why this wasn't done I don't quite get.

I would expect some care to quality given the outlet.

Response: We have fixed that issue now. This was due to spell checking used in MS Powerpoint that was used for compiling the final figures, of which we included snapshots in the submitted file. High resolution images were also separately provided as a Powerpoint file, but may have not been provided to the reviewers.

Reviewer #3:

Remarks to the Author:

The article presents a new version of the previously submitted one, with an improvement of the Coffea arabica assembly using HiFi reads, as well as additional analyses, in particular the identification of homeologous exchanges. The authors addressed all the points previously raised by the referee and made the manuscript easier to read. The article provides interesting results regarding homeologous exchanges in tetraploid coffee, as well as evidence for gene

fractionation through excision, and description of a mosaic pattern of subgenome expression dominance. In addition, the resequencing of 39 wild and cultivated *Arabica* accessions allowed to an in-depth analysis of the breeding history and dissemination routes of *C. arabica*.

Below are a few questions to address:

1/ The number of genes annotated is much higher on the new HiFi assembly than on the previous assembly, and each subgenome contains several thousands more genes than the progenitor genomes. How can the author explain this finding: were there genes missed in other annotations (maybe tandemly repeated genes, less correctly inferred in more fractionated assemblies), or is there an over prediction in *C. arabica* (HiFi) ? Can it be caused by the annotation transfer performed with GeMoMa ? Also, in table S5, some new models do not correspond to old models, but if the new models were obtained by transfer of old annotation, how can there be more?

Response: GeMoMa is a tool for *de novo* annotation where the annotations from a closely related species are used as reference. In the transfer process, a gene model may find several matches in the new genome rather than just one. Therefore the mapping is not one-to-one but rather one-to-many (or none). In the case of *C. arabica*, the new HiFi assembly is 110 Mb larger and much more contiguous; therefore, the gene content is increased in terms of tandem duplicates as well as highly similar syntelogs between CC and EE subgenomes, as the referee suspected. Since there is no direct one-to-one mapping between the gene models, some genes do not have a corresponding gene model in table S5.

2/ line 285 ; « at decreased level » (add « compared to subCC » ?)

Response: Thank you; this now added.

3/ Line 301 : « seed-type FAD2 gene expression was dominated by subCC » . This statement is not clear from figure: gene « Cara002g032150 » seems the most expressed (subEE), can the authors be more precise ?

Response: The motif in Cara002g032150 is of the type YKNNF, which does not comply with the classic C-terminal ER retrieval signal requiring two positively charged residues; this could suggest a lowered effectiveness in ER retrieval. We have revised the text to:

“Overall, the duplicates show expression patterns suggesting dominance by subCC. Interestingly, the localization signal of the one highly expressed subEE homolog (Cara002g032150) differed from the other seed-type paralogs.”

4/Line 355 : surprisingly all but one accession showed significant 3:1 towards subCC.

Supp data & Figure S36 : BMJM has a significant bias towards subEE (specific subEE region in the beginning of chr 1) . Do the authors have an interpretation as to why the dominance is skewed towards CC in all accessions except BMJM ?

Response: Thank you very much for this excellent question. In cultivar BMJM, chromosome 1 had a large homoeologous transfer block in the end, explaining the discrepancy. As for the similar skew pattern in other accessions, it had to have originated from a common ancestor. The most likely explanation is that it originates from the time when the polyploid line was established. We have now revised the text to:

“Surprisingly all but one accession (BMJMI) showed significant (p-values $<<9.8e-37$; Chi-square test) 3:1 allele bias towards subCC. Patterns were present in both wild and cultivated Arabicas, suggesting that the allele bias is an adaptive trait and not associated with breeding, and the similarity of the patterns further suggests the biases to originate from a common ancestor, possibly from the establishment of a stable tetraploid subsequent to the initial polyploid event. The bias in BMJM cultivar towards subEE was due to a single crossover in chromosome 1. The site frequency spectrum displayed a strong bias towards recent HE as well as events shared by all individuals (Fig. S38), suggesting that HE events are under strong selection. As demonstrated in BMJM, the rare HE events were also present in cultivars, suggesting that breeding does not select against these events. Altogether, the results suggest that in polyploid species with low genetic diversity, HE could be one possible cause for phenotypic variation observed in closely related accessions.”

Line 750 : *C. canephora* has one of the widest ranges : here it is not clear to what metrics « range » refers to ? Is it a range of geographical distribution?

Response: Yes, we mean geographical distribution, and we now clarify this in the text: “Since *C. canephora* has one of the widest geographic ranges”

Supplementary materials :

The text and figures are very well organized and easy to read. The supplementary material is very useful to help understanding the main text. Unfortunately, there is no paragraph on the detection of biosynthetic gene clusters, just a link to the plantimash web server. It would be useful to have a section describing the gene clusters identified and displayed in table S31.

Response: Thank you; such a section is now added.

Figure S27 : a) is missing

Response: Thank you; this is now corrected.

Figure S31 : There are some peaks in the « 2:2 » zone for *C. eugenioides* DA56 (beginning of chr1 , and chr9) : could the authors comment on those ? What could they correspond to ? (maybe differences between the accession in the reference and DA56 ?)

Response: Indeed, this was an interesting finding. It could represent an introgression to DA56 from *C. canephora* or be due to ancestral polymorphism (incomplete lineage sorting), but distinguishing among these and other hypotheses would require further analysis and preferably more *C. eugenioides* and *C. canephora* accessions. We now wrote about these ideas in the supplementary (Section 5) but did not raise it to the main paper, as the text is already extensive.

Figure S55 is truncated

Response: Thank you; this is now corrected.

Decision Letter, second revision:

30th Oct 2023

Dear Professor Albert,

Your Article, "The genome and population genomics of allopolyploid *Coffea arabica* reveal the diversification history of modern coffee cultivars" has now been seen by 2 referees. You will see from their comments below that while they find your work of interest, some important points are raised. We are interested in the possibility of publishing your study in Nature Genetics, but would like to consider your response to these concerns in the form of a revised manuscript before we make a final decision on publication.

We therefore invite you to revise your manuscript taking into account all reviewer and editor comments. Please highlight all changes in the manuscript text file. At this stage we will need you to upload a copy of the manuscript in MS Word .docx or similar editable format.

*2) If you have not done so already please begin to revise your manuscript so that it conforms to our Article format instructions, available here.

*3) Include a revised version of any required Reporting Summary:

Please be aware of our guidelines on digital image standards.

[redacted]

We hope to receive your revised manuscript within four to eight weeks. If you cannot send it within

this time, please let us know.

Sincerely,
Wei

Wei Li, PhD
Senior Editor
Nature Genetics
New York, NY 10004, USA
www.nature.com/ng

Reviewers' Comments:

Reviewer #1:

Remarks to the Author:

A contribution of the report is to updating allotetraploid coffee (from Denoeud et al., 2014, Science) with high-quality sequences including extant diploids and resequencing data. Although the authors made attempts to improve their presentation, the 2nd revision failed to clarify new insights of coffee genomes into the polyploid genomics and genome evolution. Their responses have not satisfied the major concerns about conceptual confusion, inaccurate statements, and poor quality of data presentation. Some examples are as follows.

In the first round of reviews, they confused paleopolyploid maize with allotetraploid coffee and insisted on their relevance (genome fractionation). They finally corrected this conceptual confusion in this version.

In the prior review, there was a major comment on subgenome dominance because the expression data did not support the conclusion. In their 2nd response, "After this global analysis, which did not show any statistically significant subgenome expression dominance, we 'hand-picked' examples of the three major gene families that are well-known to contribute to coffee cup quality." It's amazing that they insisted the wrong conclusion (section title, 2 pages of description, and a figure) using the "hand-picked" data points!

Genome size variation: the likely explanation is unknown progenitor(s), as seen in other allopolyploid

crops such as cotton (Wendel 1989, PNAS), etc. In their response, they cited the reference of <https://doi.org/10.1038/ng.2678> to show size variation within species (*A. thaliana*, a paleopolyploid), and used an outdated fragmented sequence of *A. suecica* (Lysak et al. 2009; [hKps://doi.org/10.1093/molbev/msn223](https://doi.org/10.1093/molbev/msn223)) to show size variation in an allotetraploid. For the latter, the two groups recently updated the sequences of *A. suecica* using different approaches in *Nature Ecology & Evolution* (2021). Similar ill-perceived responses also occur in other comments such as on “genome shock,” “genome fractionation,” “homoeologous exchanges,” etc. All in all, they failed to fully understand and clarify those comments.

Minor:

Lines 125-126: Replace “subgenome C” with “CC,” and “subgenome E” with “EE.”

Line 165: *C. eugenoides* (CE) should be (EE).

Figure 5. The revised Fig. 5B included nucleotide diversity (top), fixation index (middle), and expression patterns (bottom). Unfortunately, these data did not correlate with each other or support a meaningful conclusion. These data panels were put together by convenience (from bioinformatic pipelines) instead of biological relevance (conclusion). This type of approaches was used throughout the manuscript including Figure 2 and related statements.

In the Conclusion (lines 783-391), they stated, “Analysis of repetitive elements did not suggest a genomic shock, but in contrast, a higher LTR turnover rate in CA; this mechanism could possibly originate from CE, since CC demonstrates elevated numbers of LTRs when compared to other sequenced *Coffea* species. No evidence of genomic shock was observed in genome fractionation analyses either, since fractionation rates remained unaltered before and after the allopolyploidy event. Likewise, gene expression analyses showed no global subgenome dominance, but...similar to what has been observed in other neopolyploid crops such as rapeseed11 and cotton50.

There are several problems. Firstly, if the conclusions are correct, these coffee allotetraploid sequences do not provide new genomic insights. Secondly, the overall conclusion was somewhat contradictory, whereas in the Results the authors tried to argue the impact of fractionation, HE, etc. Thirdly, cotton polyploidization occurred 1-1.5 million years ago (Wendel 1989, PNAS; Wendel & Cronn 2003, *Advances in Agronomy* 78:139-186; Chen et al. 2020, *Nature Genetics*), which is certainly not a neopolyploid. Again, this reflects a lack in the understanding of polyploid genomics and genome evolution.

Finally, the manuscript is way too long (single-spaced, 18 pages!). It should be substantially reduced to focus on new findings and insights, perhaps following these suggestions.

Introduction: Lines 132-148: reduce this paragraph to a couple of sentences and move it as the last part of the previous paragraph.

Results and Discussion.

1. Combine “Chromosome-level assemblies...” and “Strong Conservation...” into one section and reduce the context (especially lengthy description of sequencing statistics) in half.

2. Remove Figure 2 (poor quality and did not support the conclusion) and reduce the section of “Gene-specific subgenome expression dominance” (a few sentences and reference citations) simply because the genome-wide data did not support the conclusion. These genes are “hand-picked” and moreover, the data did not support “expression dominance.”

3. Origin and domestication (4 pages!): the section should be reduced by two thirds. Many are tedious

and redundant in literature (Lashermes et al. 1999, MGG; Cenci et al. 2012, Plant Mol. Biol.; Denoeud et al. 2014, Science; Scalabrin et al. 2020, Sci. Rep.;) and do not provide any new insights.

4. Kinship (3 pages!): reduced to half, some of which are redundant to the section of "origin and domestication." (also see above)

5. Introgression: Reduce it at least one third and combine the last (TE activity) into this section.

Reviewer #2:

None

Reviewer #3:

Remarks to the Author:

The updated version of the manuscript addresses the comments made by the reviewers. The article, although mainly focused on population studies, also provides a comprehensive overview of events that can take place after allotetraploidization (TE invasions, homoeologous exchanges, gene fractionation, biased gene expression) and shows which ones arose (or not) in *Coffea arabica*.

Below are some last comments:

About homoeologous gene expression in the caffeine biosynthesis pathway, the authors should cite a recent study (doi.org/10.1093/aob/mcac041) and compare their results to that study (they seem to be very similar).

Also, the following article is describing some homoeologous exchanges in *C. arabica* ([doi: 10.1534/g3.116.030858](https://doi.org/10.1534/g3.116.030858)). Here again, it would be useful to compare the events identified to the previously identified ones.

Lines 112-116: the sentence is confusing and needs to be reformulated. What is "compatibility of the parental species", how does it affect genomic shock (define?) and subgenome dominance? How does it oppose "gradual adaptation to a new ploidy level"? if I understood correctly, the authors mean that either there are drastic changes right after the polyploidization, or no "shock" and slow return to a diploidized state. It should be written more clearly.

Lines 381-385. The authors write that the pattern similarity suggests the biases originated in a common ancestor of all sampled accessions. But the next sentences state that there is a strong bias towards recent HE as well as to events shared by all individuals, suggesting that HE events are under strong selection. Is it possible that the pattern similarity is actually caused by selection rather than a common ancestor? How to distinguish between the two hypotheses?

Additionally, the HE events are not easy to decipher from figures S31-36. It would be useful to provide a synthetic table, listing all detected events (with their positions) for all the accessions studied (data underlying figure S38?).

Author Rebuttal, second revision:

[REDACTED]

[REDACTED]

[REDACTED]

Reviewer #1:

R1. A contribution of the report is to updating allotetraploid coffee (from Denoeud et al., 2014, Science) with high-quality sequences including extant diploids and resequencing data. Although the authors made attempts to improve their presentation, the 2nd revision failed to clarify new insights of coffee genomes into the polyploid genomics and genome evolution.

A1. In Denoeud et al. (2014) we did not present the allotetraploid *Coffea arabica* genome, as the reviewer incorrectly states, but instead a short-read, poorly contiguous assembly of diploid *Coffea canephora*, one of *C. arabica*'s progenitor species. In this paper, we present highly contiguous, long-read assemblies of *C. arabica*, and of its two progenitors: the same *C. canephora* individual sequenced in Denoeud et al., and Arabica's second diploid progenitor, *C. eugenioides*.

R2. Their responses have not satisfied the major concerns about conceptual confusion, inaccurate statements, and poor quality of data presentation.

A2. Constructive criticism is always welcome when it is substantiated by data and/or suggestions on how to improve the manuscript. We would like to point out that Reviewer #2 had no further comment, while Reviewer #3 stated "*The updated version of the manuscript addresses the comments made by the reviewers.*".

R3. Some examples are as follows. In the first round of reviews, they confused paleopolyploid maize with allotetraploid coffee and insisted on their relevance (genome fractionation). They finally corrected this conceptual confusion in this version.

A3. We are happy that the reviewer acknowledges that the problem is now solved.

R4. In the prior review, there was a major comment on subgenome dominance because the expression data did not support the conclusion. In their 2nd response, "After this global analysis, which did not show any statistically significant subgenome expression dominance, we 'hand-picked' examples of the three major gene families that are well-known to contribute to coffee cup quality." It's amazing that they insisted the wrong conclusion (section title, 2 pages of description, and a figure) using the "hand-picked" data points!

A4. It was Reviewer 1, not us, who used in their previous review the term "hand-picking". In our response, we simply iterated that term within quotation marks. We note that the term "hand-picking" could imply that we chose exotic gene families to satisfy some predestined outcome, thus putting into question our intellectual honesty. We apologize if we

misinterpreted the reviewer's intent, however; the *NMT*, *TPS*, and *FAD2* gene families were not "hand-picked": they are broadly recognized as major contributors to coffee cup quality, and thus their subgenome dominance patterns would be of clear interest to the coffee community. If Reviewer 1 did not agree with what they define as "hand-picking", they could have suggested other gene families, but they did not.

Second, the hypothesis we tested was whether homologs in one subgenome consistently demonstrate higher expression values than their syntelog counterparts in the other. We statistically tested this dominance in two different ways, and in both cases, the null hypothesis (no expression difference between subgenomes) could not be ruled out. Additionally, the tests were about the distribution of dominance, not on individual syntelog pair values. As such the results do not at all mean that there are no individual syntelog pairs with clearly differing expression values; in fact, we uncovered a rich, mosaic pattern wherein for some genes (and consequently their functional processes), only one subgenome plays the dominating role, and other cases where neither subgenome dominates and the two subgenomes work in a concerted manner.

While our paper has been in review, expression bias has also been studied in *Coffea arabica* by Combes et al. (2023), with results entirely consistent with ours. Given that this publication affects the novelty of our data, we moved Fig. 2 to Extended data Fig. 2 .

R5. Genome size variation: the likely explanation is unknown progenitor(s), as seen in other allopolyploid crops such as cotton (Wendel 1989, PNAS), etc. In their response, they cited the reference of <https://doi.org/10.1038/ng.2678> to show size variation within species (*A. thaliana*, a paleopolyploid), and used an outdated fragmented sequence of *A. suecica* (Lysak et al. 2009; <https://doi.org/10.1093/molbev/msn223>) to show size variation in an allotetraploid. For the latter, the two groups recently updated the sequences of *A. suecica* using different approaches in Nature Ecology & Evolution (2021).

A5: Reviewer 1 refers to Burns et al, 2021, Nat. Ecol. & Evol., which we had already mentioned both in the manuscript and in our previous response: "*Second, and perhaps most important, the genome size of an allopolyploid species is only rarely the sum of the genome sizes of its progenitors (...)* For example, *Arabidopsis suecica* is a very recent hybrid (it formed only around 16,000 years ago), but its genome size is around 307 Mb (Burns et al. 2021), i.e. 24 Mb shorter than the sum of its progenitor genomes (~135Mb for modern *A. thaliana* and 196 Mb for modern *A. arenosa* (Lysak et al. 2009; <https://doi.org/10.1093/molbev/msn223>)".

The reviewer may not have seen the extended data figure 1f of Burns et al., 2021, in which the *A. suecica* genome size (305 Mb) is a whole 140 Mb shorter than the sum of *A. thaliana* (115 Mb) and *A. arenosa* (330 Mb). In fact, the Burns et al., 2021 paper proves our point "*the genome size of an allopolyploid species is only rarely the sum of the genome sizes of its progenitors*" even more strongly than we had cautiously stated on the basis of the Lysak et al., 2009 paper.

R6. Similar ill-perceived responses also occur in other comments such as on "genome shock," "genome fractionation," "homoeologous exchanges," etc. All in all, they failed to fully understand and clarify those comments.

A6. We strived to find, in the reviewer's present comments, the basis of his assertion, but we weren't able to find anything.

R7. Minor: Lines 125-126: Replace "subgeome C" with "CC," and "subgenome E" with "EE."
Line 165: *C. eugenoides* (CE) should be (EE).

A7. We are sorry for this confusion. We acknowledge that the previous nomenclature was somewhat confusing. Therefore in the revised text we use *Arabica*, *Robusta* (*Coffea canephora*) and *Eugenoides* to designate the plant species, and subCC/subEE to designate the *Arabica* subgenomes derived from *Robusta* and *Eugenoides*, respectively.

R8. Figure 5. The revised Fig. 5B included nucleotide diversity (top), fixation index (middle), and expression patterns (bottom). Unfortunately, these data did not correlate with each other or support a meaningful conclusion. These data panels were put together by convenience (from bioinformatic pipelines) instead of biological relevance (conclusion). This type of approaches was used throughout the manuscript including Figure 2 and related statements.

A8. This comment is not logical to us. It is common to put together analyses stemming from different experiments, or bioinformatic pipelines, to draw a conclusion about the possible function of a gene or genomic region. We believe our presentation of the data is biologically very relevant. The F_{ST} values in the figure illustrate the genome differentiation between the compared groups. High F_{ST} values between introgressed vs. cultivated individuals suggest that variation in this region has been introduced by introgression, and has not been removed in further backcrosses to cultivars. The wild vs cultivars F_{ST} comparison shows that the wild accessions do not differ greatly from the cultivars in these regions, so they cannot be the source of the increased diversity. The nucleotide diversities further illustrate this point. Gene expression data show when such levels increase upon *H. vastatrix* inoculation, thus highlighting differentially expressed genes that become good candidates for resistance loci. We clarify these points in the revised text, for the benefit of a broad audience.

R9. In the Conclusion (lines 783-391), they stated, "Analysis of repetitive elements did not suggest a genomic shock, but in contrast, a higher LTR turnover rate in CA; this mechanism could possibly originate from CE, since CC demonstrates elevated numbers of LTRs when compared to other sequenced *Coffea* species. No evidence of genomic shock was observed in genome fractionation analyses either, since fractionation rates remained unaltered before and after the allopolyploidy event. Likewise, gene expression analyses showed no global subgenome dominance, but...similar to what has been observed in other neopolyploid crops such as rapeseed11 and cotton50. There are several problems. Firstly, if the conclusions are correct, these coffee allotetraploid sequences do not provide new genomic insights. Secondly, the overall conclusion was somewhat contradictory, whereas in the Results the authors tried to argue the impact of fractionation, HE, etc. Thirdly, cotton polyploidization occurred 1-1.5 million years ago (Wendel 1989, PNAS; Wendel & Cronn 2003, *Advances in Agronomy* 78:139-186; Chen et al. 2020, *Nature Genetics*), which is certainly not a neopolyploid. Again, this reflects a lack in the understanding of polyploid genomics and genome evolution.

A9. First, the lack of genome shock and subgenome dominance is definitely a new result for *Coffea arabica*. Both similar and contrasting results have been reported on other polyploid crops. Second, in the Results section we did not “try to argue”; we reported the results on fractionation and HE, which by the way are independent processes, and therefore the results from these analyses can differ. Both processes can occur in a biased (a case of subgenome dominance) or unbiased manner. In the case of fractionation, we did not see a bias, as for each subgenome the genome fragments were discarded at equal proportions. In the case of HE, however, we did see a bias, and that is reported in the text. This is a very interesting result that would not have been captured had we not assembled the genomes of the modern representatives of the diploid progenitors. Third, the time scales with regard to the definition of neopolyploids depend on the biological context. We and others use “neopolyploidy” when referring to an event where the subgenomes are clearly distinct, with little evolution towards diploid stage. In terms of physical time, this can refer to events that have occurred within the last few million years. In this context, cotton has been defined a “neopolyploid” also by others (see for instance the discussion in Guo et al, 2014 (<https://doi.org/10.1534>), whose authors are certainly not newcomers to the polyploidy field: “Cotton and many other neopolyploid crops are genetically...”). Thus, in contrast to the reviewer’s statement, our use of the term “neopolyploid” mirrors its use by other expert authors in the same context, and thus does not indicate a general “lack of understanding of polyploid genomics and genome evolution” as the reviewer states. This said, we acknowledge that, in other contexts, researchers restrict the term to comprise only polyploids formed in the past few tens of thousands of years. To avoid confusion among the broad audience, in the revised text we use the term “relatively recent polyploids”.

R10. Finally, the manuscript is way too long (single-spaced, 18 pages!). It should be substantially reduced to focus on new findings and insights, perhaps following these suggestions.

Introduction: Lines 132-148: reduce this paragraph to a couple of sentences and move it as the last part of the previous paragraph.

Results and Discussion.

1. Combine “Chromosome-level assemblies...” and “Strong Conservation...” into one section and reduce the context (especially lengthy description of sequencing statistics) in half.
2. Remove Figure 2 (poor quality and did not support the conclusion) and reduce the section of “Gene-specific subgenome expression dominance” (a few sentences and reference citations) simply because the genome-wide data did not support the conclusion. These genes are “hand-picked” and moreover, the data did not support “expression dominance.”
3. Origin and domestication (4 pages!): the section should be reduced by two thirds. Many are tedious and redundant in literature (Lashermes et al. 1999, MGG; Cenci et al. 2012, Plant Mol. Biol.; Denoeud et al. 2014, Science; Scalabrin et al. 2020, Sci. Rep.;) and do not provide any new insights.
4. Kinship (3 pages!): reduced to half, some of which are redundant to the section of “origin

and domestication.” (also see above)

5. Introgression: Reduce it at least one third and combine the last (TE activity) into this section.

A10. In the third submission, the main text (without abstract, Methods, references, and figure legends) was 7,460 words and 6 display items. We have followed most of the reviewer’s suggestions, reducing it to approx. 4,360 words (including abstract) and 5 display items. Furthermore, to facilitate readability, the supplementary figures cited in the main text are now presented as extended data figures, and the supplementary tables are numbered in the order of citation (main text first, then supplementary).

Reviewer #2:

None

Reviewer #3:

R11. The updated version of the manuscript addresses the comments made by the reviewers. The article, although mainly focused on population studies, also provides a comprehensive overview of events that can take place after allotetraploidization (TE invasions, homoeologous exchanges, gene fractionation, biased gene expression) and shows which ones arose (or not) in *Coffea arabica*.

Below are some last comments:

About homoeologous gene expression in the caffeine biosynthesis pathway, the authors should cite a recent study (doi.org/10.1093/aob/mcac041) and compare their results to that study (they seem to be very similar).

A11. Thank you for the comment, indeed the reported results agree nicely. We have included the paper as reference in the manuscript and, since this reduces some of the novelty of our data, we moved the old Figure 2 into extended data.

R12. Also, the following article is describing some homoeologous exchanges in *C arabica* (doi: 10.1534/g3.116.030858). Here again, it would be useful to compare the events identified to the previously identified ones.

A12. Thank you, this is a good reference that is in general agreement with our work. However, technological differences make it difficult to compare the results of Lashermes et al. to our own, as their analysis was carried out using *Coffea canephora* reference and using that to identify SNPs that separate subgenomes. Since we establish the reference genomes for both progenitors and use longer shotgun reads, we believe we likely provide more accurate results. We understand the possible importance of the comparison, but we will leave it for future work. We have now added the suggested paper as a reference.

R13. Lines 112-116: the sentence is confusing and needs to be reformulated. What is “compatibility of the parental species”, how does it affect genomic shock (define?) and

subgenome dominance ? How does it oppose “gradual adaptation to a new ploidy level”? if I understood correctly, the authors mean that either there are drastic changes right after the polyploidization, or no “shock” and slow return to a diploidized state. It should be written more clearly.

A13. Thank you for the comment. We have eliminated the “compatibility” concept, which is ill-defined, listed the multiple phenomena that go under the name of “genomic shock”, and clarified that they do not show up, partially or totally, during “gradual adaptation”.

R14. Lines 381-385. The authors write that the pattern similarity suggests the biases originated in a common ancestor of all sampled accessions. But the next sentences state that there is a strong bias towards recent HE as well as to events shared by all individuals, suggesting that HE events are under strong selection. Is it possible that the pattern similarity is actually caused by selection rather than a common ancestor? How to distinguish between the two hypotheses?

A14. Thank you, we have clarified this in the revised text. The shared HE events exhibit shared breakpoints across the genomes (Extended data Fig. 4), so it is more likely that they happened in a common ancestor, rather than being the result of selection for the same set of genes.

R15. Additionally, the HE events are not easy to decipher from figures S31-36. It would be useful to provide a synthetic table, listing all detected events (with their positions) for all the accessions studied (data underlying figure S38?).

A15. Thank you, we have done this now and added it as new **Table S15**.

Decision Letter, third revision:

21st Nov 2023

Dear Dr. Albert,

Thank you for submitting your revised manuscript "The genome and population genomics of allopolyploid *Coffea arabica* reveal the diversification history of modern coffee cultivars" (NG-A60104R2). The reviewers find that the paper has improved in revision, and therefore we'll be happy in principle to publish it in Nature Genetics, pending minor revisions to satisfy the referees' final requests and to comply with our editorial and formatting guidelines.

Sincerely,
Wei

Wei Li, PhD
Senior Editor
Nature Genetics
New York, NY 10004, USA
www.nature.com/ng

Reviewer #3 (Remarks to the Author):

The authors addressed very convincingly the last comments from the reviewers.

Final Decision Letter:

23rd Feb 2024

Dear Dr. Albert,

I am delighted to say that your manuscript "The genome and population genomics of allopolyploid *Coffea arabica* reveal the diversification history of modern coffee cultivars" has been accepted for publication in an upcoming issue of Nature Genetics.

Your paper will be published online after we receive your corrections and will appear in print in the next available issue. You can find out your date of online publication by contacting the Nature Press Office (press@nature.com) after sending your e-proof corrections.

Please note that *Nature Genetics* is a Transformative Journal (TJ). Authors may publish their research with us through the traditional subscription access route or make their paper immediately open access through payment of an article-processing charge (APC). Authors will not be required to make a final decision about access to their article until it has been accepted. Find out more about Transformative Journals

Authors may need to take specific actions to achieve compliance with funder and institutional open access mandates. If your research is supported by a funder that requires immediate open access (e.g. according to Plan S principles) then you should select the gold OA route, and we will direct you to the compliant route where possible. For authors selecting the subscription publication route, the journal's standard licensing terms will need to be accepted, including <https://www.nature.com/nature-portfolio/editorial-policies/self-archiving-and-license-to-publish>. Those licensing terms will supersede any other terms that the author or any third party may assert apply to any version of the manuscript.

If you have not already done so, we invite you to upload the step-by-step protocols used in this manuscript to the Protocols Exchange, part of our on-line web resource, natureprotocols.com. If you complete the upload by the time you receive your manuscript proofs, we can insert links in your article that lead directly to the protocol details. Your protocol will be made freely available upon publication of your paper. By participating in natureprotocols.com, you are enabling researchers to more readily reproduce or adapt the methodology you use. [Natureprotocols.com](http://natureprotocols.com) is fully searchable, providing your protocols and paper with increased utility and visibility. Please submit your protocol to <https://protocolexchange.researchsquare.com/>. After entering your nature.com username and password you will need to enter your manuscript number (NG-A60104R3). Further information can be found at <https://www.nature.com/nature-portfolio/editorial-policies/reporting-standards#protocols>

Sincerely,
Wei

Wei Li, PhD

Senior Editor
Nature Genetics
New York, NY 10004, USA
www.nature.com/ng